# Neural Networks Learn Generic Multi-Index Models Near Information-Theoretic Limit

**Bohan Zhang**[*]
Peking University
2200010903@stu.pku.edu.cn

**Zihao Wang**[*]
Stanford University
zihaow@stanford.edu

**Hengyu Fu**
UC Berkeley
hengyuf@berkeley.edu

**Jason D. Lee**
UC Berkeley
jasondlee@berkeley.edu

## Abstract

In deep learning, a central issue is to understand how neural networks efficiently learn high-dimensional features. To this end, we explore the gradient descent learning of a general Gaussian Multi-index model $f(\boldsymbol{x}) = g(\boldsymbol{U}\boldsymbol{x})$ with hidden subspace $\boldsymbol{U} \in \mathbb{R}^{r \times d}$, which is the canonical setup to study representation learning. We prove that under generic non-degenerate assumptions on the link function, a standard two-layer neural network trained via layer-wise gradient descent can agnostically learn the target with $o_d(1)$ test error using $\widetilde{\mathcal{O}}(d)$ samples and $\widetilde{\mathcal{O}}(d^2)$ time. The sample and time complexity both align with the information-theoretic limit up to leading order and are therefore optimal. During the first stage of gradient descent learning, the proof proceeds via showing that the inner weights can perform a power-iteration process. This process implicitly mimics a spectral start for the whole span of the hidden subspace and eventually eliminates finite-sample noise and recovers this span. It surprisingly indicates that optimal results can only be achieved if the first layer is trained for more than $\mathcal{O}(1)$ steps. This work demonstrates the ability of neural networks to effectively learn hierarchical functions with respect to both sample and time efficiency.

## 1 Introduction

Modern large language models powered by deep neural networks Krizhevsky et al. (2012); He et al. (2016); Vaswani et al. (2023) have shown strong empirical success across many domains. It is believed that much of the effectiveness of the pretraining–finetuning paradigm comes from their ability to extract useful representations from data. In practice, neural networks often first capture important hidden features, which helps them to learn complex tasks more efficiently. This adaptability offers a clear performance advantage over conventional fixed feature methods, such as kernel techniques Wei et al. (2020); Allen-Zhu & Li (2020); Bai & Lee (2020).

Despite these empirical successes, the theoretical understanding of why neural networks can efficiently and adaptively learn important features from data remains largely incomplete. Significant effort has been devoted in this direction, with a series of works Damian et al. (2022); Ba et al. (2022); Berthier et al. (2024); Lee et al. (2024); Montanari & Urbani (2025); Montanari & Wang (2026) investigating how neural networks can uncover hidden features and subsequently learn single-index and multi-index models, where the labels depend on the $d$-dimensional data inputs only through their projection onto a one-dimensional or low-dimensional subspace, respectively. The study of learning single-index models via neural networks is relatively well developed. Notably, Lee et al. (2024) show that neural networks can successfully learn single-index models with $\widetilde{\mathcal{O}}(d)$ sample complexity, which is optimal up to leading order, while Montanari & Urbani (2025) provides a sharp DMFT theory analyzing the learning dynamics of single-index models using a two-layer neural network when the sample size and input dimension are proportional.

---

[*]Equal contribution. The order was determined by a coin flip.

However, a single-index model involves only one hidden feature, whereas in practice we often expect multiple distinct features, motivating the study of neural networks learning of multi-index models. On the contrary, positive results on neural networks learning of multi-index models remain rather limited. Note first that the information-theoretic limit for learning multi-index models is still $\Theta(d)$ samples, and there exist polynomial time algorithms Chen & Meka (2020); Troiani et al. (2025); Defilippis et al. (2025); Kovačević et al. (2025); Damian et al. (2025) that achieve this for generic generative-exponent-two multi-index models which covers almost all canonical examples. Nevertheless, in general, it is not clear whether neural networks can achieve this algorithmic limit and, if so, how. The earliest work on neural networks learning of multi-index models Damian et al. (2022) showed that standard two-layer neural networks can efficiently learn a large subclass of multi-index models with a sub-optimal $\widetilde{\Theta}(d^2)$ sample size. Subsequently, Abbe et al. (2022)Abbe et al. (2023a) studied the problem of neural network learning multi-index models with a staircase structure. However, their results are far from optimal and are restricted to special classes of functions. The closest attempt to the general problem is Montanari & Wang (2026), which shows that a two-layer network trained via gradient descent can weakly recover some of the features in a multi-index model using $\Theta(d)$ samples. Nevertheless, since the features are only weakly recovered and are not guaranteed to cover the entire span, their results do not establish that these features can be directly leveraged to successfully learn the targets in later stages of training. Therefore, the following remains open.

*Can neural networks trained via standard optimizers (e.g., gradient descent) achieve **efficient, near–information-theoretic-optimal** learning of generic multi-index models?*

## 1.1 MAIN CONTRIBUTIONS

In this work, we work on this and conclude it positively for generative-exponent-two, non-generative-staircase multi-index models. Specifically, we consider learning the target function $f(\boldsymbol{x}) = g(\boldsymbol{Ux})$ over the standard $d$-dimensional Gaussian input distribution, where $\boldsymbol{U} \in \mathbb{R}^{r \times d}$ is the hidden subspace and $g : \mathbb{R}^r \to \mathbb{R}$ is a polynomial of constant degree. This corresponds to the standard multi-index model with a polynomial link function, which generalizes the single-index model as the special case $r = 1$.

Our main result, Theorem 1, shows that under generic assumptions, a two-layer neural network trained with layer-wise gradient descent can agnostically learn the target function up to $o_d(1)$ test error using $\widetilde{\mathcal{O}}(d)$ samples and $\widetilde{\mathcal{O}}(d^2)$ training time[1]. These sample and time complexities are information-theoretically optimal up to leading order, thereby demonstrating the adaptivity and effectiveness of neural network learning. Moreover, our analysis applies to generic activation functions and a broad subclass of loss functions, including but not limited to square loss, $\ell^1$ loss, the Huber loss, and the pseudo-Huber loss. We emphasize that the assumption on the link function in our work is rather mild and generic, as all polynomials possess a generative leap exponent of two Chen & Meka (2020); Damian et al. (2025) and the loss function can be interpreted as a form of data preprocessing. The idea of data preprocessing has a rich literature in spectral methods and so on, but rarely considered in theoretical analysis of neural networks learning. The only substantial assumption in this work is that the target function has no generative-staircase structure. We acknowledge that this assumption is important and it is not clear whether and how to extend our results to the case with generative-staircase structure within a neural network framework. For further discussion, please see Section 2.1.

Our proof begins by observing that near initialization, the gradient descent dynamics is nearly independent across different neurons, with each neuron evolving similarly to power method iterations applied to the local Hessian. The local Hessian exhibits large eigenvalues corresponding to the signal, namely the hidden subspace, as well as smaller eigenvalues arising from the noise because of the finite sample size. The key insight is that if this power iteration approximation works for too long, all features will align with the directions of the largest eigenvalues, that is, the most dominant directions, while ignoring the others, leading to failure of learning. However, if the number of training steps is too small, those noisy directions cannot be sufficiently eliminated, particularly when the sample size is nearly proportional to the input dimension. Therefore, the optimal strategy is to stop training the first layer at an appropriately chosen intermediate time. The features will still span the entire subspace while the noise is simultaneously eliminated.

---

[1]We use $\widetilde{\mathcal{O}}(\cdot)$ to hide those $d^{o_d(1)}$ factors.

We note that we use a standard initialization scheme for the neural network parameters, without using a spectral initialization or a hot start as in Mignacco et al. (2021); Chen & Shen (2025). The neural networks can only learn the right features from and only from gradient descent, trained for a divering number of steps. This time scale requirement rule out all the approaches based on DMFT or state evolution, such as Agoritsas et al. (2018); Ba et al. (2022); Celentano et al. (2025); Gerbelot et al. (2023); Berthier et al. (2024) which requires either information/generative exponent one or a hot start/spectral initialization, to guarantee the learning timescale can be made constant. We also highlight that the power iteration approximation does not work if the learning rate is too small. It is important to choose a moderately large learning rate. For a more detailed discussion, please see Section 4.

We defer the detailed discussion of related works to Appendix A.

## 1.2 NOTATIONS

We shall use $X \lesssim Y$ to denote $X \leq CY$ for some absolute positive constant $C$ and $X \gtrsim Y$ is defined analogously. We also use the standard big-O notations: $\Theta(\cdot)$, $\mathcal{O}(\cdot)$ and $\Omega(\cdot)$ to only hide absolute positive constants. In addition, we use $\widetilde{\mathcal{O}}$, $\widetilde{\Theta}$ and $\widetilde{\Omega}$ to hide all the $d^{o_d(1)}$ terms throughout the paper. Let $[k] = \{1, 2, \ldots, k\}$ for $k \in \mathbb{N}^+$ and $[a; b] = \{a, \ldots, b\}$ for $a, b \in \mathbb{N}^+$. For a vector $\boldsymbol{v}$, denote by $\|\boldsymbol{v}\|_p := (\sum_i |v_i|^p)^{1/p}$ the $\ell^p$ norm. When $p = 2$, we omit the subscript for simplicity. For a matrix $\boldsymbol{A}$, let $\|\boldsymbol{A}\|_{\mathrm{op}}$ and $\|\boldsymbol{A}\|_F$ be the spectral norm and Frobenius norm, respectively. We use $\lambda_{\max}(\cdot)$ and $\lambda_{\min}(\cdot)$ to denote the maximal and the minimal eigenvalue of a real symmetric matrix.

## 2 PROBLEM SETUP

Our purpose is to learn the target $f^* : \mathbb{R}^d \to \mathbb{R}$, where $\mathbb{R}^d$ is the input domain equipped with the standard Gaussian distribution $\mathsf{N}(\boldsymbol{0}, \boldsymbol{I}_d)$. Our target follows a multi-index model $f^*(\boldsymbol{x}) = g^*(\boldsymbol{U}\boldsymbol{x})$ with hidden directions $\boldsymbol{U} \in \mathbb{R}^{r \times d}$ and the link function $g^* : \mathbb{R}^r \to \mathbb{R}$. We assume $\boldsymbol{U}$ has orthogonal unit rows without loss of generality.

**Assumption 1.** $g^*(\cdot)$ *is a polynomial of degree* $p$. *Furthermore,* $\mathbb{E}_{\boldsymbol{z} \sim \mathsf{N}(\boldsymbol{0}, \boldsymbol{I}_r)}[g^{*2}(\boldsymbol{z})] \leq C$ *where* $C$ *is an absolute constant.*

We treat $r, p$ as absolute constants throughout the paper. Next, we consider the model as a standard two-layer neural network

$$f_{\boldsymbol{\Theta}}(\boldsymbol{x}) = \sum_{j=1}^{m} a_j \sigma(\boldsymbol{w}_j^\mathsf{T} \boldsymbol{x} + b_j) \tag{1}$$

Here we use $\boldsymbol{\Theta} := \{a_j, b_j, \boldsymbol{w}_j\}_{j=1}^m$ to denote all the parameters. We make the following assumption on the activation function.

**Assumption 2.** $\sigma(\cdot) \in C^3(\mathbb{R})$ *is a smooth activation function satisfying* $\sigma'(0) = 0$, $\sigma''(0) = 1$ *and* $|\sigma^{(3)}(\cdot)| \leq M$ *where* $M$ *is an absolute constant.*

Let $\mathcal{D} := \{(\boldsymbol{x}_i, y_i)\}_{i=1}^n$ be one dataset, where $\boldsymbol{x}_i \in \mathbb{R}^d$ are drawn i.i.d. from distribution $\mathsf{N}(\boldsymbol{0}, \boldsymbol{I}_d)$ and $y_i = f^*(\boldsymbol{x}_i)$. The empirical and population loss is given respectively by

$$\widehat{\mathcal{L}}_{\mathcal{D}}(\boldsymbol{\Theta}) = \frac{1}{n} \sum_{i=1}^{n} \ell(f_{\boldsymbol{\Theta}}(\boldsymbol{x}_i), y_i), \qquad \mathcal{L}_{\mathcal{D}}(\boldsymbol{\Theta}) = \mathbb{E}\big[\ell(f_{\boldsymbol{\Theta}}(\boldsymbol{x}), y)\big] \tag{2}$$

where $\ell = \ell(t, y) : \mathbb{R} \times \mathbb{R} \to \mathbb{R}$ is a general scalar loss function. We make the following assumptions on the loss function.

**Assumption 3.** $\ell(\cdot, \cdot) \in C^2(\mathbb{R}^2)$, $\ell(t, y)_{|t=y} = 0$ *and that it is convex in its first argument. Furthermore, we assume*

$$\left|\frac{\partial}{\partial t}\ell(t, y)\right| \leq \mathrm{L} \quad \text{and} \quad \left|\frac{\partial^2}{\partial^2 t}\ell(t, y)\right| \leq \mathrm{L} \tag{3}$$

*for all* $t, y \in \mathbb{R}$ *where* $\mathrm{L}$ *is a positive absolute constant.*

We also consider a slightly relaxed version of Assumption 3 for some specific activation functions. This assumption includes square loss as a special case.

**Assumption 3′.** *We weaken the assumption in Assumption 3 by allowing the first derivative to grow linearly with the input*

$$\left| \frac{\partial}{\partial t} \ell(t, y) \right| \leq \mathrm{L}(1 + |t| + |y|) \quad \text{for all } t, y \in \mathbb{R}. \tag{4}$$

Denote $\ell'(0, y_i) := \frac{\partial}{\partial t} \ell(t, y_i)_{|t=0}$ as $\ell_i$.

**Assumption 4.** *We assume $\mathbb{E}[\ell_i] = 0$.*

We define

$$\widehat{\boldsymbol{\Sigma}}_\ell := \frac{1}{n} \sum_{i=1}^{n} \ell_i \cdot \boldsymbol{x}_i \boldsymbol{x}_i^\mathsf{T} \tag{5}$$

and also its population counterpart $\boldsymbol{\Sigma}_\ell := \mathbb{E}[\ell'(0, y) \cdot \boldsymbol{x}\boldsymbol{x}^\mathsf{T}]$. Recall that $\ell_i$ is a function of $y_i$ and that $y_i$ is itself a function of $\boldsymbol{U}\boldsymbol{x}_i$. Therefore, $\ell_i$ is a function of $\boldsymbol{U}\boldsymbol{x}_i$ and we conclude that $\mathrm{rank}(\boldsymbol{\Sigma}_\ell) \leq r$ via invoking Stein's Lemma and performing integration by parts.

**Assumption 5.** *The rank is full, i.e., $\mathrm{rank}(\boldsymbol{\Sigma}_\ell) = r$.*

*Furthermore, denote the non-zero eigenvalues as $\lambda_r \leq \cdots \leq \lambda_1$. We will assume $\lambda_r > 0$ and set $\lambda_1 = 1$ without loss of generality.*

In our setting, we assume $\lambda_r > 0$ to ensure that $\boldsymbol{\Sigma}_\ell$ is not degenerate in its support and provides a meaningful directional signal during optimization. We will denote and track the condition number $\kappa := \frac{\lambda_1}{\lambda_r}$ but will still assume $\kappa = \Theta_d(1)$.

## 2.1 DISCUSSION OF ASSUMPTIONS

We note that the first four assumptions are quite standard in theory literature. Assumption 1 is fairly generic. In Assumption 2, the key requirement are $\sigma'(0) = 0$, ensuring that the activation behaves locally as a quadratic. The smoothness assumptions can be relaxed with a more refined analysis. For Assumption 3 or 3′, convexity in the first argument of the loss is essential to guarantee that the second layer weights converge to its minimizer during the second stage of training. We note that this condition holds for a large subclass of commonly used losses such as square loss, $\ell^1$ loss and Huber loss. The bounds on the derivatives are not critical and can be replaced by a polynomial growth condition under a more careful analysis, since most of our data remain in a bounded regime and the outliers are rare. Finally, Assumption 4 is also standard and could be removed with either a more refined argument or an additional data pre-processing step.

The central assumption in our analysis is Assumption 5. Informally, this assumption requires that the second-order information of the data remain non-degenerate after applying the pre-processing map $y \mapsto \ell'(0, y)$. We admit that for one fixed loss function $\ell$, it is straightforward to construct a target task such that the second-order information of $\ell'(0, y)$ becomes degenerate. In practice, however, the target task is typically fixed, and one has the flexibility to choose among different loss functions to learn it. Recent studies on generative exponents and learning thresholds Troiani et al. (2025); Damian et al. (2024); Kovačević et al. (2025); Damian et al. (2025); Defilippis et al. (2025) show that for a large subclass of common tasks, including all polynomial targets with non-generative-staircase structure, there exists a pre-processing function that ensures the second-order information is non-degenerate, and importantly, such a function need not be highly specialized. Consequently, we expect that $\ell'(0, y)$ exhibits non-degenerate second-order information for at least some of those commonly used loss functions in practice, and at least this already includes all targets in Damian et al. (2022). We will expand on this in Section 5.

## 2.2 TRAINING ALGORITHM

We introduce our layer-wise gradient descent algorithm.

At initialization, we set $b_j$ as zero and impose a symmetric structure on $a_j$ and $\boldsymbol{w}_j$. Specifically, let the width $m$ be even, and for all $j \in [m]$ we set

$$a_j = -a_{m-j}, \quad \boldsymbol{w}_j^{(0)} = \boldsymbol{w}_{m-j}^{(0)}. \tag{6}$$

For the first half of indices , let $a_j$ be i.i.d. Rademacher variables: $a_j \overset{\text{i.i.d.}}{\sim} \text{Uniform}\{-1, +1\}$. For the first half of the features $\boldsymbol{w}_j$, we initialize them independently and uniformly on the sphere of radius $\epsilon_0$.

The network is trained via layer-wise gradient descent with sample splitting, a common approach in theoretical analysis such as Wang et al. (2023); Nichani et al. (2025). To start with, we generate two independent datasets $\mathcal{D}_i$ where $i = 1, 2$ and each consisting of $n$ independent samples. The training algorithm proceeds as follows. we first train $\boldsymbol{W}$ via gradient descent for $\mathrm{T}_1$ steps on the empirical loss $\widehat{\mathcal{L}}_{\mathcal{D}_1}(\boldsymbol{\Theta})$, then reinitialize $\boldsymbol{b}$ and train $\boldsymbol{a}$ via gradient descent for $\mathrm{T}_2$ steps on the empirical loss $\widehat{\mathcal{L}}_{\mathcal{D}_2}(\boldsymbol{\Theta})$. The pseudocode of this procedure is given below.

---
**Algorithm 1** Layer-wise Gradient Descent
---
**Require:** Learning rates $\eta_1, \eta_2$, weight decay $\beta_1, \beta_2$, initialization level $\epsilon_0$, number of steps $\mathrm{T}_1, \mathrm{T}_2$.

1: Initialize $\boldsymbol{a}, \boldsymbol{b}, \boldsymbol{W}$.
2: **Train $\boldsymbol{W}$ on dataset $\mathcal{D}_1$**
3: **for** $t = 1$ **to** $\mathrm{T}_1 - 1$ **do**
4: $\quad \boldsymbol{W}^{(t)} \leftarrow \boldsymbol{W}^{(t-1)} - \eta_1 [\nabla_{\boldsymbol{W}} \widehat{\mathcal{L}}_{\mathcal{D}_1}(\boldsymbol{\Theta}^{(t-1)}) + \beta_1 \boldsymbol{W}^{(t-1)}]$
5: **end for**
6: $\boldsymbol{W}^{(\mathrm{T}_1)} \leftarrow \boldsymbol{W}^{(\mathrm{T}_1-1)} - \eta_1 \epsilon_0^{-1} [\nabla_{\boldsymbol{W}} \widehat{\mathcal{L}}_{\mathcal{D}_1}(\boldsymbol{\Theta}^{(\mathrm{T}_1-1)}) + \beta_1 \epsilon_0 \boldsymbol{W}^{(\mathrm{T}_1-1)}]$
7: **Re-initialize**
8: $b_i^{(\mathrm{T}_1)} \sim_{iid} \text{Unif}([-3, 3]), \ i \in [m]$
9: $\boldsymbol{a}^{(\mathrm{T}_1)} \leftarrow \boldsymbol{a}^{(0)}$
10: $\boldsymbol{\Theta}^{(\mathrm{T}_1)} \leftarrow (\boldsymbol{a}^{(\mathrm{T}_1)}, \boldsymbol{W}^{(\mathrm{T}_1)}, \boldsymbol{b}^{(\mathrm{T}_1)})$
11: **end**
12: **Train $\boldsymbol{a}$ on dataset $\mathcal{D}_2$**
13: **for** $t = \mathrm{T}_1 + 1$ **to** $\mathrm{T}_1 + \mathrm{T}_2$ **do**
14: $\quad \boldsymbol{a}^{(t)} \leftarrow \boldsymbol{a}^{(t-1)} - \eta_2 [\nabla_{\boldsymbol{a}} \widehat{\mathcal{L}}_{\mathcal{D}_2}(\boldsymbol{\Theta}^{(t-1)}) + \beta_2 \boldsymbol{a}^{(t-1)}]$
15: **end for**
16: Note $\mathrm{T} = \mathrm{T}_1 + \mathrm{T}_2$
17: **return** Prediction function $f_{\boldsymbol{\Theta}^{(\mathrm{T})}}(\cdot) : \boldsymbol{x} \to \langle \boldsymbol{a}^{(\mathrm{T})}, \sigma(\boldsymbol{W}^{(\mathrm{T}_1)}\boldsymbol{x} + \boldsymbol{b}^{(\mathrm{T}_1)}) \rangle$

---

## 3 MAIN RESULTS

Our goal is to show that the network defined in (1) trained via Algorithm 1 can efficiently learn multi-index models near the information-theoretic limit.

We use $D$ to denote an arbitrarily large absolute constant and, for technical convenience, assume $n, m \lesssim d^{D/4}$ throughout the paper. Before stating the main results, we further assume below that finite linear combinations of the activation can approximate any one-dimensional monomial to arbitrary precision.

**Assumption 6.** *We assume the activation function has the monomial approximation property with exponent $\beta \geq 0$. That is, for every integer $k \geq 0$ and every $\rho > 0$ there exists a bounded weight function $v_k^{(\rho)} : \{\pm 1\} \times [-3, 3] \to \mathbb{R}$ with*

$$\|v_k^{(\rho)}\|_\infty \leq V_{k,\rho} \lesssim_{\sigma,k} \rho^{-\beta} \tag{7}$$

*where $\lesssim_{\sigma,k}$ hides absolute constants depending solely on $k$ and activations such that*

$$\sup_{|z| \leq 1} \left| \mathbb{E}_{a,b}[v_k^{(\rho)}(a, b) \, \sigma(az + b)] - z^k \right| \leq \rho \tag{8}$$

*where $a$ is a Rademacher RV i.e. $\mathbb{P}(a = +1) = \mathbb{P}(a = -1) = \frac{1}{2}$ and $b \sim \text{Unif}[-3, 3]$ independent from $a$.*

We treat $\beta$ as an absolute constant throughout. We remark that this assumption is rather mild, which only requires that a two-layer network with one-dimensional input can approximate any monomial to arbitrary precision using an expectation form. Moreover, we conjecture that for most activations one can take $\beta = 0$ and further set $\rho = 0$, as suggested by Barron (1993); E et al. (2021). For concrete examples, Wang et al. (2023) shows that we have $\beta = 0$ for ReLU activations in their Appendix D.6 and we also show in our Appendix E.3 that for the following activation

$$\sigma(t) = \begin{cases} 2|t| - 1 & |t| \geq 1 \\ t^2 & |t| < 1 \end{cases} \tag{9}$$

we provably have $\beta = 0$ and thus can set $\rho = 0$ as well.

We also require a small initialization scale, which is our last assumption.

**Assumption 7.** *We set the initialization level $\epsilon_0 \leq \frac{1}{C_\epsilon m n^{1/2} d^{7/2}} \left(\frac{4}{5}\right)^{\mathrm{T}_1}$, where $C_\epsilon$ is a universal constant.*

We note that $\epsilon_0$ is only required to be polynomially small provided $\mathrm{T}_1 = o(\log d)$. With our assumptions in place, we are ready to state our main theorem.

**Theorem 1.** *Under assumptions 1, 2, 3, 4, 5, 6, 7, or under the additional assumption that $\beta = 0$ and assumptions 1, 2, 3', 4, 5, 6, 7, we have the following result: For any first stage training time $\mathrm{T}_1 = o(\log d)$ and any sample size*

$$n \gtrsim d \log d \, \mathrm{T}_1^2 \kappa^{2\mathrm{T}_1} + d^{1 + \frac{1}{\mathrm{T}_1}} \kappa^2 \tag{10}$$

*there exists a proper choice of the hyperparameters and second stage training time $\mathrm{T}_2$ such that with probability at least $1 - \mathcal{O}(d^{-D/2})$, the Algorithm 1 returns*

$$\mathcal{L}(\boldsymbol{\Theta}^{(\mathrm{T})}) \lesssim \kappa^{4p\mathrm{T}_1} (\log d)^{4p+1} \left(\frac{1}{\sqrt{m}} + \frac{1}{\sqrt{n}}\right)^{\frac{1}{\beta+1}} \tag{11}$$

In short, Theorem 1 states that we can learn the target up to the vanishing test error with the width $m = \widetilde{\Theta}(1)$ and the sample size $n \gtrsim d \log d \, \mathrm{T}_1^2 \kappa^{2\mathrm{T}_1} + d^{1 + \frac{1}{\mathrm{T}_1}} \kappa^2$. We specify those hyperparameters later in the statement of Theorem 2 in Appendix D. We further remark that the sample size requirement $n \gtrsim d \log d \, \mathrm{T}_1^2 \kappa^{2\mathrm{T}_1} + d^{1+1/\mathrm{T}_1} \kappa^2$ guarantees that the first-stage training learns the correct representations, whereas the error term $\kappa^{4p\mathrm{T}_1} (\log d)^{4p+1} \left(\frac{1}{\sqrt{m}} + \frac{1}{\sqrt{n}}\right)^{\frac{1}{\beta+1}}$ is contributed purely by the second-stage training. We note that under the first set of assumptions, the error bound can be refined to $\kappa^{2p\mathrm{T}_1} (\log d)^{2p+1} \left(\frac{1}{\sqrt{m}} + \frac{1}{\sqrt{n}}\right)^{\frac{1}{\beta+1}}$.

Via balancing two terms in our sample complexity requirement

$$n \gtrsim d \log d \, \mathrm{T}_1^2 \kappa^{2\mathrm{T}_1} + d^{1 + \frac{1}{\mathrm{T}_1}} \kappa^2 \tag{12}$$

the optimal dependence shall be derived from setting $\mathrm{T}_1 = \left\lceil \sqrt{\frac{\log d}{\log \kappa}} \right\rceil$. Note that below we use $\widetilde{\mathcal{O}}$, $\widetilde{\Theta}$ and $\widetilde{\Omega}$ to hide all the $d^{o_d(1)}$ terms.

**Corollary 3.1.** *Consider $\mathrm{T}_1 = \left\lceil \sqrt{\frac{\log d}{\log \kappa}} \right\rceil$. Under this, we further require*

$$m = (\log d)^{8p(\beta+1)+1} d^{8p(\beta+1)\sqrt{\frac{\log \kappa}{\log d}}} \quad and \quad n = C\kappa^2 (\log d)^2 d^{1 + 2\sqrt{\frac{\log \kappa}{\log d}}} \tag{13}$$

*where $C$ is a large universal constant.*

*Then, there exists a proper choice of hyperparameters that the Algorithm 1 returns $\mathcal{L}(\boldsymbol{\Theta}^{(\mathrm{T})}) = o_d(1)$ with $\mathrm{T}_2 = \widetilde{\Theta}(1)$ and probability $1 - \mathcal{O}(d^{-D/2})$.*

We note that the overall time complexity of Algorithm 1 is $\Theta(nmd\mathrm{T})$. Under the setup of Corollary 3.1, we have $\mathrm{T} = \widetilde{\Theta}(1)$, $m = \widetilde{\Theta}(1)$, and $n = \widetilde{\Theta}(d)$, yielding an overall time of $\widetilde{\Theta}(d^2)$. This is also optimal up to leading order, since any algorithm must spend at least $\Omega(nd) = \Omega(d^2)$ time to read the full dataset.

Compared with Damian et al. (2022), we note that our results exactly and completely recover the results in Damian et al. (2022) when we set $T_1 = 1$. In this regime, successful learning is guaranteed only with a larger sample size $n = \widetilde{\Theta}(d^2)$, and Dandi et al. (2025) further indicates that $n = \widetilde{\Theta}(d^2)$ is necessary when $T_1 = 1$. Our results improve upon Damian et al. (2022) and all previous works by managing to execute a careful analysis of the multi-step gradient-descent dynamics on the empirical loss over a divering time horizon. We provide the details in Section 4.

## 4 OUTLINE OF THE PROOF

### 4.1 GRADIENT DESCENT CAN MIMIC POWER ITERATIONS

To begin, we compute the gradient with respect to the feature $\boldsymbol{w}_j$ as follows.

$$\nabla_{\boldsymbol{w}_j}\widehat{\mathcal{L}}_n(\boldsymbol{\Theta}^{(t)}) = \frac{1}{n}\sum_{i=1}^{n}\ell'(f_{\boldsymbol{\Theta}^{(t)}}(\boldsymbol{x}_i), y_i)\, a_j\sigma'\left(\boldsymbol{w}_j^{(t)\mathsf{T}}\boldsymbol{x}_i\right)\boldsymbol{x}_i \tag{14}$$

Our initialization ensures that the network output is zero at the beginning of the training. Therefore, within a moderate large time horizon, we expect $f_{\boldsymbol{\Theta}^{(t)}}(\boldsymbol{x})$ to remain near zero. Furthermore, recall that $\|\boldsymbol{w}_j^{(0)}\| = o_d(1)$. For small $\boldsymbol{w}_j^{(t)\mathsf{T}}\boldsymbol{x}_i$ we have

$$\sigma'\left(\boldsymbol{w}_j^{(t)\mathsf{T}}\boldsymbol{x}_i\right) = \sigma''(0)\boldsymbol{w}_j^{(t)\mathsf{T}}\boldsymbol{x}_i + o_d(1) \tag{15}$$

using $\sigma'(0) = 0$ and substituting these yields

$$\nabla_{\boldsymbol{w}_j}\widehat{\mathcal{L}}_n(\boldsymbol{\Theta}^{(t)}) = \frac{1}{n}\sum_{i=1}^{n}\ell'(0, y_i)a_j\sigma''(0)\boldsymbol{w}_j^{(t)\mathsf{T}}\boldsymbol{x}_i\boldsymbol{x}_i +\text{ smaller order terms.} \tag{16}$$

Letting $\ell_i := \ell'(0, y_i)$, under a proper learning rate and weight decay, the dynamics can be expressed as

$$\boldsymbol{w}_j^{(t+1)} \propto \left(\frac{1}{n}\sum_{i=1}^{n}\ell_i\boldsymbol{x}_i\boldsymbol{x}_i^{\mathsf{T}}\right)\boldsymbol{w}_j^{(t)} = \widehat{\boldsymbol{\Sigma}}_\ell\boldsymbol{w}_j^{(t)}. \tag{17}$$

We emphasize that the above approximation works whenever $f_{\boldsymbol{\Theta}^{(t)}}(\boldsymbol{x})$ and $\boldsymbol{w}_j^{(t)}$ both remain small. Under those approximations, the neurons decouple from each other.

Under these approximations, the dynamics can be viewed as a power method iteration with respect to $\widehat{\boldsymbol{\Sigma}}_\ell$. Recall that $\widehat{\boldsymbol{\Sigma}}_\ell$ is the empirical version of $\boldsymbol{\Sigma}_\ell$, satisfying $\|\widehat{\boldsymbol{\Sigma}}_\ell - \boldsymbol{\Sigma}_\ell\|_{\text{op}} = \Theta(\sqrt{d/n})$. Moreover, from the assumptions, the population one admits the eigendecomposition

$$\boldsymbol{\Sigma}_\ell = \sum_{i=1}^{r}\lambda_i\boldsymbol{u}_i\boldsymbol{u}_i^{\mathsf{T}}, \tag{18}$$

where $\{\boldsymbol{u}_i\}_{i=1}^{r}$ are orthonormal unit vectors spanning $\boldsymbol{U}$, with eigenvalues ordered as $1 = \lambda_1 \geq \cdots \geq \lambda_r = 1/\kappa$, in accordance with our assumptions. We further assume a constant eigenvalue gap only in this section for convenience, namely $\min_{i\neq j}|\lambda_i - \lambda_j| \geq c$ for some universal constant $c$; We shall emphasize that we manage to avoid this assumption in our final results via a more fine-grained analysis (see Appendix C).

**Eigenvalues and eigenvectors of $\widehat{\boldsymbol{\Sigma}}_\ell^{\mathsf{T}}$.** The empirical covariance $\widehat{\boldsymbol{\Sigma}}_\ell$ then admits the decomposition

$$\widehat{\boldsymbol{\Sigma}}_\ell = \sum_{i=1}^{d}\widehat{\lambda}_i\widehat{\boldsymbol{u}}_i\widehat{\boldsymbol{u}}_i^{\mathsf{T}}, \tag{19}$$

where $\{\widehat{\boldsymbol{u}}_i\}_{i=1}^{d}$ forms an orthonormal unit basis of $\mathbb{R}^d$. We note that we have $|\widehat{\lambda}_i - \lambda_i| = \Theta(\sqrt{d/n})$ and $\|\widehat{\boldsymbol{u}}_i - \boldsymbol{u}_i\| = \Theta(\sqrt{d/n})$ for $i \leq r$. For the others we have $|\widehat{\lambda}_i| = \Theta(\sqrt{d/n})$. This decomposition makes it straightforward to analyze powers of $\widehat{\boldsymbol{\Sigma}}_\ell$

$$\widehat{\boldsymbol{\Sigma}}_\ell^{\mathsf{T}} = \sum_{i=1}^{d}\widehat{\lambda}_i^{\mathsf{T}}\widehat{\boldsymbol{u}}_i\widehat{\boldsymbol{u}}_i^{\mathsf{T}}, \tag{20}$$

where T denotes the number of gradient descent steps in the first stage. For a random initialization $\boldsymbol{w}_j^{(0)}$, the contribution from the noise components is

$$\left\| \sum_{i=r+1}^{d} \widehat{\lambda}_i^{\mathsf{T}} \widehat{\boldsymbol{u}}_i \widehat{\boldsymbol{u}}_i^{\mathsf{T}} \boldsymbol{w}_j^{(0)} \right\| = \Theta\left( \left( \sqrt{d/n} \right)^{\mathsf{T}} \left\| \boldsymbol{w}_j^{(0)} \right\| \right), \tag{21}$$

while the signal contribution along direction $\widehat{\boldsymbol{u}}_i$ is

$$\widehat{\lambda}_i^{\mathsf{T}} |\widehat{\boldsymbol{u}}_i^{\mathsf{T}} \boldsymbol{w}_j^{(0)}| = \Theta\left( \lambda_i^{\mathsf{T}} \frac{1}{\sqrt{d}} \left\| \boldsymbol{w}_j^{(0)} \right\| \right) \gtrsim \kappa^{-\mathsf{T}} \frac{1}{\sqrt{d}} \left\| \boldsymbol{w}_j^{(0)} \right\|. \tag{22}$$

To ensure that the signal is at least comparable with the noise, we require

$$\frac{\kappa^{-\mathsf{T}}}{\sqrt{d}} \gtrsim \left( \sqrt{d/n} \right)^{\mathsf{T}} \tag{23}$$

which corresponds to the condition $n \gtrsim d^{1+1/\mathsf{T}} \kappa^2$.

Assuming the noise term is negligible, for $k \in [r]$ we now focus on the signal strength along direction $\boldsymbol{u}_k$ instead of $\widehat{\boldsymbol{u}}_k$

$$\boldsymbol{u}_k^{\mathsf{T}} \widehat{\boldsymbol{\Sigma}}_\ell^{\mathsf{T}} \boldsymbol{w}_j^{(0)} \approx \widehat{\lambda}_k^{\mathsf{T}} \boldsymbol{u}_k^{\mathsf{T}} \widehat{\boldsymbol{u}}_k \widehat{\boldsymbol{u}}_k^{\mathsf{T}} \boldsymbol{w}_j^{(0)} + \sum_{\substack{1 \leq i \leq r \\ i \neq k}} \widehat{\lambda}_i^{\mathsf{T}} \boldsymbol{u}_k^{\mathsf{T}} \widehat{\boldsymbol{u}}_i \widehat{\boldsymbol{u}}_i^{\mathsf{T}} \boldsymbol{w}_j^{(0)}. \tag{24}$$

Here we require the first term to dominate the second so that the signal strength along the direction $\boldsymbol{u}_k$ is well aligned with that along $\widehat{\boldsymbol{u}}_k$. For a typical $\boldsymbol{w}_j^{(0)}$, the magnitude of the first term is at least $\Theta\left( \kappa^{-\mathsf{T}}/\sqrt{d} \left\| \boldsymbol{w}_j^{(0)} \right\| \right)$, while we have

$$\left| \sum_{\substack{1 \leq i \leq r \\ i \neq k}} \widehat{\lambda}_i^{\mathsf{T}} \boldsymbol{u}_k^{\mathsf{T}} \widehat{\boldsymbol{u}}_i \widehat{\boldsymbol{u}}_i^{\mathsf{T}} \boldsymbol{w}_j^{(0)} \right| = \mathcal{O}\left( \left\| \boldsymbol{w}_j^{(0)} \right\| / \sqrt{n} \right) \tag{25}$$

using the standard estimates $|\boldsymbol{u}_k^{\mathsf{T}} \widehat{\boldsymbol{u}}_i| = \mathcal{O}(\sqrt{d/n})$ and $|\widehat{\boldsymbol{u}}_i^{\mathsf{T}} \boldsymbol{w}_j^{(0)}| = \Theta\left( \left\| \boldsymbol{w}_j^{(0)} \right\| \sqrt{1/d} \right)$. Therefore, it suffices to take $n \gtrsim d \log d \mathsf{T}^2 \kappa^{2\mathsf{T}}$. We emphasize and highlight that it is quite interesting to observe that two conflicting factors are involved at the same time. If T is too small, the noise is not sufficiently suppressed; whereas if T is too large, the weakest signal direction vanishes exponentially and becomes undetectable. Therefore, we have to balance these two effects to find the optimal T, which is interestingly found to be of the order $\Theta(\sqrt{\log d})$.

Therefore, assuming $n \gtrsim d \log d \mathsf{T}^2 \kappa^{2\mathsf{T}} + d^{1+\frac{1}{\mathsf{T}}} \kappa^2$, the learned feature approximately lies in and successfully spans the whole $\boldsymbol{U}$ subspace. We can then conclude that the second stage is able to learn any multi-index model with hidden subspace $\boldsymbol{U}$ from the learned features via using a standard kernel method argument. The complexity of this learning scales independent with $d$ and polynomial with $\kappa^{\mathsf{T}}$, as some compensation is required to learn the weakest direction, which is still $\widetilde{\mathcal{O}}(1)$ when $\mathsf{T} = \Theta(\sqrt{\log d})$. For a concrete explanation, we refer to the proof roadmap of the second stage in Appendix D.

### 4.2 COMPARISON WITH THE EXISTING APPROACHES

First of all, we note that our results cannot be recovered using a DMFT or state evolution framework, such as Agoritsas et al. (2018); Ba et al. (2022); Celentano et al. (2025); Gerbelot et al. (2023); Berthier et al. (2024) and many more. Essentially, in order to learn those generative-exponent two directions under the sample size $\widetilde{\Theta}(d)$, one needs a diverging time horizon to escape the initial saddle point if there is no hot start or explicit spectral initialization. Under the DMFT/state evolution propotional limit, the neurons shall not escape the initial stationary point.

Second, our methods differ from the standard power iteration in matrix/tensor PCA problems Balzano et al. (2011); Montanari & Richard (2014). If we enforce the matrix power iteration approximation for long time, the neurons shall collapse to one single direction, which prevents a successful

learning. Instead, we force a early stop (but still longer than constant) to the power iteration to ensure the neurons span the entire subspace. In practice, we believe neural networks can adaptively find the optimal stopping time when the second layer is trained together with the first layer. We provide numerical validations of our main results in Section B.

## 5 FURTHER DISCUSSIONS

We here further explain our Assumption 5, which essentially requires our target function has generative exponent two without generative-staircase structure. To start with, we briefly introduce the concept of generative exponents. The so-called generative exponents has a long standing literature (see related works) and is introduced in its most general form in recent work Damian et al. (2025), which sharply characterize the complexity of efficiently learning Gaussian multi-index models up to leading order. In the following, we use $\mathrm{h}_k$ to denote degree $k$ Hermite polynomials and $\mathrm{He}_k$ to denote degree $k$ Hermite tensors. A short introduction to these objects is provided in Appendix E.1.

We denote $\boldsymbol{z} = \boldsymbol{U}^\top \boldsymbol{x} \sim \mathsf{N}(\boldsymbol{0}, \boldsymbol{I}_r)$ as the hidden directions and $y$ as the model output. Denote further $S$ as a subspace of the $\boldsymbol{U}$ subspace, and let $\boldsymbol{z}_S \in S$ denote the orthogonal projection of $\boldsymbol{z}$ onto $S$. The generative exponent given a subspace $S$, $k_S^* = k_S^*(\boldsymbol{z}, y)$, is defined as the largest $k$ such that $\mathbb{E}[\mathcal{T}(y, \boldsymbol{z}_S)p(\boldsymbol{z}_{S\perp})] = 0$ for any measurable function $\mathcal{T}$ and any polynomial $p$ of degree lower than $k$. This indicates that the tensor $\mathbb{E}[\mathcal{T}^*(y, \boldsymbol{z}_S)\mathrm{He}_{k_S^*}(\boldsymbol{z}_{S\perp})] \neq 0$ and we will learn some new directions $S_{\mathrm{new}}$. Therefore, we get a new learned subspace $S' = S \bigcup S_{\mathrm{new}}$. Repeating this process, we will get an increasing sequence of subspace $\emptyset = S_0 \subsetneq S_1 \subsetneq \cdots \subsetneq S_L = \mathrm{span}(\boldsymbol{U})$, and the overall generative leap exponent is then formally defined as $k^* := \max_i k_{S_i}^*$.

The generative exponent is important since Damian et al. (2025) proves that the complexity of efficiently learning a multi-index model is $\Theta(d^{k^*/2})$ via providing both the LDP lower bound and a matching spectral algorithm upper bound. Further, it can be shown that almost all common multi-index models have the generative leap exponent $k^* \leq 2$. For instance, it works when the link function is 1) all the polynomials, 2) piecewise linear functions, 3) intersection of halfspace. Therefore, we can always extract information from computing $\mathbb{E}\big[\mathcal{T}(y, \boldsymbol{z}_{S_i})\mathrm{He}_2(\boldsymbol{z}_{S_i^\perp})\big]$ with a proper pre-processing function $\mathcal{T}$. In addition, Lemma F.14 of Damian et al. (2024) indicates that $\mathcal{T}$ need not to be highly specific. In fact, they prove that even random polynomials suffice in the single-index setting.

We emphasize that this observation motivates our Assumption 5. In addition to generative leap exponent two, we are essentially requiring that there exists one pre-processing function $\mathcal{T}$ which is able to extract all the spanned directions in the subspace $\boldsymbol{U}$ from the second order information. Our assumption is weaker than the previous literature of using neural networks to learn multi-index models up to vanishing test error. For instance, via using a Huber loss, we can deal with the link function $g(\boldsymbol{z}) = \sum_{i=1}^k \mathrm{h}_4(z_i)$ which is not covered in previous papers like Damian et al. (2022).

## 6 CONCLUSIONS

In this paper, we demonstrate that under Gaussian input, a two-layer neural network trained with layer-wise gradient descent can learn generic multi-index models up to $o_d(1)$ test error using $\widetilde{\mathcal{O}}(d)$ sample and $\widetilde{\mathcal{O}}(d^2)$ time complexity, which matches the information-theoretic threshold up to the leading order. We emphasize that our analysis does not rely on any particular choice of activation, loss, or link function; rather, the assumptions imposed are generic and hold for a broad class of functions. An important and somewhat unexpected insight from the proof is that the inner layer must be trained for more than a constant number of steps to successfully learn the hidden features, in order to recover the hidden features most efficiently and thus achieve optimal performance. This observation is impossible to be observed via a DMFT or state evolution framework over bounded time horizon, and has the potential to extend to other settings. Overall, our results demonstrate that neural networks are capable of effective representation learning and can fully learn the target function on this canonical task with near-optimal efficiency.

## ACKNOWLEDGEMENT

The authors thank anonymous reviewers for their valuable feedback and suggestions. Zihao Wang also thanks Alex Damian, Andrea Montanari, Eshaan Nichani, Fan Nie, Yunwei Ren, Rui Sun, Zixuan Wang, Denny Wu, Jiatong Yu for their insightful discussions and valuable feedback during the completion of this work. Jason D. Lee acknowledges support of NSF CCF 2002272, NSF IIS 2107304, NSF CIF 2212262, ONR Young Investigator Award, and NSF CAREER Award 2144994.

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

# Appendix

## A   RELATED WORKS

**Learning multi-index models.**   In a multi-index model, the output depends on a low-dimensional subspace of the input data. This framework has a rich history in statistics and ML, where it is used to model low-dimensional structures with high-dimensional data. A comprehensive survey of current algorithms for learning multi-index models is provided in Bruna & Hsu (2025). In particular, it is known that under generic conditions, the information-theoretic sample complexity required to learn such targets is proportional to the input dimension, as rigorously established in Barbier et al. (2019); Aubin et al. (2019). Nevertheless, it remains largely unclear whether and how widely used polynomial-time algorithms can efficiently recover the hidden subspace.

To start with, Barbier et al. (2019); Mondelli & Montanari (2018); Lu & Li (2019); Luo et al. (2019); Maillard et al. (2021) studies generic single-index models and sharply derives the learning threshold. In particular, Mondelli & Montanari (2018); Lu & Li (2019); Luo et al. (2019); Maillard et al. (2021) show that for a broad class of single-index models, spectral methods with suitable data pre-processing can recover the hidden direction at the optimal threshold, matching the performance

of an optimal approximate message passing (AMP) Barbier et al. (2019). In contrast, Arous et al. (2021) considers a more restrictive setting and examines the learning of single-index models when using online SGD as the optimizer without data preprocessing, and introduces the concept of an information exponent, which characterizes the learning complexity in their setup.

A substantial line of works studies how to extend these results to multi-index models such as Chen & Meka (2020); Abbe et al. (2022; 2023a); Troiani et al. (2025); Damian et al. (2025); Kovačević et al. (2025); Defilippis et al. (2025); Diakonikolas et al. (2025). On one hand, Abbe et al. (2024; 2023b) introduces the notion of leap complexity, showing that hierarchical/staircase structure in multi-index models can facilitate learning. On the other hand, Damian et al. (2024; 2025) extend this line of work by formalizing the generative exponent for single-index models and the generative leap exponent for multi-index models, extending the previous results to higher order and incorporating the benefits of data pre-processing. In particular, we highlight Damian et al. (2025) for its two key contributions.

- For multi-index models with generative exponent $k^*$, the authors establish up to leading order a lower bound $\widetilde{\Omega}(d^{k^*/2})$ under the low-degree polynomial (LDP) framework and a matching upper bound $\widetilde{\mathcal{O}}(d^{k^*/2})$ achieved by a spectral polynomial-time algorithm for the sample complexity.
- Furthermore, it shows that the generative exponent is at most two for almost all common multi-index models, indicating that $\mathcal{O}(d)$ samples are sufficient to efficiently recover the hidden subspace.

These papers are in general not tight in terms of leading order constants. We also note that Diakonikolas et al. (2025) introduces a related notion, i.e., the $m$-well-behaved multi-index model, which coincides with the generative exponent in many cases. While their sample complexity exhibits a worse dependence on $d$ compared to Damian et al. (2025), their work offers two advantages: it targets agnostic PAC learning rather than subspace recovery, and it provides uniform sample complexity guarantees that apply agnostically to all multi-index models within a certain given class.

Subsequently, for the precise (weak recovery) learning threshold for multi-index models, Troiani et al. (2025) non-rigorously derives the sharp optimal sample size required for weakly learning generative-leap-exponent-two multi-index models by performing a local analysis of Bayes AMP, extending the results of Barbier et al. (2019); Mondelli & Montanari (2018) for single-index models when the generative exponent of the target equals two. Moreover, as two follow-up works, Kovačević et al. (2025); Defilippis et al. (2025) study learning multi-index models using an optimal spectral method, although their sample thresholds in Kovačević et al. (2025) are in general not always optimal, and none of those works can deal with the generative-staircase setup. We emphasize that the above works depend on the special structure of multi-index models and therefore do not use general-purpose algorithms such as learning with standard neural networks trained via gradient descent.

**Feature learning in neural networks.** The term feature learning refers to the following phenomenon. In practice, neural networks trained with local gradient-based algorithms such as gradient descent can adapt to the low-dimensional structure of the tasks and efficiently learn useful hidden features, which in turn facilitates successful learning and enables effective transfer learning to downstream tasks. Among the first several works in this direction, as we have mentioned previously, Damian et al. (2022) demonstrates that neural networks can learn generic multi-index models at a sub-optimal sample size of $n = \widetilde{\Theta}(d^2)$ under some additional assumptions. Similarly, Abbe et al. (2024; 2023b) establish a sub-optimal rate of $n = \widetilde{\mathcal{O}}(d^L)$ for neural network learning, where $L$ denotes the leap complexity. Ba et al. (2022) studies the problem under a much more restrictive assumption of an information exponent one, and in this setting shows that neural networks can succeed with $\mathcal{O}(d)$ samples.

Furthermore, as noted earlier, Lee et al. (2024) shows that neural networks can learn generic single-index models with generative exponent at most two with sample complexity $\widetilde{\mathcal{O}}(d)$ through data re-using while Montanari & Urbani (2025) provides a sharp DMFT theory analyzing the learning dynamics of single-index models using a standard two-layer neural network when the sample size and input dimension are proportional. Together, these two works essentially conclude the studies of learning single-index models with neural networks. Next, Dandi et al. (2024) demonstrates that

neural networks can learn multi-index models with generative leap exponent one using $n = \mathcal{O}(d)$ samples in constant time, leveraging DMFT theory. Finally, as noted previously, Arnaboldi et al. (2025) provides a loose result, showing that neural networks with $\mathcal{O}(d \log^2 d)$ samples can recover the hidden subspace of multi-index models but in a weak recovery sense when the generative leap exponent of the target equals two. Besides linear features, we note that there is also a substantial line of work on learning nonlinear features with deep neural networks, including Nichani et al. (2023); Ren et al. (2023); Wang et al. (2023); Fu et al. (2024).

We are aware of and highlight this recent concurrent work Montanari & Wang (2026). Our works differ in the following aspects. Montanari & Wang (2026) studies learning a multi-index model using a two-layer network trained with standard gradient descent instead of modified gradient-based algorithms. Their arguments are performed in the proportional regime, that is, $n/d \to \delta \in (0, +\infty)$, leveraging toolboxes from discrete time DMFT and random matrix theory. Besides, their results are sharp in constants, and they derive the sharp phase transition threshold at which the trained neural network can start to weakly recover the first hard direction (with generative exponent two). In contrast, Montanari & Wang (2026) does not establish full subspace recovery and therefore cannot provide an end-to-end analysis of the problem. Instead our work provides an end-to-end learning guarantee via a somehow different mechanism, albeit at the cost of a slightly worse sample complexity and the use of a modified layer-wise training scheme.

## B    NUMERICAL ILLUSTRATIONS

We conduct preliminary numerical experiments to validate our theoretical predictions.

We consider the following target function $f^*(\boldsymbol{x}) = (x_1^2 + \frac{1}{2}x_2^2)/\sqrt{5/2}$ where $\boldsymbol{x} \in \mathbb{R}^d$ and $x_i$ denotes the $i$-th coordinate. This is a multi-index model with $r = 2$ hidden directions. We use a two-layer neural network with width $m = 4$ and quadratic activation function $\sigma(t) = t^2$. Both layers use standard Kaiming uniform initialization for weights and biases. For each dimension $d$ and sample size exponent $\alpha$, we generate $n = \lfloor d^\alpha \rfloor$ training samples from $\mathsf{N}(\boldsymbol{0}, \boldsymbol{I}_d)$. We train the network using Adam optimizer with learning rate $\eta = 0.005$ for up to 1000 epochs with batch size 32. We vary the dimension $d$ logarithmically and search the sample size exponent $\alpha$ in the range $[1.1, 1.8]$ with step size 0.01. For each $(d, \alpha)$ pair, we run 5 independent trials with different random seeds and report the average minimal sample size exponent $\alpha$ required to achieve test error below threshold $\epsilon$.

### B.1    RESULTS

Figure 1 shows the minimal-sample-size exponent $\alpha$ required to achieve test error below threshold $\epsilon$ converges to one as $d$ increases for different thresholds $\epsilon \in \{1.0, 0.1, 0.01\}$. Each point represents the average over 5 random seeds. These experiments validate that neural networks can indeed learn multi-index models with near-optimal sample complexity. The decreasing trend in $\alpha$ as dimension grows provides empirical support for our theoretical finding that $n = \widetilde{\mathcal{O}}(d)$ samples suffice for successful learning, achieving the information-theoretic limit up to leading order.

### B.2    PROPER CHOICE OF LOSS FUNCTION

We demonstrate that the choice of loss function critically impacts learning performance for a subset of targets. We consider a challenging target $\mathrm{h}_4(x_1) + \mathrm{h}_4(x_2)$ where $\mathrm{h}_4(z) = \frac{1}{\sqrt{24}}(z^4 - 6z^2 + 3)$ is the degree-4 Hermite polynomial. This target has generative exponent 2 but information exponent 4 which is larger than generative exponent and requires a properly chosen loss function rather than square loss to ensure the second-order information is non-degenerate. To this end, we train a two-layer neural network with width $m = 4$ and cosine activation $\sigma(t) = \cos(t)$ using two different loss functions: Mean Squared Error (MSE) and Huber loss. We measure the learning process by the alignment metric $\cos(\text{best})$, defined as the maximum cosine similarity between any first-layer weight vector and the true directions $\boldsymbol{e}_1$ or $\boldsymbol{e}_2$. A value close to one indicates successful recovery of the hidden directions. For dimensions $d \in \{200, 300, 500\}$, we vary the sample size $n$ and plot the alignment $\cos(\text{best})$ as a function of sample complexity ratio $n/d$. For each $(d, n)$ pair, we run 10 independent trials with different random seeds and report the median along with 30th and 70th percentiles.

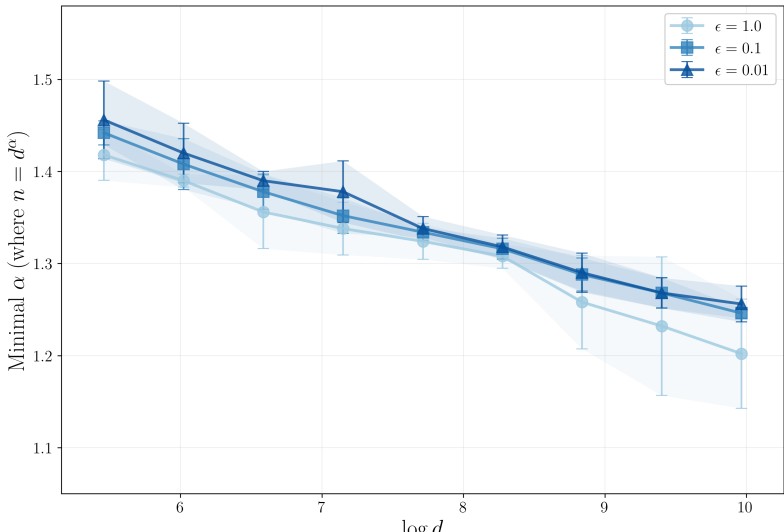

Figure 1: Minimal sample size exponent $\alpha$ (where $n = d^\alpha$) required to achieve test error $\leq \epsilon$ as a function of dimension $d$. Three thresholds are shown: $\epsilon = 1.0$ (light blue), $\epsilon = 0.1$ (medium blue), and $\epsilon = 0.01$ (dark blue). More stringent error thresholds require larger $\alpha$ (more samples), but all curves exhibit similar downward trends.

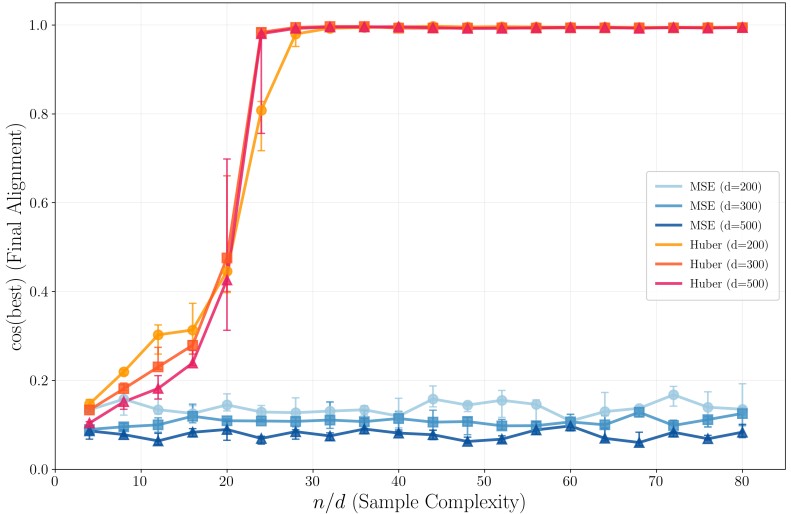

Figure 2: MSE and Huber loss for learning $h_4(x_1) + h_4(x_2)$ across different dimensions $d \in \{200, 300, 500\}$. The $x$-axis shows $n/d$, and the $y$-axis shows the cos similarity $\cos(\text{best})$. Lines show median with error bars indicating 30th to 70th percentiles over 10 random seeds. Blue shades represent MSE and red/orange shades represent Huber loss.

Figure 2 reveals a striking difference between the two loss functions. MSE loss consistently fails to learn the target function, with alignment remaining near zero even when $n/d = 80$. In stark contrast, Huber loss exhibits a clear phase transition: the alignment rapidly increases from near $0$ to near $1$ once the sample size exceeds a critical threshold around $n/d = 20$. This threshold is consistent across all three dimensions. This experiment validates our theoretical assumption (Assumption 5) that the pre-processing function $\ell'(0, y)$ must ensure non-degenerate second-order information. For this particular target, MSE loss corresponds to a degenerate pre-processing (since $\mathbb{E}[y\boldsymbol{x}\boldsymbol{x}^\mathsf{T}] = \boldsymbol{0}$), while Huber loss provides the necessary non-degeneracy. This demonstrates that: (i) the choice of loss function is not merely a matter of optimization convenience but fundamentally influences

whether the information-theoretic limit can be achieved, and (ii) our framework correctly predicts that with proper loss selection, neural networks can learn targets beyond previous theoretical guarantees with linear sample complexity.

## C PROOF OF LEMMA C.1: FEATURE LEARNING

We will use $\eta$ to refer to the learning rate $\eta_1$ and use $\beta$ to refer to $\beta_1$ throughout this section. We recall that we use $D$ to refer to a large absolute constant.

We will first introduce several more notations to help us to do the proof. In the first stage, we expect the prediction $f_{\Theta^{(t)}}(x_i)$ to remain small, since the norm of the features should remain small. We apply a first-order Taylor expansion of the first argument of $\ell_i^{\prime(t)}$ around zero

$$\ell_i^{\prime(t)} = \ell'(0, y_i) + \ell''(\zeta_i^{(t)}, y_i) f_{\Theta^{(t)}}(x_i) \quad \text{for some } \left|\zeta_i^{(t)}\right| \leq |f_{\Theta^{(t)}}(x_i)| \tag{26}$$

Denote the remainder term by

$$\Delta_i^{(t)} := \ell''(\zeta_i^{(t)}, y_i) f_{\Theta^{(t)}}(x_i) \tag{27}$$

which satisfies the bound $\left|\Delta_i^{(t)}\right| \leq \mathrm{L} \left|f_{\Theta^{(t)}}(x_i)\right|$ and should be small in principle. Substituting the expansion into the gradient expression for the $j$-th neuron, we get

$$\nabla_{w_j} \widehat{\mathcal{L}}_n(\Theta^{(t)}) = \frac{1}{n} \sum_{i=1}^n (\ell'(0, y_i) + \Delta_i^{(t)}) a_j \sigma'(w_j^{(t)\mathsf{T}} x_i) x_i. \tag{28}$$

Grouping terms, we have

$$\nabla_{w_j} \widehat{\mathcal{L}}_n(\Theta^{(t)}) = a_j \left[ \underbrace{\frac{1}{n} \sum_{i=1}^n \ell'(0, y_i) \sigma'(w_j^{(t)\mathsf{T}} x_i) x_i}_{\text{Zeroth-order term}} + \underbrace{\frac{1}{n} \sum_{i=1}^n \Delta_i^{(t)} \sigma'(w_j^{(t)\mathsf{T}} x_i) x_i}_{\text{Perturbation}} \right]. \tag{29}$$

To analyze the gradient dynamics, we further expand the derivative of the activation function using a second-order Taylor approximation around zero. Since $\sigma'(0) = 0$, we obtain

$$\sigma'(z) = \sigma''(0) z + \frac{1}{2} \sigma^{(3)}(\xi) z^2, \quad \text{for some } |\xi| \leq |z|. \tag{30}$$

Substituting the expansion of $\sigma'(\cdot)$ into $\nabla_{w_j} \widehat{\mathcal{L}}$, we obtain

$$\nabla_{w_j} \widehat{\mathcal{L}} = a_j \sigma''(0) \widehat{\Sigma}_\ell w_j + \underbrace{\frac{a_j}{2n} \sum_{i=1}^n \ell_i \sigma^{(3)}(\xi_i)(w_j^\mathsf{T} x_i)^2 x_i + \frac{a_j}{n} \sum_{i=1}^n \Delta_i^{(t)} \sigma'(w_j^\mathsf{T} x_i) x_i}_{\text{Perturbation}}$$

$$= a_j \sigma''(0) \widehat{\Sigma}_\ell w_j + \frac{a_j}{2n} \left( \sum_{i=1}^n \ell_i \sigma^{(3)}(\xi_i)(w_j^\mathsf{T} x_i)^2 x_i \right. \tag{31}$$

$$\left. + 2\sigma''(0) \sum_{i=1}^n \Delta_i^{(t)}(w_j^\mathsf{T} x_i) x_i + \sum_{i=1}^n \Delta_i^{(t)} \sigma^{(3)}(\xi_i)(w_j^\mathsf{T} x_i)^2 x_i \right).$$

The first term captures the leading-order linear dynamics that should be dominating, while the second term is a higher-order perturbation term that will be controlled using norm bounds on $w_j^{(t)}$ and also bounds on $\Delta_i^{(t)}$.

Under gradient descent with a proper weight decay (recall that we set $\beta = \frac{1}{\eta}$) and the learning rate as $\eta$, the update rule for $w_j$ becomes

$$w_j^{(t+1)} = w_j^{(t)} - \eta(\nabla_{w_j} \widehat{\mathcal{L}} + \beta w_j^{(t)}) = -\eta a_j \sigma''(0) \widehat{\Sigma}_\ell w_j^{(t)} + R_j^{(t)} \tag{32}$$

where the residual term $R_j^{(t)}$ is defined by

$$R_j^{(t)} := -\frac{\eta a_j}{2n}\left(\sum_{i=1}^{n}\ell_i\sigma^{(3)}(\xi_i^{(t)})(\boldsymbol{w}_j^{(t)\mathsf{T}}\boldsymbol{x}_i)^2\boldsymbol{x}_i\right.$$
$$\left.+2\sum_{i=1}^{n}\Delta_i^{(t)}(\boldsymbol{w}_j^{(t)\mathsf{T}}\boldsymbol{x}_i)\boldsymbol{x}_i+\sum_{i=1}^{n}\Delta_i^{(t)}\sigma^{(3)}(\xi_i^{(t)})(\boldsymbol{w}_j^{(t)\mathsf{T}}\boldsymbol{x}_i)^2\boldsymbol{x}_i\right)\qquad(33)$$

Since $(\boldsymbol{w}_j^{(t)\mathsf{T}}\boldsymbol{x}_i)^2\boldsymbol{x}_i = (\boldsymbol{w}_j^{(t)\mathsf{T}}\boldsymbol{x}_i)\boldsymbol{x}_i\boldsymbol{x}_i^{\mathsf{T}}\boldsymbol{w}_j^{(t)}$, we can write $R_j^{(t)}$ as $R_j^{(t)} = -\eta a_j\boldsymbol{Q}^{(t)}\boldsymbol{w}_j^{(t)}$ where $\boldsymbol{Q}^{(t)} \in \mathbb{R}^{d\times d}$ is a symmetric matrix defined by

$$\boldsymbol{Q}^{(t)} := \frac{1}{2n}\left(\sum_{i=1}^{n}\ell_i\sigma^{(3)}(\xi_i^{(t)})(\boldsymbol{w}_j^{(t)\mathsf{T}}\boldsymbol{x}_i)\boldsymbol{x}_i\boldsymbol{x}_i^{\mathsf{T}}\right.$$
$$\left.+2\sum_{i=1}^{n}\Delta_i^{(t)}\boldsymbol{x}_i\boldsymbol{x}_i^{\mathsf{T}}+\sum_{i=1}^{n}\Delta_i^{(t)}\sigma^{(3)}(\xi_i^{(t)})(\boldsymbol{w}_j^{(t)\mathsf{T}}\boldsymbol{x}_i)\boldsymbol{x}_i\boldsymbol{x}_i^{\mathsf{T}}\right)\qquad(34)$$

Now the entire update can be expressed as

$$\boldsymbol{w}_j^{(t+1)} = -\eta a_j(\widehat{\boldsymbol{\Sigma}}_\ell + \boldsymbol{Q}^{(t)})\boldsymbol{w}_j^{(t)}.\qquad(35)$$

Define
$$\boldsymbol{M}^{(t)} := \widehat{\boldsymbol{\Sigma}}_\ell + \boldsymbol{Q}^{(t)},\quad\text{so that}\quad \boldsymbol{w}_j^{(t+1)} = -\eta a_j\boldsymbol{M}^{(t)}\boldsymbol{w}_j^{(t)}.\qquad(36)$$

Through all the calculations above, we obtain a compact form of the update rule with a clear linear structure. Then by repeatedly following the updates for $\mathrm{T}_1$ iterations, we achieve

$$\boldsymbol{w}_j^{(\mathrm{T}_1)} = \frac{1}{\epsilon_0}(-a_j)^{\mathrm{T}_1}\eta^{\mathrm{T}_1}\left(\prod_{t=0}^{\mathrm{T}_1-1}\boldsymbol{M}^{(t)}\right)\boldsymbol{w}_j^{(0)}.\qquad(37)$$

This shows that $\boldsymbol{w}_j^{(t)}$ lies in the image of a composition of linear operators, each consisting of a dominant direction determined by $\widehat{\boldsymbol{\Sigma}}_\ell$, plus a time-dependent perturbation $\boldsymbol{Q}^{(t)}$.

The general picture is that $\boldsymbol{Q}^{(t)}$ is negligible when $t = o(\log d)$. In this case, we have $\boldsymbol{w}_j^{(t+1)} \propto \widehat{\boldsymbol{\Sigma}}_\ell\boldsymbol{w}_j^{(t)}$ and the dynamics of GD will be similar to a power method iteration. For now note that our power method iteration is with respect to $\widehat{\boldsymbol{\Sigma}}_\ell$, not $\boldsymbol{\Sigma}_\ell$, since the sample size is finite.

This section is devoted to a complete proof to bound the difference between $\boldsymbol{w}_j^{(\mathrm{T}_1)}$ and the ideal update

$$\widehat{\boldsymbol{w}_j}^{(\mathrm{T}_1)} := \frac{1}{\epsilon_0}(-a_j)^{\mathrm{T}_1}\eta^{\mathrm{T}_1}\boldsymbol{\Sigma}_\ell^{\mathrm{T}_1}\boldsymbol{w}_j^{(0)}\qquad(38)$$

Note that we have

$$\left\|\boldsymbol{w}_j^{(\mathrm{T}_1)} - \widehat{\boldsymbol{w}_j}^{(\mathrm{T}_1)}\right\| = \frac{1}{\epsilon_0}\eta^{\mathrm{T}_1}\left\|\left(\prod_{t=0}^{\mathrm{T}_1-1}\boldsymbol{M}^{(t)} - \boldsymbol{\Sigma}_\ell^{\mathrm{T}_1}\right)\boldsymbol{w}_j^{(0)}\right\|\qquad(39)$$

and therefore we just need to bound $\left\|\left(\prod_{t=0}^{\mathrm{T}_1-1}\boldsymbol{M}^{(t)} - \boldsymbol{\Sigma}_\ell^{\mathrm{T}_1}\right)\boldsymbol{w}_j^{(0)}\right\|$.

**Lemma C.1** (Total Perturbation Error). *With probability at least $1 - \mathcal{O}(d^{-D})$, assuming $n \gtrsim d\mathrm{T}_1^2$ and $m \leq d^{D/4}$, for all $j \in [m]$, the total perturbation error is bounded by*

$$\left\|\left(\prod_{t=0}^{\mathrm{T}_1-1}\boldsymbol{M}^{(t)} - \boldsymbol{\Sigma}_\ell^{\mathrm{T}_1}\right)\boldsymbol{w}_j^{(0)}\right\| \lesssim \epsilon_0\mathrm{T}_1\gamma\sqrt{\frac{\log d}{d}} + \epsilon_0\gamma^{\mathrm{T}_1},\quad\text{where}\quad \gamma = C\sqrt{\frac{d}{n}}\qquad(40)$$

*where $C$ is a universal constant.*

The proof of Lemma C.1 is provided in Appendix C.1.

## C.1   PROOF OF LEMMA C.1

We first give a sharp characterization of $\widehat{\boldsymbol{\Sigma}}_\ell$. We will bound the difference of the empirical and the population. Define

$$\boldsymbol{H}_\ell := \frac{1}{n} \sum_{i=1}^n \ell_i \boldsymbol{x}_i \boldsymbol{x}_i^\top - \mathbb{E}[\ell \boldsymbol{x} \boldsymbol{x}^\top]. \tag{41}$$

**Lemma C.2** (Operator Norm Bound for $\boldsymbol{H}_\ell$). *With probability* $1 - \mathcal{O}(d^{-2D})$ *we have* $\|\boldsymbol{H}_\ell\|_{\mathrm{op}} \lesssim \sqrt{\frac{d}{n}}$.

From this lemma, we expect a sharp concentration once $n \gg d$. This is standard in literature. Next, we bound the magnitude of $f_{\boldsymbol{\Theta}^{(t)}}(\boldsymbol{x})$.

**Lemma C.3.** *Let* $f_{\boldsymbol{\Theta}^{(t)}}(\boldsymbol{x}) = \sum_{j=1}^m a_j \sigma(\boldsymbol{w}_j^{(t)\top} \boldsymbol{x})$ *be the output of a two-layer neural network at iteration* $t$, *where each* $\boldsymbol{w}_j^{(t)} \in \mathbb{R}^d$ *satisfies* $\|\boldsymbol{w}_j^{(t)}\| \le \epsilon$ *and the input* $\boldsymbol{x} \in \mathbb{R}^d$ *satisfies* $\|\boldsymbol{x}\| \le 2\sqrt{d}$.

*Then, for all such* $\boldsymbol{x}$, *the network output satisfies the bound*

$$|f_{\boldsymbol{\Theta}^{(t)}}(\boldsymbol{x})| \le m \left( 2\epsilon^2 d + \frac{4}{3} M \epsilon^3 d^{3/2} \right). \tag{42}$$

Next, we start to control the operator norm of the matrix $\boldsymbol{Q}^{(t)}$.

**Lemma C.4** (Operator Norm Bound for $\boldsymbol{Q}^{(t)}$). *Suppose the following holds. Each input satisfies* $\|\boldsymbol{x}_i\| \le 2\sqrt{d}$. *Each weight vector satisfies* $\|\boldsymbol{w}_j^{(t)}\| \le \epsilon$. *The network prediction at iteration* $t$ *satisfies* $|f_{\boldsymbol{\Theta}^{(t)}}(\boldsymbol{x}_i)| \le C_f$.

*Then the operator norm of* $\boldsymbol{Q}^{(t)}$ *satisfies*

$$\|\boldsymbol{Q}^{(t)}\|_{\mathrm{op}} \le 4\epsilon d^{3/2}(\mathrm{L}M + \mathrm{L}C_f M) + 4\mathrm{L}C_f d \tag{43}$$

Plugging in the concrete estimation for the constant $C_f$, also note that without loss of generality we can assume $\mathrm{L} \ge 1$ and $M \ge 1$. we have the following.

**Corollary C.5.** *Suppose the following holds. Each input satisfies* $\|\boldsymbol{x}_i\| \le 2\sqrt{d}$. *Each weight vector satisfies* $\|\boldsymbol{w}_j^{(t)}\| \le \epsilon$.

*Then* $\boldsymbol{Q}^{(t)}$ *satisfies*

$$\|\boldsymbol{Q}^{(t)}\|_{\mathrm{op}} \le 4\mathrm{L}M\epsilon d^{3/2} + 8\mathrm{L}m\epsilon^2 d^2 + \frac{40}{3}\mathrm{L}Mm\epsilon^3 d^{5/2} + \frac{16}{3}\mathrm{L}M^2 m\epsilon^4 d^3 \le 32\mathrm{L}^2 M^2 m d^3 \epsilon. \tag{44}$$

With all the lemmas and corollaries above, define the following high probability events. First, we define the event $\mathcal{E}_{\boldsymbol{H}}$ (finite-sample noise bound) as follows:

$$\mathcal{E}_{\boldsymbol{H}} := \left\{ \|\boldsymbol{H}_\ell\|_{\mathrm{op}} \lesssim \gamma \text{ where } \gamma = C\sqrt{\frac{d}{n}} \right\}. \tag{45}$$

By Lemma C.2, this holds with probability at least $1 - \mathcal{O}(d^{-2D})$. Furthermore, provided $n \gtrsim d\mathrm{T}_1^2$, we know that $\gamma \le \frac{1}{4}$ also $\gamma \le \frac{1}{2\mathrm{T}_1}$. Second, we define the event $\mathcal{E}_{\boldsymbol{D}}$ (truncated norm) as follows:

$$\mathcal{E}_{\boldsymbol{D}} := \{ \|\boldsymbol{x}_i\| \le 2\sqrt{d} \quad \text{for } i \in [n] \}. \tag{46}$$

By Lemma E.18, this holds with probability at least $1 - n\exp(-cd)$, thus at least $1 - \mathcal{O}(d^{-2D})$. Third, we define the event $\mathcal{E}_{\boldsymbol{Q}}^{(t)}$ as $\mathcal{E}_{\boldsymbol{Q}}^{(t)} := \left\{ \left\|\boldsymbol{Q}^{(t)}\right\|_{\mathrm{op}} \le \frac{1}{4} \right\}$. Note that under event $\mathcal{E}_{\boldsymbol{D}}$, provided $\|\boldsymbol{w}_j^{(t)}\| \le \frac{1}{128m\mathrm{L}^2 M^2 d^3}$, we have $\mathcal{E}_{\boldsymbol{Q}}^{(t)}$ happens. We notice that $\mathcal{E}_{\boldsymbol{Q}}^{(t)}$ has no independent randomness, it is defined just for notation convenience.

In the following we analyze the gradient update conditioned on the event $\mathcal{E}_{\boldsymbol{H}} \cap \mathcal{E}_{\boldsymbol{D}}$, which holds with high probability. First bound the norm of $\boldsymbol{w}_j^{(t)}$.

**Lemma C.6** (Norm Bound after $T_1$ Steps). *Suppose the event $\mathcal{E}_{\boldsymbol{H}} \cap \mathcal{E}_{\boldsymbol{D}}$ holds, and let the learning rate be set as*

$$\eta = \frac{1}{C_\eta} \left(\frac{d}{r\iota^2}\right)^{1/(2T_1)} \tag{47}$$

*for a sufficiently large but fixed constant $C_\eta$ where $\iota = 4D\log d$. Suppose*

$$\|\boldsymbol{w}_j^{(0)}\| = \epsilon_0 \leq \frac{1}{128mL^2M^2d^{7/2}}. \tag{48}$$

*Then for all $0 \leq t \leq T_1 - 1$, we have*

$$\|\boldsymbol{w}_j^{(t)}\| \leq \sqrt{d}\epsilon_0 \quad \text{and} \quad \|\boldsymbol{w}_j^{(t)}\| \leq \frac{1}{128mL^2M^2d^3} \tag{49}$$

Next, we remove $\boldsymbol{Q}^{(t)}$ out of $\boldsymbol{M}^{(t)} = \widehat{\boldsymbol{\Sigma}}_\ell + \boldsymbol{Q}^{(t)}$ and compute the perturbation error.

**Lemma C.7.** *Suppose the event $\mathcal{E}_{\boldsymbol{H}} \cap \mathcal{E}_{\boldsymbol{D}}$ holds. Define $\Delta := \left(\prod_{t=0}^{T_1-1} \boldsymbol{M}^{(t)} - \widehat{\boldsymbol{\Sigma}}_\ell^{T_1}\right) \boldsymbol{w}_j^{(0)}$. Assume that $\epsilon_0 \leq \frac{1}{128mL^2M^2d^{7/2}}$. Then the perturbation error is bounded by*

$$\|\Delta\| \lesssim T_1 \left(\frac{5}{4}\right)^{T_1-1} md^{7/2}\epsilon_0^2. \tag{50}$$

Next, we deal with the difference between $\boldsymbol{\Sigma}_\ell$ and $\widehat{\boldsymbol{\Sigma}}_\ell$.

**Lemma C.8.** *Let $\boldsymbol{w} \sim \mathrm{Unif}(\mathbb{S}^{d-1}(\epsilon_0))$ be a random independent vector. Then with probability at least $1 - \mathcal{O}(d^{-2D})$, we have*

$$\|(\boldsymbol{\Sigma}_\ell + \boldsymbol{H}_\ell)^{T_1}\boldsymbol{w} - \boldsymbol{\Sigma}_\ell^{T_1}\boldsymbol{w}\| \lesssim \epsilon_0 \left((\|\boldsymbol{\Sigma}_\ell\|_{\mathrm{op}} + \|\boldsymbol{H}_\ell\|_{\mathrm{op}})^{T_1} - \|\boldsymbol{\Sigma}_\ell\|_{\mathrm{op}}^{T_1}\right) \sqrt{\frac{\log d}{d}}$$
$$+ \epsilon_0\|\boldsymbol{H}_\ell\|_{\mathrm{op}}^{T_1}. \tag{51}$$

**Corollary C.9.** *Under the setup of Lemma C.8 and note we assumed $\|\boldsymbol{H}_\ell\|_{\mathrm{op}} \leq \gamma$ with $\gamma \leq 1/(2T_1)$. Then with probability at least $1 - \mathcal{O}(d^{-2D})$, we have*

$$\left\|(\boldsymbol{\Sigma}_\ell + \boldsymbol{H}_\ell)^{T_1}\boldsymbol{w} - \boldsymbol{\Sigma}_\ell^{T_1}\boldsymbol{w}\right\| \lesssim \epsilon_0 \left(T_1\gamma\sqrt{\frac{\log d}{d}} + \gamma^{T_1}\right). \tag{52}$$

*Proof.* From the binomial expansion,

$$(1+\gamma)^{T_1} - 1 = \sum_{j=1}^{T_1} \binom{T_1}{j}\gamma^j = T_1\gamma + \sum_{j=2}^{T_1} \binom{T_1}{j}\gamma^j. \tag{53}$$

Since $\sum_{j=2}^{\infty}(T_1\gamma)^j \leq (T_1\gamma)^2/(1 - T_1\gamma) \leq 2(T_1\gamma)^2 \leq T_1\gamma$, we get $(1+\gamma)^{T_1} - 1 \leq 2T_1\gamma$. On the other hand, as in Lemma C.8, one shows under the same event that

$$\left\|(\boldsymbol{\Sigma}_\ell + \boldsymbol{H}_\ell)^{T_1}\boldsymbol{w} - \boldsymbol{\Sigma}_\ell^{T_1}\boldsymbol{w}\right\| \lesssim \epsilon_0 \left((1+\gamma)^{T_1} - 1\right)\sqrt{\frac{\log d}{d}} + \gamma^{T_1}\right). \tag{54}$$

Combining the two bounds yields the claimed result. $\square$

By putting all the things together, we obtain the final bound of Lemma C.1.

*Proof of Lemma C.1.* We first prove another lemma. We aim to first show that under the stated condition on $\epsilon_0$, the perturbation error from the time-varying iterates is smaller than the leading-order perturbation induced by $\boldsymbol{H}_\ell$.

**Lemma C.10.** *Suppose the event $\mathcal{E}_{\boldsymbol{H}} \cap \mathcal{E}_{\boldsymbol{D}}$ holds. Recall $\Delta = \left( \prod_{t=0}^{\mathrm{T}_1 - 1} \boldsymbol{M}^{(t)} - \widehat{\boldsymbol{\Sigma}}_\ell^{\mathrm{T}_1} \right) \boldsymbol{w}_j^{(0)}$ denote the error from time-varying perturbations.*

*Then, provided that $\epsilon_0 \leq \frac{1}{m\mathrm{L}M^2 d^{7/2}} \sqrt{\frac{\log d}{n}} (4/5)^{\mathrm{T}_1}$ we have*

$$\|\Delta\| \lesssim \mathrm{T}_1 \gamma \sqrt{\frac{\log d}{d}} \cdot \epsilon_0 \tag{55}$$

*Proof.* Recall the bounds established in previous lemmas. We have the one-step expansion error from iterated time-varying perturbations

$$\|\Delta\| \lesssim \mathrm{T}_1 \left( \frac{5}{4} \right)^{\mathrm{T}_1 - 1} m d^{7/2} \epsilon_0^2 \tag{56}$$

We plug in one $\epsilon_0$

$$\begin{aligned}
\|\Delta\| &\lesssim \mathrm{T}_1 \left( \frac{5}{4} \right)^{\mathrm{T}_1 - 1} m d^{7/2} \epsilon_0^2 \\
&\lesssim \mathrm{T}_1 \left( \frac{5}{4} \right)^{\mathrm{T}_1 - 1} m d^{7/2} \epsilon_0 \frac{1}{m d^{7/2}} \left( \frac{4}{5} \right)^{\mathrm{T}_1} \sqrt{\frac{\log d}{n}} \\
&\lesssim \mathrm{T}_1 \gamma \sqrt{\frac{\log d}{d}} \epsilon_0 \tag{57}
\end{aligned}$$

and the proof is complete. $\qquad\square$

Finally we decompose the total perturbation error into two parts

$$\begin{aligned}
\left( \prod_{t=0}^{\mathrm{T}_1 - 1} \boldsymbol{M}^{(t)} - \boldsymbol{\Sigma}_\ell^{\mathrm{T}_1} \right) \boldsymbol{w}_j^{(0)} &= \left( \prod_{t=0}^{\mathrm{T}_1 - 1} \boldsymbol{M}^{(t)} - \widehat{\boldsymbol{\Sigma}}_\ell^{\mathrm{T}_1} \right) \boldsymbol{w}_j^{(0)} + \left( \widehat{\boldsymbol{\Sigma}}_\ell^{\mathrm{T}_1} - \boldsymbol{\Sigma}_\ell^{\mathrm{T}_1} \right) \boldsymbol{w}_j^{(0)} \\
&\lesssim \mathrm{T}_1 \gamma \sqrt{\frac{\log d}{d}} \epsilon_0 + \epsilon_0 \left( \mathrm{T}_1 \gamma \sqrt{\frac{\log d}{d}} + \gamma^{\mathrm{T}_1} \right) \\
&\lesssim \epsilon_0 \left( \mathrm{T}_1 \gamma \sqrt{\frac{\log d}{d}} + \gamma^{\mathrm{T}_1} \right) \tag{58}
\end{aligned}$$

The conclusion is then straightforward if we adapt the previous two lemmas to bound these two parts separately. $\qquad\square$

## C.2 Proof of supporting lemmas in Appendix C.1

### C.2.1 Proof of Lemma C.2

*Proof.* We analyze the operator norm via a net argument.

Recall that $\|\boldsymbol{H}_\ell\|_{\mathrm{op}} = \sup_{\|\boldsymbol{u}\| = \|\boldsymbol{v}\| = 1} \boldsymbol{u}^\mathsf{T} \boldsymbol{H}_\ell \boldsymbol{v}$. Let $M^{1/4}, N^{1/4}$ be two $\frac{1}{4}$-nets of the unit sphere in $\mathbb{R}^d$. Then

$$\|\boldsymbol{H}_\ell\|_{\mathrm{op}} \leq 2 \max_{\boldsymbol{u} \in M^{1/4}, \boldsymbol{v} \in N^{1/4}} \boldsymbol{u}^\mathsf{T} \boldsymbol{H}_\ell \boldsymbol{v} \tag{59}$$

Fix any $\boldsymbol{u}, \boldsymbol{v}$, and define

$$Z_i := (\boldsymbol{u}^\mathsf{T} \boldsymbol{x}_i)(\boldsymbol{v}^\mathsf{T} \boldsymbol{x}_i)\ell_i - \mathbb{E}[(\boldsymbol{u}^\mathsf{T} \boldsymbol{x})(\boldsymbol{v}^\mathsf{T} \boldsymbol{x})\ell]. \tag{60}$$

Note that we have $\boldsymbol{u}^\mathsf{T} \boldsymbol{H}_\ell \boldsymbol{v} = \frac{1}{n} \sum_{i=1}^n Z_i$. Since $\boldsymbol{x}_i$ is standard Gaussian and $\ell_i \in [-\mathrm{L}, \mathrm{L}]$, we have $\|Z_i\|_{\psi_1} \lesssim \mathrm{L}$. Therefore we can invoke Bernstein's inequality and do a union bound over the net to obtain

$$\mathbb{P}(\|\boldsymbol{H}_\ell\|_{\mathrm{op}} \geq t) \leq 81^d \exp\left( -\frac{cnt^2}{\mathrm{L}^2} \right). \tag{61}$$

Choosing $t = C\mathrm{L}\sqrt{\frac{d}{n}}$ for sufficiently large $C$, we obtain the desired conclusion. $\qquad\square$

### C.2.2 PROOF OF LEMMA C.3

*Proof.* Using the Taylor expansion of $\sigma(\cdot)$ at 0 with $\sigma'(0) = 0$ and $\sigma''(0) = 1$, we have

$$\sigma(z) = \sigma(0) + \frac{1}{2}z^2 + \frac{1}{6}\sigma^{(3)}(\xi)z^3 \quad \text{for some } |\xi| \leq |z| \tag{62}$$

Then

$$f_{\Theta^{(t)}}(\boldsymbol{x}) = \sum_{j=1}^{m} a_j \sigma(\boldsymbol{w}_j^{(t)\mathsf{T}}\boldsymbol{x}) = \sum_{j=1}^{m} a_j \left[ \sigma(0) + \frac{1}{2}(\boldsymbol{w}_j^{(t)\mathsf{T}}\boldsymbol{x})^2 + \frac{1}{6}\sigma^{(3)}(\xi_j^{(t)})(\boldsymbol{w}_j^{(t)\mathsf{T}}\boldsymbol{x})^3 \right] \tag{63}$$

Since $\sum_j a_j = 0$, the constant term cancels and

$$f_{\Theta^{(t)}}(\boldsymbol{x}) = \frac{1}{2}\sum_{j=1}^{m} a_j (\boldsymbol{w}_j^{(t)\mathsf{T}}\boldsymbol{x})^2 + \frac{1}{6}\sum_{j=1}^{m} a_j \sigma^{(3)}(\xi_j^{(t)})(\boldsymbol{w}_j^{(t)\mathsf{T}}\boldsymbol{x})^3 \tag{64}$$

We then bound each term. We have

$$|\boldsymbol{w}_j^{(t)\mathsf{T}}\boldsymbol{x}| \leq \|\boldsymbol{w}_j^{(t)}\|\|\boldsymbol{x}\| \leq 2\epsilon\sqrt{d}. \tag{65}$$

Therefore $(\boldsymbol{w}_j^{(t)\mathsf{T}}\boldsymbol{x})^2 \leq 4\epsilon^2 d$ and $(\boldsymbol{w}_j^{(t)\mathsf{T}}\boldsymbol{x})^3 \leq 8\epsilon^3 d^{3/2}$. Then

$$\left| \sum_{j=1}^{m} a_j (\boldsymbol{w}_j^{(t)\mathsf{T}}\boldsymbol{x})^2 \right| \leq 4m\epsilon^2 d, \quad \left| \sum_{j=1}^{m} a_j \sigma^{(3)}(\xi_j^{(t)})(\boldsymbol{w}_j^{(t)\mathsf{T}}\boldsymbol{x})^3 \right| \leq 8mM\epsilon^3 d^{3/2} \tag{66}$$

and the following is straightforward

$$|f_{\Theta^{(t)}}(\boldsymbol{x})| \leq \frac{1}{2}(4m\epsilon^2 d) + \frac{1}{6}(8mM\epsilon^3 d^{3/2}) = 2m\epsilon^2 d + \frac{4}{3}mM\epsilon^3 d^{3/2}. \tag{67}$$

The proof is complete. $\qquad\square$

### C.2.3 PROOF OF LEMMA C.4

*Proof.* We upper bound the operator norm of $\boldsymbol{Q}^{(t)}$ by treating the three additive components separately.

Let

$$\boldsymbol{Q}_I^{(t)} := \frac{1}{2n}\sum_{i=1}^{n} \ell_i \sigma^{(3)}(\xi_i^{(t)})(\boldsymbol{w}_j^\mathsf{T}\boldsymbol{x}_i)\boldsymbol{x}_i\boldsymbol{x}_i^\mathsf{T} \tag{68}$$

We bound each component $|\boldsymbol{w}_j^\mathsf{T}\boldsymbol{x}_i| \leq \|\boldsymbol{w}_j\|\|\boldsymbol{x}_i\| \leq 2\epsilon\sqrt{d}$, $\|\boldsymbol{x}_i\boldsymbol{x}_i^\mathsf{T}\|_{\mathrm{op}} = \|\boldsymbol{x}_i\|^2 \leq 4d$, $|\ell_i\sigma^{(3)}(\xi_i^{(t)})| \leq \mathrm{L}M$. Thus,

$$\|\boldsymbol{Q}_I^{(t)}\|_{\mathrm{op}} \leq \frac{1}{2n}n\mathrm{L}M(2\epsilon\sqrt{d})(4d) = 4\mathrm{L}M\epsilon d^{3/2}. \tag{69}$$

Let

$$\boldsymbol{Q}_{II}^{(t)} := \frac{1}{n}\sum_{i=1}^{n} \Delta_i^{(t)}\boldsymbol{x}_i\boldsymbol{x}_i^\mathsf{T}. \tag{70}$$

Each term satisfies $|\Delta_i^{(t)}| \leq \mathrm{L}C_f$, $\|\boldsymbol{x}_i\boldsymbol{x}_i^\mathsf{T}\|_{\mathrm{op}} \leq 4d$, so

$$\|\boldsymbol{Q}_{II}^{(t)}\|_{\mathrm{op}} \leq \mathrm{L}C_f(4d). \tag{71}$$

Let

$$\boldsymbol{Q}_{III}^{(t)} := \frac{1}{2n}\sum_{i=1}^{n} \Delta_i^{(t)}\sigma^{(3)}(\xi_i^{(t)})(\boldsymbol{w}_j^\mathsf{T}\boldsymbol{x}_i)\boldsymbol{x}_i\boldsymbol{x}_i^\mathsf{T}. \tag{72}$$

We again use $|\Delta_i^{(t)}\sigma^{(3)}(\xi_i^{(t)})| \leq \mathrm{L}C_f M$, $|\boldsymbol{w}_j^\mathsf{T}\boldsymbol{x}_i| \leq 2\epsilon\sqrt{d}$, $\|\boldsymbol{x}_i\boldsymbol{x}_i^\mathsf{T}\|_{\mathrm{op}} \leq 4d$, to obtain

$$\|\boldsymbol{Q}_{III}^{(t)}\|_{\mathrm{op}} \leq \frac{1}{2n}n\mathrm{L}C_f M(2\epsilon\sqrt{d})(4d) = 4\mathrm{L}C_f M\epsilon d^{3/2}. \tag{73}$$

We combine the operator norm bounds

$$\|\boldsymbol{Q}^{(t)}\|_{\mathrm{op}} \leq \|\boldsymbol{Q}_I^{(t)}\|_{\mathrm{op}} + \|\boldsymbol{Q}_{II}^{(t)}\|_{\mathrm{op}} + \|\boldsymbol{Q}_{III}^{(t)}\|_{\mathrm{op}} \leq 4\epsilon d^{3/2}(\mathrm{L}M + \mathrm{L}C_f M) + 4\mathrm{L}C_f d. \tag{74}$$

The proof is complete. $\qquad\square$

### C.2.4 PROOF OF LEMMA C.6

*Proof.* We first prove the following supporting lemma.

**Lemma C.11** (One-Step Norm Control). *Suppose the event $\mathcal{E}_{\boldsymbol{H}} \cap \mathcal{E}_{\boldsymbol{D}} \cap \mathcal{E}_{\boldsymbol{Q}}^{(t)}$ holds, and let the learning rate be set as*

$$\eta = \frac{1}{C_\eta} \left( \frac{d}{r\iota^2} \right)^{1/(2\mathrm{T}_1)} \tag{75}$$

*for a sufficiently large constant $C_\eta$ where $\iota = 4D \log d$. Then the gradient update satisfies*

$$\|\boldsymbol{w}_j^{(t+1)}\| \le d^{1/(2\mathrm{T}_1)} \|\boldsymbol{w}_j^{(t)}\| \tag{76}$$

*Proof.* Under the event $\mathcal{E}_{\boldsymbol{H}} \cap \mathcal{E}_{\boldsymbol{Q}}^{(t)}$, we have

$$\|\boldsymbol{M}^{(t)}\|_{\mathrm{op}} \le \|\boldsymbol{\Sigma}_\ell\|_{\mathrm{op}} + \|\boldsymbol{H}_\ell\|_{\mathrm{op}} + \left\|\boldsymbol{Q}^{(t)}\right\|_{\mathrm{op}} \le 1 + \frac{1}{4} + \frac{1}{4} = \frac{3}{2} \tag{77}$$

Then,

$$\|\boldsymbol{w}_j^{(t+1)}\| = \eta \|\boldsymbol{M}^{(t)} \boldsymbol{w}_j^{(t)}\| \le \eta \|\boldsymbol{M}^{(t)}\|_{\mathrm{op}} \cdot \|\boldsymbol{w}_j^{(t)}\| \le \frac{3}{2} \eta \cdot \|\boldsymbol{w}_j^{(t)}\|. \tag{78}$$

Plug the formula for $\eta$ into the above one and we obtain the desired conclusion.

$\square$

We prove by induction on $t \in \{0, 1, \dots, \mathrm{T}_1 - 1\}$ that $\|\boldsymbol{w}_j^{(t)}\| \le d^{t/(2\mathrm{T}_1)} \epsilon_0$ where we assume $\epsilon_0 \le \frac{1}{128 m \mathrm{L}^2 M^2 d^{7/2}}$. The $t = 0$ case is trivial. Suppose for some $t \le \mathrm{T}_1$, we have

$$\|\boldsymbol{w}_j^{(t)}\| \le d^{t/(2\mathrm{T}_1)} \epsilon_0. \tag{79}$$

Plugging $\epsilon_0$ in, $\|\boldsymbol{w}_j^{(t)}\| \le \frac{1}{128 m \mathrm{L}^2 M^2 d^3}$, thus the event $\mathcal{E}_{\boldsymbol{H}} \cap \mathcal{E}_{\boldsymbol{D}} \cap \mathcal{E}_{\boldsymbol{Q}}^{(t)}$ holds, apply the previous lemma C.11, we get

$$\|\boldsymbol{w}_j^{(t+1)}\| \le d^{1/(2\mathrm{T}_1)} \|\boldsymbol{w}_j^{(t)}\| \le d^{(t+1)/(2\mathrm{T}_1)} \epsilon_0. \tag{80}$$

Repeating this induction, the result is straightforward. In particular, for all $0 \le t \le \mathrm{T}_1 - 1$, we have $\|\boldsymbol{w}_j^{(t)}\| \le \sqrt{d} \epsilon_0$. $\square$

### C.2.5 PROOF OF LEMMA C.7

*Proof.* We begin by expanding the product of perturbed operators using a standard multiplicative expansion

$$\prod_{t=0}^{\mathrm{T}_1 - 1} \boldsymbol{M}^{(t)} = \widehat{\boldsymbol{\Sigma}}_\ell^{\mathrm{T}_1} + \sum_{s=1}^{\mathrm{T}_1} \sum_{0 \le \mathrm{T}_1 < \dots < \mathrm{T}_1} \widehat{\boldsymbol{\Sigma}}_\ell^{\mathrm{T}_1 - t_s - 1} \boldsymbol{Q}^{(t_s)} \widehat{\boldsymbol{\Sigma}}_\ell^{t_s - 1 - 1} \cdots \boldsymbol{Q}^{(\mathrm{T}_1)} \widehat{\boldsymbol{\Sigma}}_\ell^{\mathrm{T}_1}. \tag{81}$$

Subtracting $\widehat{\boldsymbol{\Sigma}}_\ell^{\mathrm{T}_1}$ from both sides yields

$$\prod_{t=0}^{\mathrm{T}_1 - 1} \boldsymbol{M}^{(t)} - \widehat{\boldsymbol{\Sigma}}_\ell^{\mathrm{T}_1} = \sum_{s=1}^{\mathrm{T}_1} \sum_{0 \le \mathrm{T}_1 < \dots < \mathrm{T}_1} \widehat{\boldsymbol{\Sigma}}_\ell^{\mathrm{T}_1 - t_s - 1} \boldsymbol{Q}^{(t_s)} \widehat{\boldsymbol{\Sigma}}_\ell^{t_s - 1} \cdots \boldsymbol{Q}^{(\mathrm{T}_1)} \widehat{\boldsymbol{\Sigma}}_\ell^{\mathrm{T}_1}. \tag{82}$$

We now apply this expansion to $\boldsymbol{w}_j^{(0)}$, and estimate the resulting norm. For each term in the expansion with $s$ perturbation matrices, we use the operator norm bound

$$\|\widehat{\boldsymbol{\Sigma}}_\ell^{\mathrm{T}_1 - t_s - 1} \boldsymbol{Q}^{(t_s)} \cdots \boldsymbol{Q}^{(\mathrm{T}_1)} \widehat{\boldsymbol{\Sigma}}_\ell^{\mathrm{T}_1} \boldsymbol{w}_j^{(0)}\| \le \|\widehat{\boldsymbol{\Sigma}}_\ell\|_{\mathrm{op}}^{\mathrm{T}_1 - s} \cdot \prod_{k=1}^{s} \|\boldsymbol{Q}^{(t_k)}\|_{\mathrm{op}} \cdot \|\boldsymbol{w}_j^{(0)}\| \tag{83}$$

Note that the event $\mathcal{E}_{\boldsymbol{H}} \cap \mathcal{E}_{\boldsymbol{D}}$ holds, $\|\widehat{\boldsymbol{\Sigma}}_\ell\|_{\mathrm{op}} \le \|\boldsymbol{\Sigma}_\ell\|_{\mathrm{op}} + \|\boldsymbol{H}_\ell\|_{\mathrm{op}} \le \frac{5}{4}$. Combining with

$$\epsilon_0 \le \frac{1}{128 m \mathrm{L}^2 M^2 d^{7/2}} \tag{84}$$

by Lemma C.6, for all $0 \leq t \leq \mathrm{T}_1 - 1 \, \|\boldsymbol{w}_j^{(t)}\| \leq \sqrt{d}\epsilon_0$. Moreover, using Corollary C.5, it is easy to know that

$$\left\|\boldsymbol{Q}^{(t)}\right\|_{\mathrm{op}} \leq 32 \mathrm{L} M^2 m d^{7/2} \epsilon_0 \tag{85}$$

Hence, each term with $s$ perturbation matrices is bounded by

$$\left(\frac{5}{4}\right)^{\mathrm{T}_1 - s} (32\mathrm{L}^2 M^2 m d^{7/2} \epsilon_0)^s \epsilon_0. \tag{86}$$

Summing over all such terms (total of $\binom{\mathrm{T}_1}{s}$ per order $s$), we have

$$\left\|\left(\prod_{t=0}^{\mathrm{T}_1 - 1} \boldsymbol{M}^{(t)} - \widehat{\boldsymbol{\Sigma}}_\ell^{\mathrm{T}_1}\right)\boldsymbol{w}_j^{(0)}\right\| \leq \sum_{s=1}^{\mathrm{T}_1} \binom{\mathrm{T}_1}{s} \left(\frac{5}{4}\right)^{\mathrm{T}_1 - s} (32\mathrm{L}^2 M^2 m d^{7/2} \epsilon_0)^s \epsilon_0. \tag{87}$$

This sum can be rewritten as

$$\epsilon_0 \left[\left(\frac{5}{4} + 32\mathrm{L}^2 M^2 m d^{7/2} \epsilon_0\right)^{\mathrm{T}_1} - \left(\frac{5}{4}\right)^{\mathrm{T}_1}\right]. \tag{88}$$

Applying the first-order binomial expansion which is valid for the small enough $\epsilon_0$ in our assumption and we conclude

$$\|\Delta\| \lesssim \mathrm{T}_1 \left(\frac{5}{4}\right)^{\mathrm{T}_1 - 1} m M^2 d^{7/2} \epsilon_0^2. \tag{89}$$

The proof is complete. $\qquad\square$

### C.2.6 PROOF OF LEMMA C.8

*Proof.* We first introduce several standard concentration results.

**Lemma C.12.** *Suppose $\boldsymbol{A}$ is a matrix in $\mathbb{R}^{d \times d}$, and $\boldsymbol{w} \sim \mathrm{Unif}(\mathbb{S}^{d-1})$. Then with probability at least $1 - \mathcal{O}(d^{-2D})$, we have*

$$|(\boldsymbol{A}\boldsymbol{w})_s| \lesssim \|\boldsymbol{A}\|_{\mathrm{op}}\sqrt{\frac{\log d}{d}} \quad \text{for all } s \in [r]. \tag{90}$$

*Proof.* Fix $s \in [r]$, and define the function $f(\boldsymbol{w}) = (\boldsymbol{A}\boldsymbol{w})_s = \boldsymbol{e}_s^{\mathsf{T}}\boldsymbol{A}\boldsymbol{w}$.

This function is Lipschitz with

$$\|f\|_{\mathrm{Lip}} = \|\boldsymbol{e}_s^{\mathsf{T}}\boldsymbol{A}\| \leq \|\boldsymbol{A}\|_{\mathrm{op}}. \tag{91}$$

By the standard concentration results for Lipschitz function on the sphere for instance Theorem 5.1.4 in Vershynin (2018), and noting that $\mathbb{E}[f(\boldsymbol{w})] = 0$, we have the following concentration

$$\mathbb{P}(|f(\boldsymbol{w})| \geq t) \leq 2\exp\left(-\frac{cdt^2}{\|\boldsymbol{A}\|^2}\right). \tag{92}$$

Set $t = 2\|\boldsymbol{A}\|_{\mathrm{op}}\sqrt{\frac{D\log d}{cd}}$. Then we obtain

$$\mathbb{P}\left(|(\boldsymbol{A}\boldsymbol{w})_s| \geq 2\|\boldsymbol{A}\|_{\mathrm{op}}\sqrt{\frac{D\log d}{cd}}\right) \leq d^{-2D}. \tag{93}$$

Applying a union bound over all the $r$ coordinates and the desired result follows. $\qquad\square$

**Lemma C.13.** *Let $\boldsymbol{w} \sim \mathrm{Unif}(\mathbb{S}^{d-1}(\epsilon_0))$. Let $\boldsymbol{H}_\ell \in \mathbb{R}^{d \times d}$ be a fixed matrix. Let $\boldsymbol{\Pi}^* \in \mathbb{R}^{r \times d}$ denote the projection onto the first $r$ coordinates, that is $(\boldsymbol{\Pi}^*\boldsymbol{v})_s = v_s$ for $s \in [r]$.*

*Then with probability at least $1 - \mathcal{O}(d^{-2D})$, we have*

$$\|\boldsymbol{\Pi}^*\boldsymbol{H}_\ell^k\boldsymbol{w}\| \lesssim \epsilon_0 \cdot \|\boldsymbol{H}_\ell^k\|_{\mathrm{op}} \cdot \sqrt{\frac{r\log d}{d}}, \quad \text{for all } k = 0, 1, \ldots, \mathrm{T}_1. \tag{94}$$

*Proof.* Fix $k \in \{0, 1, \ldots, T_1\}$, and define the matrix $\boldsymbol{A} = \boldsymbol{H}_\ell^k$. By Lemma C.12, for each fixed $s \in [r]$, with probability at least $1 - \mathcal{O}(d^{-2D})$, we have

$$|(\boldsymbol{A}\boldsymbol{w})_s| = |(\boldsymbol{H}_\ell^k \boldsymbol{w})_s| \le \epsilon_0 \|\boldsymbol{H}_\ell^k\|_{\text{op}} \sqrt{\frac{4D \log d}{cd}}, \tag{95}$$

for some small absolute constant $c$. Applying a union bound over all $s \in [r]$, we get the following bound holds

$$\forall s \in [r] \quad |(\boldsymbol{H}_\ell^k \boldsymbol{w})_s| \le \epsilon_0 \|\boldsymbol{H}_\ell^k\|_{\text{op}} \sqrt{\frac{4D \log d}{cd}} \tag{96}$$

Hence

$$\|\boldsymbol{\Pi}^* \boldsymbol{H}_\ell^k \boldsymbol{w}\|^2 = \sum_{s=1}^r ((\boldsymbol{H}_\ell^k \boldsymbol{w})_s)^2 \le r \left( \epsilon_0 \|\boldsymbol{H}_\ell^k\|_{\text{op}} \sqrt{\frac{4D \log d}{cd}} \right)^2 \tag{97}$$

and the conclusion follows with a square root calculation and a union bound over time. $\square$

To prove Lemma C.8, we invoke the following auxiliary results.

(i) **Operator norm bound.** $\|\boldsymbol{H}_\ell\|_{\text{op}} \lesssim \sqrt{\frac{d}{n}}$ with probability at least $1 - \mathcal{O}(d^{-2D})$.

(ii) **Projection bound.** For any integer $0 \le \ell \le T_1$, we have by Lemma C.13

$$\|\boldsymbol{\Pi}^* \boldsymbol{H}_\ell^\ell \boldsymbol{w}\| \lesssim \epsilon_0 \cdot \|\boldsymbol{H}_\ell\|_{\text{op}}^\ell \cdot \sqrt{\frac{r \log d}{d}} \quad \text{with probability at least } 1 - \mathcal{O}(d^{-2D}). \tag{98}$$

Under those two events, we expand

$$(\boldsymbol{\Sigma}_\ell + \boldsymbol{H}_\ell)^{T_1} \boldsymbol{w} - \boldsymbol{\Sigma}_\ell^{T_1} \boldsymbol{w} = \sum_{k=1}^{T_1} M_k(\boldsymbol{w}), \tag{99}$$

where $M_k(\boldsymbol{w})$ collects all degree-$T_1$ monomials with exactly $k$ occurrences of $\boldsymbol{H}_\ell$. Among them we have two kinds of terms.

First, the pure $\boldsymbol{H}_\ell$ term. There is exactly one such term:

$$\|\boldsymbol{H}_\ell^{T_1} \boldsymbol{w}\| \le \epsilon_0 \|\boldsymbol{H}_\ell\|_{\text{op}}^{T_1}. \tag{100}$$

Second, the mixed terms. Each $M_k(\boldsymbol{w})$ consists of expressions of the form

$$T(\boldsymbol{w}) = \boldsymbol{A}\boldsymbol{\Sigma}_\ell^m \boldsymbol{H}_\ell^\ell \boldsymbol{w}, \tag{101}$$

where $\boldsymbol{A}$ consists of a product of $T_1 - k - m$ $\boldsymbol{\Sigma}_\ell$'s and $k - \ell$ $\boldsymbol{H}_\ell$'s. Because $\boldsymbol{\Sigma}_\ell$ is rank-$r$, and leftmost $\boldsymbol{\Sigma}_\ell^m$ acts as projection to top-$r$ eigenspace, we may write

$$T(\boldsymbol{w}) = \boldsymbol{A}\boldsymbol{\Sigma}_\ell^m \boldsymbol{\Pi}^* (\boldsymbol{H}_\ell^\ell \boldsymbol{w}). \tag{102}$$

Now, we estimate each term.

$$\|\boldsymbol{A}\|_{\text{op}} \le \|\boldsymbol{\Sigma}_\ell\|_{\text{op}}^{T_1 - k - m} \|\boldsymbol{H}_\ell\|_{\text{op}}^{k - \ell}; \tag{103}$$

$$\|\boldsymbol{\Sigma}_\ell^m\|_{\text{op}} = \|\boldsymbol{\Sigma}_\ell\|_{\text{op}}^m; \tag{104}$$

$$\|\boldsymbol{\Pi}^* (\boldsymbol{H}_\ell^\ell \boldsymbol{w})\| \lesssim \epsilon_0 \|\boldsymbol{H}_\ell\|_{\text{op}}^\ell \sqrt{\frac{r \log d}{d}}; \tag{105}$$

and therefore

$$\|T(\boldsymbol{w})\| \lesssim \epsilon_0 \|\boldsymbol{\Sigma}_\ell\|_{\text{op}}^{T_1 - k} \|\boldsymbol{H}_\ell\|_{\text{op}}^k \sqrt{\frac{r \log d}{d}}. \tag{106}$$

The number of such terms is at most $\binom{T_1}{k}$, and summing over $1 \le k < T_1$

$$\epsilon_0 \sum_{k=1}^{T_1 - 1} \binom{T_1}{k} \|\boldsymbol{\Sigma}_\ell\|_{\text{op}}^{T_1 - k} \|\boldsymbol{H}_\ell\|_{\text{op}}^k \sqrt{\frac{r \log d}{d}}$$

$$\lesssim \epsilon_0 \left( (\|\boldsymbol{\Sigma}_\ell\|_{\text{op}} + \|\boldsymbol{H}_\ell\|_{\text{op}})^{T_1} - \|\boldsymbol{\Sigma}_\ell\|_{\text{op}}^{T_1} - \|\boldsymbol{H}_\ell\|_{\text{op}}^{T_1} \right) \sqrt{\frac{r \log d}{d}}. \tag{107}$$

Adding the pure $\boldsymbol{H}_\ell$ term back, we conclude the final result with $1 - \mathcal{O}(d^{-2D})$ probability

$$\|(\boldsymbol{\Sigma}_\ell + \boldsymbol{H}_\ell)^{\mathrm{T}_1}\boldsymbol{w} - \boldsymbol{\Sigma}_\ell^{\mathrm{T}_1}\boldsymbol{w}\| \lesssim \epsilon_0 \left( (\|\boldsymbol{\Sigma}_\ell\|_{\mathrm{op}} + \|\boldsymbol{H}_\ell\|_{\mathrm{op}})^{\mathrm{T}_1} - \|\boldsymbol{\Sigma}_\ell\|_{\mathrm{op}}^{\mathrm{T}_1} \right) \sqrt{\frac{r \log d}{d}} + \epsilon_0 \|\boldsymbol{H}_\ell\|_{\mathrm{op}}^{\mathrm{T}_1}. \tag{108}$$

The proof is complete. $\qquad\square$

## D  PROOF OF THEOREM 1

### D.1  PROOF ROADMAP OF THEOREM 1

**General roadmap.**   For notational convenience, given parameters $\boldsymbol{\Theta}$ we denote the population loss and its empirical counterpart on the second-stage training set as

$$\mathcal{L}(\boldsymbol{\Theta}) := \mathbb{E}_{\boldsymbol{x}}[\ell(f_{\boldsymbol{\Theta}}(\boldsymbol{x}), f^\star(\boldsymbol{x}))] \quad \text{and} \quad \widehat{\mathcal{L}}(\boldsymbol{\Theta}) := \frac{1}{n}\sum_{i=1}^{n} \ell\left(f_{\boldsymbol{\Theta}}(\boldsymbol{x}_i), f^\star(\boldsymbol{x}_i)\right) \tag{109}$$

To prove Theorem 2, i.e., upper bounding the population loss $\mathcal{L}(\boldsymbol{\Theta}^{(\mathrm{T})})$ where $\boldsymbol{\Theta}^{(\mathrm{T})}$ is returned by Algorithm 1, we first construct an "ideal" parameter candidate $\boldsymbol{\Theta}^\star$ based on our feature learning analysis (Appendix C.1) and analysis of standard random feature models as a transition term, which lies in a proper parameter class $\boldsymbol{\Theta}^\star \in \mathcal{F}$ with well-controlled $\|\boldsymbol{\Theta}^\star\|$ and yields small empirical error. Then, we can decompose the error $\mathcal{L}(\boldsymbol{\Theta}^{(\mathrm{T})})$ as

$$\mathcal{L}(\boldsymbol{\Theta}^{(\mathrm{T})}) = \underbrace{\widehat{\mathcal{L}}(\boldsymbol{\Theta}^\star)}_{\text{Lemma D.17 (subtle issue)}} + \underbrace{\widehat{\mathcal{L}}(\boldsymbol{\Theta}^{(\mathrm{T})}) - \widehat{\mathcal{L}}(\boldsymbol{\Theta}^\star)}_{\text{Lemma D.18}} + \underbrace{\mathcal{L}(\boldsymbol{\Theta}^{(\mathrm{T})}) - \widehat{\mathcal{L}}(\boldsymbol{\Theta}^{(\mathrm{T})})}_{\text{Lemma D.19}}. \tag{110}$$

Next, by choosing a proper weight decay hyperparameter which relates to control the complexity of $\mathcal{F}$ and leveraging the convexity of loss function with respect to $\boldsymbol{a}$, we show in Lemma D.18 that with $\widetilde{\mathcal{O}}(1)$ second-stage training steps, the second term can be bounded appropriately. Then, given that $\boldsymbol{\Theta}^{(\mathrm{T})} \in \mathcal{F}$, we use standard Rademacher complexity analysis in Lemma D.19 with proper truncation arguments to bound the generalization error (the third term) via uniform convergence arguments. Now it suffices to construct the ideal $\boldsymbol{\Theta}^\star$ with a controllable norm and a small error.

**Subtle issues in constructing $\boldsymbol{\Theta}^\star$.**   An intricate problem arises when constructing $\boldsymbol{\Theta}^\star$ and estimating $\widehat{\mathcal{L}}(\boldsymbol{\Theta}^\star)$. We note that each neuron $\boldsymbol{w}_j^{(\mathrm{T})}$ after training depends simultaneously on all neurons at initialization in a complicated manner. Consequently, the network after time T cannot be expressed as an average of conditionally independent terms, which hinders a direct Monte Carlo sampling analysis.

To overcome this, we introduce the so-called decoupled parameter group $\widetilde{\boldsymbol{\Theta}} = (\boldsymbol{a}, \boldsymbol{b}, \widetilde{\boldsymbol{W}})$. In this construction, each row $\widetilde{\boldsymbol{w}}_j$ preserves its dependence on the corresponding initialization $\boldsymbol{w}_j^{(0)}$ but is independent of the other neurons at initialization. Then, we can further decompose the error $\widehat{\mathcal{L}}(\boldsymbol{\Theta}^\star)$ as

$$\widehat{\mathcal{L}}(\boldsymbol{\Theta}^\star) = \underbrace{\widehat{\mathcal{L}}(\widetilde{\boldsymbol{\Theta}^*})}_{\text{Lemma }D.15} + \underbrace{\widehat{\mathcal{L}}(\boldsymbol{\Theta}^\star) - \widehat{\mathcal{L}}(\widetilde{\boldsymbol{\Theta}^*})}_{\text{Lemma D.16}}. \tag{111}$$

For the first term, we invoke standard Monte Carlo concentration arguments (Lemma D.14) to bound the squared loss, and then use the Lipschitz continuity of $\ell$ to extend the squared loss to the general loss $\widehat{\mathcal{L}}(\widetilde{\boldsymbol{\Theta}^*})$ (Lemma D.15). Moreover, we show that the empirical risks of $\boldsymbol{\Theta}^\star$ and $\widetilde{\boldsymbol{\Theta}^*}$ differ only by a negligible amount (Lemma D.16), which is well-controlled by the Lipschitz continuity of the loss function $\ell$ and the small distance between $\|\boldsymbol{W}^{(T)} - \widetilde{\boldsymbol{W}}^{(T)}\|$ (Appendix C.1), ensuring that $\boldsymbol{\Theta}^\star$ and $\widetilde{\boldsymbol{\Theta}^*}$ are sufficiently close.

Now by combining all the above analysis, we can give an upper bound for $\mathcal{L}(\boldsymbol{\Theta}^{(\mathrm{T})})$ (Proposition D.20). Last, we manage to get a balance among the error terms with undetermined coefficients related to the monomial approximation property of the target function in Lemma D.21 and compute the time complexity explicitly in Lemma D.26, concluding our formal proof of Theorem $1^2$.

---

$^2$The formal version is stated in Theorem 2, where we explicitly specify all the hyperparameters.

## D.2 EVENTS OF HIGH PROBABILITIES

We will first define the following events.

Briefly, we define $\mathcal{E}_\alpha$ and $\mathcal{E}_\beta$ as events that the features are ensured to be well learned in the first stage, which has been proved in Appendix C.1, we then define $\mathcal{E}_\eta$ and $\mathcal{E}_\zeta$ to control the range of the learned features and the second-stage training data for convenience of learning the outer polynomial link function $g$ in the second training stage.

First, we define the events solely related to our first dataset. Recall from Lemma C.2 that the concentration error $\boldsymbol{H}_\ell$ admits the high-probability operator-norm bound $\|\boldsymbol{H}_\ell\|_{\mathrm{op}} \leq \gamma := C\sqrt{d/n}$ where $C$ is an absolute constant. We note that for sufficiently but moderately large $n$ ($n \gtrsim d\mathrm{T}_1^2$) the perturbation bound satisfies $\gamma \leq \frac{1}{2\mathrm{T}_1}$. We will assume this regime throughout.

We define

$$\mathcal{E}_\alpha := \left\{\|\boldsymbol{H}_\ell\|_{\mathrm{op}} \leq \gamma\right\} \bigcap \left(\bigcap_{i \in [n]} \left\{\frac{1}{2}\|\boldsymbol{x}_i\|^2 \leq d\right\}\right) \tag{112}$$

and we know from Lemma C.2 and Lemma E.18 that $\mathbb{P}(\mathcal{E}_\alpha) \geq 1 - \mathcal{O}(d^{-2D})$. Next, for a fixed perturbation $\boldsymbol{H}_\ell$, the projection control event (for $\boldsymbol{w}$ drawn uniformly from the sphere $\epsilon_0 \mathbb{S}^{d-1}$) for a single feature is

$$\widetilde{\mathcal{E}_\beta} := \left\{\frac{\|\boldsymbol{\Pi}^*(\boldsymbol{H}_\ell^\ell \boldsymbol{w})\|}{\epsilon_0} \lesssim \|\boldsymbol{H}_\ell^\ell\|_{\mathrm{op}}\sqrt{\frac{\log d}{d}} \quad \text{for all } \ell \leq \mathrm{T}_1\right\} \tag{113}$$

and we know from Lemma C.13 that $\mathbb{P}(\widetilde{\mathcal{E}_\beta}|\boldsymbol{H}_\ell) \geq 1 - \mathcal{O}(d^{-2D})$. Next, we consider the events related to the sampling of our features at initialization. Note that because we use symmetric initialization, only the first $m/2$ weight vectors are independent. For a fixed perturbation $\boldsymbol{H}_\ell$ and $1 \leq j \leq m/2$, the projection control event $\mathcal{E}_\beta^{(j)}$ is

$$\mathcal{E}_\beta^{(j)} := \left\{\frac{\|\boldsymbol{\Pi}^*(\boldsymbol{H}_\ell^\ell)\boldsymbol{w}_j^{(0)}\|}{\epsilon_0} \lesssim \|\boldsymbol{H}_\ell^\ell\|_{\mathrm{op}}\sqrt{\frac{\log d}{d}} \quad \text{for all } \ell \leq \mathrm{T}_1\right\} \tag{114}$$

and we know from Lemma C.13 that $\mathbb{P}(\mathcal{E}_\beta^{(j)}|\boldsymbol{H}_\ell) \geq 1 - \mathcal{O}(d^{-2D})$. Next, define the desired event as the intersection $\mathcal{E}_\beta := \bigcap_{j=1}^{m/2} \mathcal{E}_\beta^{(j)}$. Applying a union bound gives

$$\mathbb{P}(\mathcal{E}_\beta|\boldsymbol{H}_\ell) \geq 1 - \mathcal{O}(md^{-2D}) \tag{115}$$

From here, in this section we will write $\eta_1$ as $\eta$ from time to time. Next, conditioning on the first dataset and the features, after the first stage of training, we define new events over the randomness of the second dataset. We need to introduce more notations first. From the previous section, we have the following formula for our final feature

$$\boldsymbol{w}_j^{(\mathrm{T}_1)} = \frac{1}{\epsilon_0}(-a_j)^{\mathrm{T}_1}\eta^{\mathrm{T}_1}\left(\prod_{t=0}^{\mathrm{T}_1-1} \boldsymbol{M}^{(t)}\right)\boldsymbol{w}_j^{(0)} \tag{116}$$

and recall the ideal feature $\widehat{\boldsymbol{w}_j}^{(\mathrm{T}_1)} := \frac{1}{\epsilon_0}(-a_j)^{\mathrm{T}_1}\eta^{\mathrm{T}_1}\boldsymbol{\Sigma}_\ell^{\mathrm{T}_1}\boldsymbol{w}_j^{(0)}$. For technical convenience, we next define the approximate feature to remove the influence of the interaction between neurons.

$$\widetilde{\boldsymbol{w}}_j^{(\mathrm{T}_1)} = \frac{1}{\epsilon_0}(-a_j)^{\mathrm{T}_1}\eta^{\mathrm{T}_1}\left(\widehat{\boldsymbol{\Sigma}}_\ell^{\mathrm{T}_1}\right)\boldsymbol{w}_j^{(0)} \tag{117}$$

To help the proof, we further define

$$\boldsymbol{h}_n(\boldsymbol{w}) := \frac{1}{\epsilon_0}\left(\widehat{\boldsymbol{\Sigma}}_\ell^{\mathrm{T}_1}\right)\boldsymbol{w} \quad \text{and} \quad \boldsymbol{h}(\boldsymbol{w}) := \frac{1}{\epsilon_0}\boldsymbol{\Sigma}_\ell^{\mathrm{T}_1}\boldsymbol{w} \tag{118}$$

and the residual $\boldsymbol{r}(\boldsymbol{w}) := \boldsymbol{h}_n(\boldsymbol{w}) - \boldsymbol{h}(\boldsymbol{w})$ for notational convenience. These approximate features are useful for our future analysis since $\widehat{\boldsymbol{\Sigma}}_\ell$ does not depend on $\{\boldsymbol{w}_j\}_{j=1}^m$. Therefore, the approximate

dynamics then will decouple across neurons. We can bound the difference between the approximate features and true features afterwards with some additional efforts.

We next define the following event over the randomness of the second dataset conditioning on the first dataset for one feature

$$\widetilde{\mathcal{E}}_\eta := \left\{ |\boldsymbol{h}_n(\boldsymbol{w})^\mathsf{T}\boldsymbol{x}_i| \leq \|\boldsymbol{h}_n(\boldsymbol{w})\|\sqrt{2\iota} \quad \text{for all } n+1 \leq i \leq 2n \right\} \tag{119}$$

where $\iota = 4D \log d$. Via invoking the Gaussian tail bound plus a union bound over dataset, we have $\mathbb{P}(\widetilde{\mathcal{E}}_\eta|\boldsymbol{H}_\ell, \boldsymbol{w}) \geq 1 - \mathcal{O}(nd^{-2D})$. Then we define

$$\mathcal{E}_\eta^{(j)} := \left\{ |\boldsymbol{h}_n(\boldsymbol{w}_j^{(0)})^\mathsf{T}\boldsymbol{x}_i| \leq \|\boldsymbol{h}_n(\boldsymbol{w}_j^{(0)})\|\sqrt{2\iota} \quad \text{for all } n+1 \leq i \leq 2n \right\} \tag{120}$$

where again $\iota = 4D \log d$ for each individual neuron. We have $\mathbb{P}(\mathcal{E}_\eta^{(j)}|\boldsymbol{H}_\ell, \{\boldsymbol{w}_j^{(0)}\}_{j=0}^m) \geq 1 - \mathcal{O}(nd^{-2D})$ due to the same reasons as above. Define the desired event as the intersection $\mathcal{E}_\eta := \bigcap_{j=1}^{m/2} \mathcal{E}_\eta^{(j)}$. Applying a union bound over these indices $j \in [m/2]$ gives $\mathbb{P}(\mathcal{E}_\eta|\boldsymbol{H}_\ell, \{\boldsymbol{w}_j^{(0)}\}_{j=0}^m) \geq 1 - \mathcal{O}(nmd^{-2D})$.

Last, we also need to define another event depending on $\{\boldsymbol{x}_i\}_{i=n+1}^{2n}$, that is, $\mathcal{E}_\zeta := \bigcap_{i\in[n]} \left\{ \frac{1}{2}\|\boldsymbol{x}_{i+n}\|^2 \leq d \right\} \bigcap_{i\in[n]} \left\{ |(\boldsymbol{x}_{i+n})_k| \lesssim \sqrt{\log d} \text{ for all } k \leq r \right\}$. By Lemma E.18 and Lemma E.19 we have that $\mathbb{P}(\mathcal{E}_\zeta) \geq 1 - \mathcal{O}(d^{-D})$.

Finally, we denote $\mathcal{E}_\mu := \mathcal{E}_\alpha \bigcap \mathcal{E}_\beta \bigcap \mathcal{E}_\eta \bigcap \mathcal{E}_\zeta$ from here. We will condition on this event throughout this section unless we particularly specify.

### D.3 PROOF OF THEOREM 1

We first recall some basic facts from the results of the first stage. We have $\|\boldsymbol{H}_\ell\|_{\mathrm{op}} \leq \gamma$ under $\mathcal{E}_\alpha$, which gives rise to

$$\|\widehat{\boldsymbol{\Sigma}}_\ell\|_{\mathrm{op}} = \|\boldsymbol{\Sigma}_\ell + \boldsymbol{H}_\ell\|_{\mathrm{op}} \leq \|\boldsymbol{\Sigma}_\ell\|_{\mathrm{op}} + \|\boldsymbol{H}_\ell\|_{\mathrm{op}} \leq 1 + \gamma \tag{121}$$

and taking the product over $t$ yields $\left\|\widehat{\boldsymbol{\Sigma}}_\ell^{\mathrm{T}_1}\right\|_{\mathrm{op}} \leq (1+\gamma)^{\mathrm{T}_1}$.

On those events we can obtain the crude uniform bound

$$\|\boldsymbol{h}_n(\boldsymbol{w})\| = \frac{\left\|\widehat{\boldsymbol{\Sigma}}_\ell^{\mathrm{T}_1}\boldsymbol{w}\right\|}{\epsilon_0} \leq \frac{(1+\gamma)^{\mathrm{T}_1}}{\epsilon_0}\|\boldsymbol{w}\| = (1+\gamma)^{\mathrm{T}_1} \tag{122}$$

that we might use later. Note that this estimation is uniform for all the features. We denote this upper bound by $V := (1+\gamma)^{\mathrm{T}_1}$.

Furthermore, we know by Corollary C.9, for $\boldsymbol{w}$ drawn uniformly from the sphere $\epsilon_0 \mathbb{S}^{d-1}$, under $\mathcal{E}_\alpha \cap \widetilde{\mathcal{E}}_\beta$, the residual term is dominated by

$$\|\boldsymbol{r}(\boldsymbol{w})\| \leq C\left( \gamma\mathrm{T}_1\sqrt{\frac{rD\log d}{d}} + \gamma^{\mathrm{T}_1} \right) \tag{123}$$

where $C$ is a universal constant. For notational convenience, we denote the right side as $K$. We note that this is also uniformly correct for all features under the full event $\mathcal{E}$.

Next, we need to do the following moment estimation for the residual term. The below is not straightforward since we need to take the expectation of $\boldsymbol{w}$.

**Lemma D.1.** *Under the high probability event $\mathcal{E}_\alpha$, for any integer $1 \leq j \leq 4p$ we have*

$$\left(\mathbb{E}_{\boldsymbol{w}}\|\boldsymbol{\Pi}^*\boldsymbol{r}(\boldsymbol{w})\|^j\right)^{1/j} \lesssim K. \tag{124}$$

The next lemma computes the sample size needed to control $K$.

**Lemma D.2.** *For each absolute constant c, in order to bound K as $d\kappa^{2T_1}K^2 \leq c^2$ it is just sufficient to require*

$$n \gtrsim T_1^2 d \log d\kappa^{2T_1} + d^{1+1/T_1}\kappa^2. \tag{125}$$

**Lemma D.3.** *Given $n \gtrsim T_1^2 d \log d\kappa^{2T_1} + d^{1+1/T_1}\kappa^2$, under the high probability event of $\mathcal{E}_\alpha \cap \widetilde{\mathcal{E}_\beta}$, we have*

$$\|\boldsymbol{h}_n(\boldsymbol{w})\| \lesssim \sqrt{\log d}\frac{1}{\sqrt{d}}. \tag{126}$$

The proofs of the three lemmas above are provided in Appendices D.5.1, D.5.2 and D.5.3, respectively. Then the following corollary holds.

**Corollary D.4.** *Let $\boldsymbol{w} \sim \epsilon_0 \mathrm{Unif}(\mathbb{S}^{d-1})$. Given $n \gtrsim T_1^2 d \log d\kappa^{2T_1} + d^{1+1/T_1}\kappa^2$, under the high probability event of $\mathcal{E}_\alpha \cap \widetilde{\mathcal{E}_\beta} \cap \widetilde{\mathcal{E}_\eta}$ we have*

$$\max_{j\in[n+1,2n]} \|\boldsymbol{h}_n(\boldsymbol{w})^\mathsf{T}\boldsymbol{x}_j\| \lesssim \frac{\log d}{\sqrt{d}}. \tag{127}$$

*Proof of Corollary D.4.* From previous proof we can see under the high probability event of $\mathcal{E}_\alpha \cap \widetilde{\mathcal{E}_\beta} \cap \widetilde{\mathcal{E}_\eta}$ we have

$$\max_{j\in[n+1,2n]} |\boldsymbol{h}_n(\boldsymbol{w})^\mathsf{T}\boldsymbol{x}_j| \leq \|\boldsymbol{h}_n(\boldsymbol{w})\|\sqrt{4D\log d}. \tag{128}$$

Also by Lemma D.3, as $\mathcal{E}_\alpha \cap \widetilde{\mathcal{E}_\beta}$ happens, we have $\|\boldsymbol{h}_n(\boldsymbol{w})\| \lesssim \sqrt{\log d}\frac{1}{\sqrt{d}}$. Plug this in and we get our conclusion. $\square$

Before proving the results below, we introduce the notation used throughout. First, let $S^\star = \mathrm{span}\{\boldsymbol{e}_1, \ldots, \boldsymbol{e}_r\} \subset \mathbb{R}^d$ be the subspace spanned by the first $r$ standard basis vectors, and recall $\boldsymbol{\Pi}^\star : \mathbb{R}^d \to S^\star$ denote the orthogonal projector onto this subspace (i.e. $\boldsymbol{\Pi}^\star$ preserves the first $r$ coordinates and removes other coordinates). Then, for a symmetric tensor $\boldsymbol{T} \in \mathrm{Sym}^{2k}(\mathbb{R}^d)$, write $\mathrm{Mat}(\boldsymbol{T})$ for its canonical matricization as a linear operator on $\mathrm{Sym}^k(\mathbb{R}^d)$; equivalently, $\mathrm{Mat}(\boldsymbol{T})$ is the unique linear operator satisfying

$$\left\langle \boldsymbol{u}^{\otimes k}, \mathrm{Mat}(\boldsymbol{T})\boldsymbol{v}^{\otimes k} \right\rangle = \boldsymbol{T}(\boldsymbol{u}^{\otimes k}, \boldsymbol{v}^{\otimes k}) \quad \text{for all } \boldsymbol{u}, \boldsymbol{v} \in \mathbb{R}^d. \tag{129}$$

The next lemma will be important for our proof. We can now show that the obtained features $\boldsymbol{h}_n(\boldsymbol{w})$ are sufficiently expressive to allow us to efficiently represent any polynomial of degree $p$ restricted to the principal subspace $S^\star$.

**Lemma D.5.** *Under the high probability event of $\mathcal{E}_\alpha$, given the sample size $n \gtrsim T_1^2 d \log d\kappa^{2T_1} + d^{1+1/T_1}\kappa^2$, for any $k \leq p$, we have*

$$\mathrm{Mat}\left(\mathbb{E}_{\boldsymbol{w}}[(\boldsymbol{\Pi}^\star\boldsymbol{h}_n(\boldsymbol{w}))^{\otimes 2k}]\right) \gtrsim \left(\frac{d}{\lambda_r^{2T_1}}\right)^{-k} \boldsymbol{\Pi}_{Sym^k(S^\star)} \tag{130}$$

*where $\boldsymbol{\Pi}_{Sym^k(S^\star)}$ denotes the projection onto symmetric k tensors restricted to $S^\star$.*

In this lemma we will use monomial basis to construct the following approximations for $\langle \boldsymbol{T}, \boldsymbol{x}^{\otimes k} \rangle$ for any $k \leq p$ and any $k$ tensor $\boldsymbol{T}$ supported on $S^* = \{\boldsymbol{e}_1, \ldots, \boldsymbol{e}_r\}$.

**Lemma D.6.** *Assume the sample size $n \gtrsim T_1^2 d \log d\kappa^{2T_1} + d^{1+1/T_1}\kappa^2$. Then under the high probability event $\mathcal{E}_\alpha$, for each $k \leq p$ and any symmetric k tensor $\boldsymbol{T}$ supported on $S^\star$, there exists $z_{\boldsymbol{T}}(\boldsymbol{w})$ such that*

$$\mathbb{E}_{\boldsymbol{w}}\left[z_{\boldsymbol{T}}(\boldsymbol{w})(\boldsymbol{h}_n(\boldsymbol{w})^\mathsf{T}\boldsymbol{x})^k\right] = \left\langle \boldsymbol{T}, \boldsymbol{x}^{\otimes k} \right\rangle \tag{131}$$

*and we have the bounds*

$$\mathbb{E}_{\boldsymbol{w}}\left[z_{\boldsymbol{T}}(\boldsymbol{w})^2\right] \lesssim \left(\frac{d}{\lambda_r^{2T_1}}\right)^k \|\boldsymbol{T}\|_F^2 \quad \text{and} \quad |z_{\boldsymbol{T}}(\boldsymbol{w})| \lesssim \left(\frac{d}{\lambda_r^{2T_1}}\right)^k \|\boldsymbol{T}\|_F \|\boldsymbol{h}_n(\boldsymbol{w})\|^k. \tag{132}$$

The proofs are provided in Appendices D.5.4 and D.5.5.

In the following, we show that bias reinitialization together with our learned features can be used to transform a wide class of activations $\sigma$ into (or uniformly approximate by) the monomial $x^p$, thereby making it able to approximate low-dimensional polynomials.

**Definition D.7.** *Let $a$ be a Rademacher RV i.e. $\mathbb{P}(a = +1) = \mathbb{P}(a = -1) = \frac{1}{2}$ and let $b \sim Unif[-3, 3]$, independent from $a$.*

*We say the activation function $\sigma(\cdot)$ has the monomial approximation property with exponent $\beta \geq 0$ if for every integer $k \geq 0$ and every positive $\rho$ there exists a bounded weight function $v_k^{(\rho)} : \{\pm 1\} \times [-3, 3] \to \mathbb{R}$ with*

$$\|v_k^{(\rho)}\|_\infty \leq V_{k,\rho} \leq C_\sigma(k)\rho^{-\beta} \tag{133}$$

*such that*

$$\sup_{|z| \leq 1} \left| \mathbb{E}_{a,b}\left[v_k^{(\rho)}(a, b)\sigma(az + b)\right] - z^k \right| \leq \rho \tag{134}$$

The uniform approximation property in Definition D.7 is expected to hold for many standard activations, with different growth behavior of the required weights $V_{k,\rho}$. For example, we provably have $\beta = 0$ for the activation function below.

**Definition D.8.** *The locally quadratic activation is defined as*

$$\sigma(t) = \begin{cases} 2|t| - 1, & |t| \geq 1, \\ t^2, & |t| < 1. \end{cases} \tag{135}$$

In Secttion E.3 we show that $\beta = 0$ for locally quadratic function.

We first prove the following lemma in preparation for the proof that follows.

**Lemma D.9.** *Under the sample-size condition $n \gtrsim \mathrm{T}_1^2 d \log d\kappa^{2\mathrm{T}_1} + d^{1+1/\mathrm{T}_1}\kappa^2$. Then under the high probability event $\mathcal{E}_\mu$ we have for all features $j \in [m]$*

$$\left\|\widetilde{\boldsymbol{w}}_j^{(\mathrm{T}_1)}\right\| \leq 1 \quad and \quad \max_{i \in [n+1, 2n]} |(\widetilde{\boldsymbol{w}}_j^{(\mathrm{T}_1)})^\mathsf{T}\boldsymbol{x}_i| \leq 1 \quad and \quad \left\|\boldsymbol{w}_j^{(\mathrm{T}_1)}\right\| \lesssim 1 \tag{136}$$

Note that we already constructed the approximations for $\langle \boldsymbol{T}, \boldsymbol{x}^{\otimes k} \rangle$ using monomial basis, now we will utilize those bounds above to construct the following approximations for $\langle \boldsymbol{T}, \boldsymbol{x}^{\otimes k} \rangle$ for any $k \leq p$ and any $k$ tensor $\boldsymbol{T}$ supported on $S^* = \{\boldsymbol{e}_1, \ldots, \boldsymbol{e}_r\}$ using activation function $\sigma$.

**Lemma D.10.** *Under the sample-size condition $n \gtrsim \mathrm{T}_1^2 d \log d\kappa^{2\mathrm{T}_1} + d^{1+1/\mathrm{T}_1}\kappa^2$ also $\mathrm{T}_1 = o(\log d)$. Then under the high probability event $\mathcal{E}_\alpha \cap \mathcal{E}_\zeta$, Consider*

$$f_{h_{\boldsymbol{T}}}(\boldsymbol{x}) := \mathbb{E}_{a,b,\boldsymbol{w}}\left[h_{\boldsymbol{T}}(a, b, \boldsymbol{w})\sigma(\widetilde{\boldsymbol{w}}^{(\mathrm{T}_1)\mathsf{T}}\boldsymbol{x} + b)\right] \tag{137}$$

*where $\boldsymbol{w}^{(\mathrm{T}_1)} = (-\eta a)^{\mathrm{T}_1}\boldsymbol{h}_n(\boldsymbol{w})$. Here*

$$h_{\boldsymbol{T}}(a, b, \boldsymbol{w}) := \frac{v_k^{(\epsilon_k)}(a, b)z_{\boldsymbol{T}}(\boldsymbol{w})}{(-\eta)^{k\mathrm{T}_1}} \mathbf{1}_{\eta^{\mathrm{T}_1}\|\boldsymbol{h}_n(\boldsymbol{w})\| \leq 1} \prod_{i=1}^{n} \mathbf{1}_{|\eta^{\mathrm{T}_1}\boldsymbol{h}_n(\boldsymbol{w})^\mathsf{T}\boldsymbol{x}_{i+n}| \leq 1} \tag{138}$$

*where $v_k^{(\epsilon_k)}$ is constructed from Definition D.7 and $z_{\boldsymbol{T}}(\boldsymbol{w})$ is the same from Corollary D.6. We further assume $\epsilon_k \geq d^{-\frac{D}{12\beta}}$.*

*Then, for any training point $\boldsymbol{x} \in \{\boldsymbol{x}_i\}_{i=n+1}^{2n}$, it holds thats*

$$|f_{h_{\boldsymbol{T}}}(\boldsymbol{x}) - \langle \boldsymbol{T}, \boldsymbol{x}^{\otimes k} \rangle| - \Delta_k \lesssim d^{-\frac{D}{8}}\|\boldsymbol{T}\|_F \tag{139}$$

*where*

$$\Delta_k := \frac{\epsilon_k}{\eta^{k\mathrm{T}_1}}\left|\mathbb{E}_{\boldsymbol{w}}[z_{\boldsymbol{T}}(\boldsymbol{w})]\right| \lesssim \epsilon_k \kappa^{k\mathrm{T}_1} \log^k d\|\boldsymbol{T}\|_F. \tag{140}$$

*Additionally we have the following moment estimations where $V_{k,\epsilon_k}$ is from Definition D.7.*

$$\mathbb{E}[h_{\boldsymbol{T}}^2] \lesssim V_{k,\epsilon_k}^2(\kappa^{2\mathrm{T}_1}\log^2 d)^k\|\boldsymbol{T}\|_F^2, \quad \|h_{\boldsymbol{T}}\|_\infty \lesssim V_{k,\epsilon_k}(\kappa^{2\mathrm{T}_1}\log^2 d)^k\|\boldsymbol{T}\|_F. \tag{141}$$

The proofs of Lemmas D.9 and D.10 are provided in Appendices D.5.6 and D.5.7, respectively.

The previous lemmas establish the pointwise approximation guarantee for an arbitrary data point $\boldsymbol{x}$ under the event $\mathcal{E}_\alpha \cap \mathcal{E}_\zeta$. The next corollary will bound the empirical squared loss over the dataset $\{\boldsymbol{x}_i\}_{i=n+1}^{2n}$.

**Corollary D.11.** *Under the sample-size condition $n \gtrsim \mathrm{T}_1^2 d \log d\kappa^{2\mathrm{T}_1} + d^{1+1/\mathrm{T}_1}\kappa^2$ also $\mathrm{T}_1 = o(\log d)$. Then under the event $\mathcal{E}_\alpha \cap \mathcal{E}_\zeta$, for any $k \leq p$ and any $k$ tensor $\boldsymbol{T}$ supported on $S^* = \{\boldsymbol{e}_1, \ldots, \boldsymbol{e}_r\}$, we have the empirical squared loss bound*

$$\frac{1}{n}\sum_{i=n+1}^{2n}\left(f_{h_{\boldsymbol{T}}}(\boldsymbol{x}_i) - \langle \boldsymbol{T}, \boldsymbol{x}_i^{\otimes k}\rangle\right)^2 \lesssim \Delta_k^2 + d^{-D/4}\|\boldsymbol{T}\|_F^2. \tag{142}$$

*where $f_{h_{\boldsymbol{T}}}$ is defined in Lemma D.10.*

*Proof of Corollary D.11.* Let $e_i := f_{h_{\boldsymbol{T}}}(\boldsymbol{x}_i) - \langle \boldsymbol{T}, \boldsymbol{x}_i^{\otimes k}\rangle$. By Lemma D.10 we have, for every training point $\boldsymbol{x}_i$, $n+1 \leq i \leq 2n$,

$$|e_i| \lesssim \Delta_k + d^{-\frac{D}{8}}\|\boldsymbol{T}\|_F, \tag{143}$$

where

$$\Delta_k := \frac{\epsilon_k}{\eta^{k\mathrm{T}_1}}\left|\mathbb{E}_{\boldsymbol{w}}[z_{\boldsymbol{T}}(\boldsymbol{w})]\right| \lesssim \epsilon_k \kappa^{k\mathrm{T}_1}\log^k d\|\boldsymbol{T}\|_F. \tag{144}$$

Squaring and averaging, we get the following bound as desired.

$$\frac{1}{n}\sum_{i=n+1}^{2n}e_i^2 \lesssim \frac{1}{n}\sum_{i=1}^{n}(\Delta_k + d^{-\frac{D}{8}}\|\boldsymbol{T}\|_F)^2 \leq 2\Delta_k^2 + 2d^{-\frac{D}{4}}\|\boldsymbol{T}\|_F^2. \tag{145}$$

The proof is complete. $\square$

**Definition D.12.** *For accuracy parameters $\{\epsilon_k\}_{k\leq p}$ define*

$$V_{p,\epsilon} := \max_{0\leq k\leq p} V_{k,\epsilon_k} \tag{146}$$

*where $V_{k,\epsilon_k}$ is the uniform bound on $\|v_k^{(\epsilon_k)}\|_\infty$ from Definition D.7.*

Notice that we already have empirical squared loss bounds for all $\boldsymbol{T}$ supported on $S^*$ in the Corollary D.11. Next we will derive empirical squared loss bounds for $f^*(\boldsymbol{x})$.

**Lemma D.13.** *We assume the event $\mathcal{E}_\alpha \cap \mathcal{E}_\zeta$, the sample-size condition $n \gtrsim \mathrm{T}_1^2 d \log d\kappa^{2\mathrm{T}_1} + d^{1+1/\mathrm{T}_1}\kappa^2$ and $\mathrm{T}_1 = o(\log d)$. Further, let $h(a,b,\boldsymbol{w}) := \sum_{k\leq p}h_{\boldsymbol{T}_k}(a,b,\boldsymbol{w})$, where $\{\boldsymbol{T}_k\}_{k=1}^p$ is defined from tensor expansion of $f^*$ introduced in Lemma E.21 $f^*(\boldsymbol{x}) = \sum_{k\leq p}\langle \boldsymbol{T}_k, \boldsymbol{x}^{\otimes k}\rangle$ and $h_{\boldsymbol{T}_k}(\cdot)$ is given by Lemma D.10 with accuracy parameters $\{\epsilon_k\}_{k\leq p}$.*

*Then we have the following empirical squared loss bound for $f^*(\boldsymbol{x})$*

$$\frac{1}{n}\sum_{i=n+1}^{2n}\left(f_h(\boldsymbol{x}_i) - f^*(\boldsymbol{x}_i)\right)^2 \lesssim \left(\sum_{k\leq p}\Delta_k\right)^2 + d^{-D/4}, \tag{147}$$

*Where $f_h(\boldsymbol{x})$ is defined as*

$$f_h(\boldsymbol{x}) := \mathbb{E}_{a,b,\boldsymbol{w}}\left[h(a,b,\boldsymbol{w})\sigma(\widetilde{\boldsymbol{w}}^{(\mathrm{T}_1)\mathsf{T}}\boldsymbol{x} + b)\right] \tag{148}$$

*Additionally we have the following moment estimation*

$$\mathbb{E}[h^2] \lesssim V_{p,\epsilon}^2(\kappa^{\mathrm{T}_1})^{2p}(\log d)^{2p}, \qquad \|h\|_\infty \lesssim V_{p,\epsilon}(\kappa^{\mathrm{T}_1})^{2p}(\log d)^{2p}. \tag{149}$$

The proof is provided in Appendix D.5.8. We next introduce the following lemma using Monte-Carlo Approximation.

**Lemma D.14.** *Under the sample-size condition $n \gtrsim \mathrm{T}_1^2 d \log d \kappa^{2\mathrm{T}_1} + d^{1+1/\mathrm{T}_1}\kappa^2$ also $\mathrm{T}_1 = o(\log d)$. Consider $a_j^* := \frac{1}{m}h(a_j, b_j, \boldsymbol{w}_j^{(0)})$ where $h(\cdot)$ is defined in Lemma D.13 and $a_j, b_j, \boldsymbol{w}_j^{(0)}$ is initialized as in Algorithm 1. Denote $\widetilde{\boldsymbol{\Theta}}^* = (\boldsymbol{a}^*, \boldsymbol{b}^{(\mathrm{T}_1)}, \widetilde{\boldsymbol{W}}^{(\mathrm{T}_1)})$.*

*Then under the event $\mathcal{E}_\mu$, we have*

$$\frac{1}{n}\sum_{i=n+1}^{2n}\left(f_{\widetilde{\boldsymbol{\Theta}}^*}(\boldsymbol{x}_i) - f^*(\boldsymbol{x}_i)\right)^2 \lesssim \left(\sum_{k\leq p}\Delta_k\right)^2 + d^{-D/4} + \frac{V_{p,\epsilon}^2(\kappa^{\mathrm{T}_1})^{4p}(\log d)^{4p+2}}{m} \tag{150}$$

*and $\|\boldsymbol{a}^*\|^2 \lesssim \frac{1}{m}V_{p,\epsilon}^2(\kappa^{\mathrm{T}_1})^{4p}(\log d)^{4p}$.*

The proof is in Appendix D.5.9. For notational simplicity, given the parameter group $\boldsymbol{\Theta} = (\boldsymbol{a}, \boldsymbol{b}, \boldsymbol{W})$, we denote the population loss and its empirical counterpart on the second-stage training set as

$$\mathcal{L}(\boldsymbol{\Theta}) := \mathbb{E}_{\boldsymbol{x}}[\ell(f_{\boldsymbol{\Theta}}(\boldsymbol{x}), f^\star(\boldsymbol{x}))] \quad \text{and} \quad \widehat{\mathcal{L}}(\boldsymbol{\Theta}) := \frac{1}{n}\sum_{i=1}^n \ell\left(f_{\boldsymbol{\Theta}}(\boldsymbol{x}_i), f^\star(\boldsymbol{x}_i)\right) \tag{151}$$

respectively. Also write $\widehat{\mathcal{L}}(\boldsymbol{\Theta})$ for $\widehat{\mathcal{L}}(f_{\boldsymbol{\Theta}})$.

In this lemma, we bound our general loss using the squared loss bound.

**Lemma D.15.** *Under the sample-size condition $n \gtrsim \mathrm{T}_1^2 d \log d \kappa^{2\mathrm{T}_1} + d^{1+1/\mathrm{T}_1}\kappa^2$, also $\mathrm{T}_1 = o(\log d)$ and under the event $\mathcal{E}_\mu$, we have*

$$\widehat{\mathcal{L}}(\widetilde{\boldsymbol{\Theta}}^*) \lesssim \mathrm{L}\left(\sum_{k\leq p}\Delta_k + \sqrt{\frac{V_{p,\epsilon}^2(\kappa^{\mathrm{T}_1})^{4p}(\log d)^{4p+2}}{m}}\right) \tag{152}$$

*where $\Delta_k$ is defined from Lemma D.10.*

The proof is presented in Appendix D.5.10.

Next, we consider the gap between the real features we learn $\boldsymbol{W}^{(\mathrm{T}_1)}$ and the approximate features $\widetilde{\boldsymbol{W}}^{(\mathrm{T}_1)}$. Recall $\widetilde{\boldsymbol{\Theta}}^* = (\boldsymbol{a}^*, \boldsymbol{b}^{(\mathrm{T}_1)}, \widetilde{\boldsymbol{W}}^{(\mathrm{T}_1)})$ and $\boldsymbol{\Theta}^* = (\boldsymbol{a}^*, \boldsymbol{b}^{(\mathrm{T}_1)}, \boldsymbol{W}^{(\mathrm{T}_1)})$.

**Lemma D.16.** *Under the sample-size condition $n \gtrsim \mathrm{T}_1^2 d \log d \kappa^{2\mathrm{T}_1} + d^{1+1/\mathrm{T}_1}\kappa^2$, also $\mathrm{T}_1 = o(\log d)$ and under the event $\mathcal{E}_\mu$, we have*

$$\left|\widehat{\mathcal{L}}(\boldsymbol{\Theta}^*) - \widehat{\mathcal{L}}(\widetilde{\boldsymbol{\Theta}}^*)\right| \lesssim \mathrm{L}\sqrt{\frac{V_{p,\epsilon}^2(\kappa^{\mathrm{T}_1})^{4p}(\log d)^{4p}}{m}}. \tag{153}$$

We provide the proof in Appendix D.5.11.

Combining the above two lemmas, we directly get the bound for $\widehat{\mathcal{L}}(f_{\boldsymbol{\Theta}^*})$.

**Lemma D.17.** *Under the same condition as the last lemma we have*

$$\widehat{\mathcal{L}}(f_{\boldsymbol{\Theta}^*}) \lesssim \mathrm{L}\left(\sum_{k\leq p}\Delta_k + \sqrt{\frac{V_{p,\epsilon}^2(\kappa^{\mathrm{T}_1})^{4p}(\log d)^{4p+2}}{m}}\right). \tag{154}$$

The proof of this bound is provided in D.5.12.

The preceding lemmas construct a parameter $\boldsymbol{\Theta}^*$ with small empirical risk $\widehat{\mathcal{L}}(f_{\boldsymbol{\Theta}^*})$. We now analyze the parameter output by the second-stage procedure,

$$\boldsymbol{\Theta}^{(\mathrm{T})} = (\boldsymbol{a}^{(\mathrm{T})}, \boldsymbol{b}^{(\mathrm{T}_1)}, \boldsymbol{W}^{(\mathrm{T}_1)}), \qquad \mathrm{T} = \mathrm{T}_1 + \mathrm{T}_2 \tag{155}$$

where $\boldsymbol{a}^{(\mathrm{T})}$ denotes the iterate after $\mathrm{T}$ iterations. We recall that only $\boldsymbol{a}$ is updated in the second stage while $\boldsymbol{b}$ and $\boldsymbol{W}$ remain at their values at time $\mathrm{T}_1$. The next lemma gives an upper bound on the empirical risk $\widehat{\mathcal{L}}(\boldsymbol{\Theta}^{(\mathrm{T})})$ of this output.

For notation convenience, we define $J := \mathrm{L}\left(\sum_{k\leq p}\Delta_k + \sqrt{\frac{V_{p,\epsilon}^2(\kappa^{\mathrm{T}_1})^{4p}(\log d)^{4p+2}}{m}}\right)$ and $U = \sqrt{\frac{V_{p,\epsilon}^2(\kappa^{\mathrm{T}_1})^{4p}(\log d)^{4p}}{m}}$.

**Lemma D.18.** *We consider doing gradient descent on the ridge-regularized empirical loss*

$$F(\boldsymbol{a}) := \widehat{\mathcal{L}}\big((\boldsymbol{a}, \boldsymbol{b}^{(\mathrm{T}_1)}, \boldsymbol{W}^{(\mathrm{T}_1)})\big) + \frac{\beta_2}{2}\|\boldsymbol{a}\|^2 \tag{156}$$

*with the hyperparameters $\beta_2 := \frac{J}{U^2}$ and $\eta_2 = \frac{U^2}{L\sqrt{m}U^2+2J}$.*

*Then for any $\mathrm{T}_2 \gtrsim \frac{U^2 Lm}{J}\log(Um)$ also under the event $\mathcal{E}_\mu$, our second stage algorithm returns $\boldsymbol{a}^{(\mathrm{T})}$ such that for $\boldsymbol{\Theta}^{(\mathrm{T})} := (\boldsymbol{a}^{(\mathrm{T})}, \boldsymbol{b}^{(\mathrm{T}_1)}, \boldsymbol{W}^{(\mathrm{T}_1)})$*

$$\widehat{\mathcal{L}}(\boldsymbol{\Theta}^{(\mathrm{T})}) \lesssim J \quad \text{and} \quad \|\boldsymbol{a}^{(\mathrm{T})}\| \lesssim U. \tag{157}$$

The proof is provided in Appendix D.5.13. The following lemma bounds the gap between empirical loss and population loss for $\boldsymbol{\Theta}^{(\mathrm{T})}$.

**Lemma D.19.** *Under the event $\mathcal{E}$, where $\mathcal{E}$ is defined in Defination D.37 and the sample-size condition $n \gtrsim \mathrm{T}_1^2 d \log d \kappa^{2\mathrm{T}_1} + d^{1+1/\mathrm{T}_1}\kappa^2$, also $\mathrm{T}_1 = o(\log d)$ we have*

$$\left|\mathcal{L}(f_{\boldsymbol{\Theta}^{(\mathrm{T})}}) - \widehat{\mathcal{L}}(f_{\boldsymbol{\Theta}^{(\mathrm{T})}})\right| \lesssim L\sqrt{\frac{V_{p,\epsilon}^2 (\kappa^{\mathrm{T}_1})^{4p}(\log d)^{4p+1}}{n}}. \tag{158}$$

The proof is provided in Appendix D.5.14. Then the following proposition combines the previous lemmas and gives a quick proof map of bounding the excess population risk $\mathcal{L}(f_{\boldsymbol{\Theta}^{(\mathrm{T})}})$.

**Proposition D.20.** *On the event $\mathcal{E}$ and the sample-size condition $n \gtrsim \mathrm{T}_1^2 d \log d \kappa^{2\mathrm{T}_1} + d^{1+1/\mathrm{T}_1}\kappa^2$, also $\mathrm{T}_1 = o(\log d)$ the excess population risk satisfies*

$$\mathcal{L}(f_{\boldsymbol{\Theta}^{(\mathrm{T})}}) - \mathcal{L}(f^*) \lesssim L\left(\sum_{k\leq p}\Delta_k + \sqrt{\frac{V_{p,\epsilon}^2 \kappa^{4p\mathrm{T}_1}(\log d)^{4p+1}}{n}} + \sqrt{\frac{V_{p,\epsilon}^2 \kappa^{4p\mathrm{T}_1}(\log d)^{4p+2}}{m}}\right). \tag{159}$$

*Proof of Proposition D.20.* Note that to upper bound the population loss $\mathcal{L}(\boldsymbol{\Theta}^{(\mathrm{T})})$ of the parameter group $\boldsymbol{\Theta}^{(\mathrm{T})}$ returned by Algorithm 1, we first construct an "ideal" parameter group $\boldsymbol{\Theta}^\star$ which lies in a proper parameter class $\boldsymbol{\Theta}^\star \in \mathcal{F}$ (small $\|\boldsymbol{\Theta}^\star\|$) and yields small empirical error.

So the first several lemmas this section give an upper bound for $\widehat{\mathcal{L}}(f_{\boldsymbol{\Theta}^*})$ through $\widetilde{\boldsymbol{\Theta}}^*$ , which is introduced to decouple the dependence

$$\underbrace{\widehat{\mathcal{L}}(f_{\boldsymbol{\Theta}^*})}_{\text{Lemma D.17}} = \underbrace{\left[\widehat{\mathcal{L}}(f_{\boldsymbol{\Theta}^*}) - \widehat{\mathcal{L}}(f_{\widetilde{\boldsymbol{\Theta}}^*})\right]}_{\text{Lemma D.16}} + \underbrace{\left[\widehat{\mathcal{L}}(f_{\widetilde{\boldsymbol{\Theta}}^*}) - \widehat{\mathcal{L}}(f^*)\right]}_{\text{Lemma D.15}}$$

$$\lesssim L\sqrt{\frac{V_{p,\epsilon}^2 (\kappa^{\mathrm{T}_1})^{4p}(\log d)^{4p}}{m}} + L\left(\sum_{k\leq p}\Delta_k + \sqrt{\frac{V_{p,\epsilon}^2 (\kappa^{\mathrm{T}_1})^{4p}(\log d)^{4p+2}}{m}}\right) \tag{160}$$

$$\lesssim L\left(\sum_{k\leq p}\Delta_k + \sqrt{\frac{V_{p,\epsilon}^2 (\kappa^{\mathrm{T}_1})^{4p}(\log d)^{4p+2}}{m}}\right).$$

And recall that we denote $J := L\left(\sum_{k\leq p}\Delta_k + \sqrt{\frac{V_{p,\epsilon}^2(\kappa^{\mathrm{T}_1})^{4p}(\log d)^{4p+2}}{m}}\right)$.

Also note that we can decompose the error $\mathcal{L}(\boldsymbol{\Theta}^{(\mathrm{T})})$ in the equation that follows

$$\mathcal{L}(f_{\boldsymbol{\Theta}^{(\mathrm{T})}}) - \widehat{\mathcal{L}}(f_{\boldsymbol{\Theta}^*}) = \underbrace{\left[\mathcal{L}(f_{\boldsymbol{\Theta}^{(\mathrm{T})}}) - \widehat{\mathcal{L}}(f_{\boldsymbol{\Theta}^{(\mathrm{T})}})\right]}_{\text{Lemma D.19}} + \underbrace{\left[\widehat{\mathcal{L}}(f_{\boldsymbol{\Theta}^{(\mathrm{T})}}) - \widehat{\mathcal{L}}(f_{\boldsymbol{\Theta}^*})\right]}_{\text{Lemma D.18}}$$

$$\lesssim L\sqrt{\frac{V_{p,\epsilon}^2 (\kappa^{\mathrm{T}_1})^{4p}(\log d)^{4p+1}}{n}} + J$$

$$\lesssim L\left(\sum_{k\leq p}\Delta_k + \sqrt{\frac{V_{p,\epsilon}^2 \kappa^{4p\mathrm{T}_1}(\log d)^{4p+1}}{n}} + \sqrt{\frac{V_{p,\epsilon}^2 \kappa^{4p\mathrm{T}_1}(\log d)^{4p+2}}{m}}\right). \tag{161}$$

Combining this two yields

$$\mathcal{L}(f_{\boldsymbol{\Theta}^{(\mathrm{T})}}) - \mathcal{L}(f^*) \lesssim \mathrm{L}\left(\sum_{k \le p} \Delta_k + \sqrt{\frac{V_{p,\epsilon}^2 (\kappa^{\mathrm{T}_1})^{4p}(\log d)^{4p+1}}{n}} + \sqrt{\frac{V_{p,\epsilon}^2 (\kappa^{\mathrm{T}_1})^{4p}(\log d)^{4p+2}}{m}}\right) \tag{162}$$

which is precisely the claimed bound. This completes the proof.

$\square$

Back to the main proof, in the previous proposition, we have the generalization bound

$$\mathcal{L}(f_{\boldsymbol{\Theta}^{(\mathrm{T})}}) \lesssim \mathrm{L}\left(\sum_{k \le p} \Delta_k + \sqrt{\frac{V_{p,\epsilon}^2 (\kappa^{\mathrm{T}_1})^{4p}(\log d)^{4p+1}}{n}} + \sqrt{\frac{V_{p,\epsilon}^2 (\kappa^{\mathrm{T}_1})^{4p}(\log d)^{4p+2}}{m}}\right) \tag{163}$$

We note that the bound is depending on both $\{\Delta_k\}_{k=0}^p$ and $V_{p,\epsilon}$, which are left unspecified. Fixing $\beta > 0$, we notice that $\Delta_k$ has the bound relying on $\epsilon_k$ which is

$$\begin{aligned} \Delta_k &= \frac{\epsilon_k}{\eta^{k\mathrm{T}_1}}\left|\mathbb{E}_{\boldsymbol{w}}[z_{\boldsymbol{T}_k}(\boldsymbol{w})]\right| \\ &\lesssim \epsilon_k \kappa^{k\mathrm{T}_1} \log^k d \|\boldsymbol{T}_k\|_F \lesssim \epsilon_k \kappa^{p\mathrm{T}_1} \log^p d. \end{aligned} \tag{164}$$

On the other hand, we observe that $V_{k,\epsilon_k} \lesssim \epsilon_k^{-\beta}$. Thus we need to choose proper $\{\epsilon_k\}_{k=0}^p$ to keep the balance of $V_{k,\epsilon_k}$ and $\Delta_k$. For notation convenience, let

$$A := \kappa^{2p\mathrm{T}_1}(\log d)^{2p+1}, \qquad C_m := \sqrt{\frac{1}{m}}, \qquad C_n := \sqrt{\frac{1}{n}}. \tag{165}$$

Thus the $\epsilon$-dependent contribution is upper bounded (up to absolute constants) by

$$\Phi(\{\epsilon_k\}_{k \le p}) := A\left(\sum_{k \le p} \epsilon_k + (C_m + C_n)V_{p,\epsilon}\right) \tag{166}$$

Specifically, when $\beta = 0$, from Definition D.7, we know that the approximation precision $\rho$ can be zero, which indicates that $\epsilon_k = 0$ for all $\{\epsilon_k\}_{k=0}^p$ and also $V_{p,\epsilon} \lesssim 1$. Thus

$$\Phi = A\left((C_m + C_n)V_{p,\epsilon}\right) \lesssim A(C_m + C_n) \tag{167}$$

Furthermore, for $\beta > 0$, plugging in $V_{k,\epsilon_k} \lesssim \epsilon_k^{-\beta}$, we have

$$\Phi(\{\epsilon_k\}_{k \le p}) := A\left(\sum_{k \le p} \epsilon_k + (C_m + C_n)\max_{k \le p}\left\{\epsilon_k^{-\beta}\right\}\right) \tag{168}$$

The following lemma gives the minimum value of $\Phi$.

**Lemma D.21.** *Fix $\beta > 0$. Consider $\Phi$ as a function of $\{\epsilon_k\}_{k=0}^p$. Then the minimizer is unique and attained at the symmetric point*

$$\epsilon_0^* = \cdots = \epsilon_p^* = \left(\frac{\beta(C_m + C_n)}{p+1}\right)^{\frac{1}{\beta+1}}. \tag{169}$$

*Moreover, it holds that*

$$\min_{\{\epsilon_k > 0\}} \Phi = A \cdot (\beta+1)\beta^{-\frac{\beta}{\beta+1}}(p+1)^{\frac{\beta}{\beta+1}}(C_m + C_n)^{\frac{1}{\beta+1}} \lesssim A(C_m + C_n)^{\frac{1}{\beta+1}}. \tag{170}$$

The proof is provided in Appendix D.5.15. The following corollary is direct from the previous statement.

**Corollary D.22.** *When $\beta = 0$, we have* $\inf \Phi = A(C_m + C_n)$.

Now we state the main theorem. For the theorem stated below, we recall that we assume $\mathrm{T}_1 = o(\log d)$, $n \gtrsim d \log d\kappa^{2\mathrm{T}_1} + d^{1+1/\mathrm{T}_1}\kappa^2$ in addition to the assumptions in Section 2 and Section 3.

**Theorem 2** (Main Theorem). *Under the assumptions above, there exists a set of hyperparameters*

$$\eta_1 = \frac{1}{C_\eta}\left(\frac{d}{4D^2\log^2 d}\right)^{\frac{1}{2T_1}}, \quad \beta_1 = \frac{1}{\eta_1}, \quad \eta_2 = \frac{U^2}{L\sqrt{m}U^2 + 2J}, \quad \beta_2 = \frac{J}{U^2} \quad (171)$$

*such that for any $T_2 \gtrsim \frac{U^2 L m}{J}\log(Um)$, the population risk satisfies*

$$\mathbb{E}_{\boldsymbol{x}}[\ell(f_{\boldsymbol{\Theta}^{(T)}}(\boldsymbol{x}), y)] \lesssim \kappa^{2pT_1}(\log d)^{2p+1}\left(\sqrt{\frac{1}{m}} + \sqrt{\frac{1}{n}}\right)^{\frac{1}{\beta+1}} \quad (172)$$

*under the high probability event $\mathcal{E}$, which happens with probability at least $1 - \mathcal{O}(d^{-D/2})$, where $C_\eta$ is an absolute constant from Lemma C.6 and $J, U$ are quantities defined in D.3.*

We further estimate sample and time complexity in the next lemmas, showing that they are both nearly optimal.

**Lemma D.23.** *Under the setting $T_1 := \lceil\sqrt{\frac{\log d}{\log \kappa}}\rceil$ we have*

$$T_1^2 d\log d\kappa^{2T_1} + d^{1+1/T_1}\kappa^2 \lesssim \kappa^2 d^{1+2\sqrt{\frac{\log \kappa}{\log d}}}(\log d)^2 \quad (173)$$

The proof is provided in Apendix D.5.16. Note that from Theorem 2, for $n \gtrsim T_1^2 d\log d\kappa^{2T_1} + d^{1+1/T_1}\kappa^2$ the main result applies. Since our sample-complexity upper bound coincides with that threshold, the corollary follows immediately.

**Corollary D.24** (Sample Complexity). *The sample complexity is $\widetilde{\Theta}(\kappa^2 d^{1+2\sqrt{\frac{\log \kappa}{\log d}}})$, where $d^{2\sqrt{\frac{\log \kappa}{\log d}}}$ is subpolynomial in $d$.*

Next we first estimate $T_2$ under particular $m, n$ choices and other hyperparameters.

**Lemma D.25.** *Fix the set of hyperparameters*

$$T_1 := \lceil\sqrt{\frac{\log d}{\log \kappa}}\rceil \qquad \beta_2 := \frac{J}{U^2} \qquad \eta_2 = \frac{U^2}{L\sqrt{m}U^2 + 2J} \qquad T_2 = \frac{U^2 L m}{J}\log(Um) \quad (174)$$

*where $J$ and $U$ is defined as*

$$J := L\left(\sum_{k\le p}\Delta_k + \sqrt{\frac{V_{p,\epsilon}^2(\kappa^{T_1})^{4p}(\log d)^{4p+2}}{m}}\right) \text{ and } U = \sqrt{\frac{V_{p,\epsilon}^2(\kappa^{T_1})^{4p}(\log d)^{4p}}{m}}. \quad (175)$$

*$\{\Delta_k\}_{k\le p}$ is defined in Lemma D.10 and $V_{p,\epsilon}$ is defined in Defination D.12. Note that $\{\epsilon_k\}_{k=0}^p$ was fixed in Lemma D.21, conclusively $J$ and $U$ are also fixed.*

*Furthermore, fix the sample size and net width as $m = (\log d)^{4p(\beta+1)+1}d^{4p(\beta+1)\sqrt{\frac{\log \kappa}{\log d}}}$, $n = C\kappa^2(\log d)^2 d^{1+2\sqrt{\frac{\log \kappa}{\log d}}}$ where $C$ is a large universal constant. Then under the above setting,*

$$T_2 \lesssim \kappa^{5p}(\log d)^{8p\beta+13p+2}d^{(8p\beta+13p)\sqrt{\frac{\log \kappa}{\log d}}}. \quad (176)$$

*By ignoring the logarithmic terms, we can get $T_2 = \widetilde{\mathcal{O}}(\kappa^{5p}d^{(8p\beta+13p)\sqrt{\frac{\log \kappa}{\log d}}})$.*

The proof is provided in Appendix D.5.17.

We note that the total time complexity is upper bound by $nmd \cdot T$ up to absolute constants, so plugging $n$ and $m$, the next lemma follows directly.

**Lemma D.26** (Time complexity). *Under the same condition as the last lemma, we have $T_{complexity} \lesssim \kappa^{5p+2}(\log d)^{12p\beta+17p+5}d^{2+(12p\beta+17p+2)\sqrt{\frac{\log \kappa}{\log d}}}$*

The proof is provided in Appendix D.5.18.

## D.4   Proof under $\beta = 0$ and Assumption $3'$

We now extend our analysis to a more general loss class (Assumption $3'$) under an additional assumption $\beta = 0$. For clarity, we provide a complete proof here.

Under Assumption $3'$ and $\beta = 0$, Lemma D.5.1 to Lemma D.14 still hold. In particular, given that $\beta = 0$ we can conclude that $\Delta_k = 0$ for all $k \leq p$ and $V_{p,\epsilon}$ is bounded by universal constant. We restate this as the following lemma, which is parallel to Lemma D.14 in the previous section.

**Lemma D.27.** *Under the sample-size condition* $n \gtrsim \mathrm{T}_1^2 d \log d \kappa^{2\mathrm{T}_1} + d^{1+1/\mathrm{T}_1} \kappa^2$ *also* $\mathrm{T}_1 = o(\log d)$. *Consider* $a_j^* := \frac{1}{m} h(a_j, b_j, \boldsymbol{w}_j^{(0)})$ *where* $h(\cdot)$ *is defined in Lemma D.13 and* $a_j, b_j, \boldsymbol{w}_j^{(0)}$ *is initialized as in Algorithm 1. Denote* $\widetilde{\boldsymbol{\Theta}}^* = (\boldsymbol{a}^*, \boldsymbol{b}^{(\mathrm{T}_1)}, \widetilde{\boldsymbol{W}}^{(\mathrm{T}_1)})$.

*Then under the event* $\mathcal{E}_\mu$, *we have*

$$\frac{1}{n} \sum_{i=n+1}^{2n} \left( f_{\widetilde{\boldsymbol{\Theta}}^*}(\boldsymbol{x}_i) - f^*(\boldsymbol{x}_i) \right)^2 \lesssim d^{-D/4} + \frac{(\kappa^{\mathrm{T}_1})^{4p} (\log d)^{4p+2}}{m} \tag{177}$$

*and* $\|\boldsymbol{a}^*\|^2 \lesssim \frac{1}{m} (\kappa^{\mathrm{T}_1})^{4p} (\log d)^{4p}$.

We sequentially present the parallel versions of the other theorems in the previous section. In the following lemma, we bound our general loss using the squared loss bound. For notation convenience, we denote $\mathrm{L}_{\mathsf{aug}} := \mathrm{L}\kappa^{2p\mathrm{T}_1}(\log d)^{2p}$. It can be observed that the original conclusion remains almost unchanged, except that L is replaced by the new $\mathrm{L}_{\mathsf{aug}}$.

**Lemma D.28.** *Under* $n \gtrsim \mathrm{T}_1^2 d \log d \kappa^{2\mathrm{T}_1} + d^{1+1/\mathrm{T}_1} \kappa^2$, *also* $\mathrm{T}_1 = o(\log d)$ *and under the event* $\mathcal{E}_\mu$, *we have*

$$\widehat{\mathcal{L}}(\widetilde{\boldsymbol{\Theta}}^*) \lesssim \mathrm{L}_{\mathsf{aug}} \sqrt{\frac{\kappa^{4p\mathrm{T}_1}(\log d)^{4p+2}}{m}} \tag{178}$$

.

*Proof.* Under Assumption $3'$, for any $z, z', y \in \mathbb{R}$,

$$|\ell(z, y) - \ell(z', y)| \leq \mathrm{L} \left( 1 + |z| + |z'| + |y| \right) |z - z'|. \tag{179}$$

Combining with the above Lemma, we have

$$\widehat{\mathcal{L}}(\widetilde{\boldsymbol{\Theta}}^*) = \frac{1}{n} \sum_{i=1}^{n} \ell \left( f_{\widetilde{\boldsymbol{\Theta}}^*}(\boldsymbol{x}_i), f^\star(\boldsymbol{x}_i) \right) = \frac{1}{n} \sum_{i=1}^{n} \left( \ell \left( f_{\widetilde{\boldsymbol{\Theta}}^*}(\boldsymbol{x}_i), f^\star(\boldsymbol{x}_i) \right) - \ell \left( f^\star(\boldsymbol{x}_i), f^\star(\boldsymbol{x}_i) \right) \right)$$

$$\lesssim \frac{1}{n} \sum_{i=1}^{n} \mathrm{L} \left( 1 + |f_{\widetilde{\boldsymbol{\Theta}}^*}(\boldsymbol{x}_i)| + |f^\star(\boldsymbol{x}_i)| \right) |f_{\widetilde{\boldsymbol{\Theta}}^*}(\boldsymbol{x}_i) - f^\star(\boldsymbol{x}_i)| \tag{180}$$

Furthermore, by the same argument as in Lemma D.9, we have $\left| (\boldsymbol{w}_j^{(\mathrm{T})})^\mathsf{T} \boldsymbol{x} \right| \lesssim 1$ for all $j$, thus we have

$$\left\| \sigma(\widetilde{\boldsymbol{W}}^{(\mathrm{T}_1)} \boldsymbol{x} + \boldsymbol{b}) \right\| \lesssim \sqrt{m} \tag{181}$$

$$|f_{\widetilde{\boldsymbol{\Theta}}^*}(\boldsymbol{x}_i)| = \boldsymbol{a}^* \sigma(\widetilde{\boldsymbol{W}}^{(\mathrm{T}_1)} \boldsymbol{x} + \boldsymbol{b}) \lesssim U\sqrt{m}. \tag{182}$$

Also note that $U \lesssim \sqrt{\frac{(\kappa^{\mathrm{T}_1})^{4p}(\log d)^{4p}}{m}}$ and $|f^\star(\boldsymbol{x}_i)| \lesssim (\log d)^p$. Thus by Jensen's inequality and the Lipschitz property of $\ell$, we have

$$\widehat{\mathcal{L}}(\widetilde{\boldsymbol{\Theta}}^*) \leq \frac{1}{n} \sum_{i=1}^{n} \mathrm{L} \left( 1 + |f_{\widetilde{\boldsymbol{\Theta}}^*}(\boldsymbol{x}_i)| + |f^\star(\boldsymbol{x}_i)| \right) |f_{\widetilde{\boldsymbol{\Theta}}^*}(\boldsymbol{x}_i) - f^\star(\boldsymbol{x}_i)| \tag{183}$$

$$\lesssim \mathrm{L}(\kappa^{\mathrm{T}_1})^{2p}(\log d)^{2p} \frac{1}{n} \left( \sum_{i=1}^{n} |f_{\widetilde{\boldsymbol{\Theta}}^*}(\boldsymbol{x}_i) - f^\star(\boldsymbol{x}_i)| \right) \lesssim \mathrm{L}_{\mathsf{aug}} \sqrt{\frac{\kappa^{4p\mathrm{T}_1}(\log d)^{4p+2}}{m}}. \tag{184}$$

The proof is complete. $\qquad\square$

Next, we consider the gap between the real features we learn $\boldsymbol{W}^{(\mathrm{T}_1)}$ and the approximate features $\widetilde{\boldsymbol{W}}^{(\mathrm{T}_1)}$. Recall $\widetilde{\boldsymbol{\Theta}}^* = (\boldsymbol{a}^*, \boldsymbol{b}^{(\mathrm{T}_1)}, \widetilde{\boldsymbol{W}}^{(\mathrm{T}_1)})$ and $\boldsymbol{\Theta}^* = (\boldsymbol{a}^*, \boldsymbol{b}^{(\mathrm{T}_1)}, \boldsymbol{W}^{(\mathrm{T}_1)})$.

**Lemma D.29.** *Under the sample-size condition* $n \gtrsim \mathrm{T}_1^2 d \log d \kappa^{2\mathrm{T}_1} + d^{1+1/\mathrm{T}_1}\kappa^2$, *also* $\mathrm{T}_1 = o(\log d)$ *and under the event* $\mathcal{E}_\mu$, *we have*

$$\left| \widehat{\mathcal{L}}(\boldsymbol{\Theta}^*) - \widehat{\mathcal{L}}(\widetilde{\boldsymbol{\Theta}}^*) \right| \lesssim \mathrm{L}_{\mathsf{aug}} \sqrt{\frac{(\kappa^{\mathrm{T}_1})^{4p}(\log d)^{4p}}{m}}. \tag{185}$$

We omit the proof here, since the only change is to change L to $\mathrm{L}_{\mathsf{aug}}$.

Combining the above two lemmas, we directly get the bound for $\widehat{\mathcal{L}}(f_{\boldsymbol{\Theta}^*})$.

**Lemma D.30.** *Under the same condition as Lemma D.28, we have* $\widehat{\mathcal{L}}(f_{\boldsymbol{\Theta}^*}) \lesssim \mathrm{L}_{\mathsf{aug}} \sqrt{\frac{(\kappa^{\mathrm{T}_1})^{4p}(\log d)^{4p+2}}{m}}$.

We now analyze the parameter output by the second-stage procedure,

$$\boldsymbol{\Theta}^{(\mathrm{T})} = (\boldsymbol{a}^{(\mathrm{T})}, \boldsymbol{b}^{(\mathrm{T}_1)}, \boldsymbol{W}^{(\mathrm{T}_1)}) \qquad \mathrm{T} = \mathrm{T}_1 + \mathrm{T}_2 \tag{186}$$

in the new setting. Define $J' = \mathrm{L}_{\mathsf{aug}} \sqrt{\frac{(\kappa^{\mathrm{T}_1})^{4p}(\log d)^{4p+2}}{m}}$ and $U' = \sqrt{\frac{(\kappa^{\mathrm{T}_1})^{4p}(\log d)^{4p}}{m}}$. This is similar to the above section, but simpler under the new setting.

**Lemma D.31.** *We consider doing gradient descent on the ridge-regularized empirical loss*

$$F(\boldsymbol{a}) := \widehat{\mathcal{L}}\big((\boldsymbol{a}, \boldsymbol{b}^{(\mathrm{T}_1)}, \boldsymbol{W}^{(\mathrm{T}_1)})\big) + \frac{\beta_2}{2}\|\boldsymbol{a}\|^2 \tag{187}$$

*with the hyperparameters* $\beta_2 := \frac{J'}{U'^2}$ *and* $\eta_2 = \frac{U'^2}{LmU'^2+J'}$.

*Then for any* $\mathrm{T}_2 \gtrsim \frac{m}{\log d}\log(m\kappa^\mathrm{T}\log d)$ *also under the event* $\mathcal{E}_\mu$, *our second stage algorithm returns* $\boldsymbol{a}^{(\mathrm{T})}$ *such that for* $\boldsymbol{\Theta}^{(\mathrm{T})} := (\boldsymbol{a}^{(\mathrm{T})}, \boldsymbol{b}^{(\mathrm{T}_1)}, \boldsymbol{W}^{(\mathrm{T}_1)})$

$$\widehat{\mathcal{L}}\big(\boldsymbol{\Theta}^{(\mathrm{T})}\big) \lesssim J' \quad and \quad \|\boldsymbol{a}^{(\mathrm{T})}\| \lesssim U'. \tag{188}$$

*Proof.* Consider the training objective $F(\boldsymbol{a}) = \widehat{\mathcal{L}}\big((\boldsymbol{a}, \boldsymbol{b}^{(\mathrm{T}_1)}, \boldsymbol{W}^{(\mathrm{T}_1)})\big) + \frac{1}{2}\beta_2\|\boldsymbol{a}\|^2$ with $\beta_2 = \frac{J'}{U'^2}$.

Let

$$\boldsymbol{a}^{(\infty)} := \arg\min_{\boldsymbol{a}\in\mathbb{R}^m} F(\boldsymbol{a}) \tag{189}$$

be the minimizer of $F$.

We further denote $\epsilon_0$ is the prescribed optimization tolerance $\|\boldsymbol{a}^{(\mathrm{T})} - \boldsymbol{a}^{(\infty)}\| \le \epsilon_0$. Furthermore, let $\epsilon_0 = \frac{1}{m\mathrm{L}_{\mathsf{aug}}}$.

By optimality of $\boldsymbol{a}^{(\infty)}$ and the *constructed* comparator $\boldsymbol{a}^*$ we have

$$F(\boldsymbol{a}^{(\infty)}) \le F(\boldsymbol{a}^*) = \widehat{\mathcal{L}}(\boldsymbol{a}^*) + \frac{\beta_2}{2}\|\boldsymbol{a}^*\|_2^2. \tag{190}$$

Here we only use that $\boldsymbol{a}^*$ is a feasible point with $\|\boldsymbol{a}^*\| \lesssim U'$ and $\widehat{\mathcal{L}}(\boldsymbol{a}^*) \lesssim J'$ (Lemma D.30).

On the other hand,

$$F(\boldsymbol{a}^{(\infty)}) = \widehat{\mathcal{L}}(\boldsymbol{a}^{(\infty)}) + \frac{\beta_2}{2}\|\boldsymbol{a}^{(\infty)}\|^2 \ge \frac{\beta_2}{2}\|\boldsymbol{a}^{(\infty)}\|^2, \tag{191}$$

so combining with (331) gives

$$\|\boldsymbol{a}^{(\infty)}\|^2 \le \|\boldsymbol{a}^*\|^2 + \frac{2}{\beta_2}\widehat{\mathcal{L}}(\boldsymbol{a}^*) \lesssim U'^2 + \frac{2J'}{\beta_2} = U'^2 + \frac{2J'}{J'/U'^2} \lesssim U'^2. \tag{192}$$

Thus

$$\|\boldsymbol{a}^{(\infty)}\| \lesssim U'. \tag{193}$$

and also

$$\|\boldsymbol{a}^{(\mathrm{T})}\| \leq \|\boldsymbol{a}^{(\infty)}\| + \|\boldsymbol{a}^{(\mathrm{T})} - \boldsymbol{a}^{(\infty)}\| \leq \epsilon_0 + U' \lesssim U' \tag{194}$$

Moreover, from (331) and (334),

$$\widehat{\mathcal{L}}(\boldsymbol{a}^{(\infty)}) = F(\boldsymbol{a}^{(\infty)}) - \frac{\beta_2}{2}\|\boldsymbol{a}^{(\infty)}\|_2^2 \leq F(\boldsymbol{a}^*) \lesssim J' + \frac{\beta_2}{2}U'^2 \lesssim J'. \tag{195}$$

Thus we have

$$|F(\boldsymbol{a}^{(\mathrm{T})}) - F(\boldsymbol{a}^{(\infty)})| \lesssim \sup_{i\in[n+1,2n]} \left\{ \mathrm{L}_{\mathsf{aug}} \big| \boldsymbol{a}^{(\mathrm{T})^{\mathsf{T}}} \sigma\big(\boldsymbol{W}^{(\mathrm{T})}\boldsymbol{x}_i + \boldsymbol{b}^{(\mathrm{T})}\big) - \boldsymbol{a}^{(\infty)^{\mathsf{T}}} \sigma\big(\boldsymbol{W}^{(\mathrm{T})}\boldsymbol{x}_i + \boldsymbol{b}^{(\mathrm{T})}\big) \big| \right\}$$
$$+ \beta_2(\|\boldsymbol{a}^{(\mathrm{T})}\| + \|\boldsymbol{a}^{(\infty)}\|)\|\boldsymbol{a}^{(\mathrm{T})} - \boldsymbol{a}^{(\infty)}\|$$
$$\lesssim \mathrm{L}_{\mathsf{aug}}\|\boldsymbol{a}^{(\mathrm{T})} - \boldsymbol{a}^{(\infty)}\| \cdot \sup_{i\in[n+1,2n]} \|\sigma\big(\boldsymbol{W}^{(\mathrm{T})}\boldsymbol{x}_i + \boldsymbol{b}^{(\mathrm{T})}\big)\|$$
$$+ \beta_2(\|\boldsymbol{a}^{(\mathrm{T})}\| + \|\boldsymbol{a}^{(\infty)}\|)\|\boldsymbol{a}^{(\mathrm{T})} - \boldsymbol{a}^{(\infty)}\| \tag{196}$$

By previous proof $\|\sigma\big(\boldsymbol{W}^{(\mathrm{T})}\boldsymbol{x}_i + \boldsymbol{b}^{(\mathrm{T})}\big)\|_\infty$ can be bounded by an absolute constant for all $i$ under event $\mathcal{E}$. Thus we can have a loose $\ell^2$ norm bound and

$$|F(\boldsymbol{a}^{(\mathrm{T})}) - F(\boldsymbol{a}^{(\infty)})| \lesssim \mathrm{L}_{\mathsf{aug}}\|\boldsymbol{a}^{(\mathrm{T})} - \boldsymbol{a}^{(\infty)}\| \cdot \sup_{i\in[n+1,2n]} \|\sigma\big(\boldsymbol{W}^{(\mathrm{T})}\boldsymbol{x}_i + \boldsymbol{b}^{(\mathrm{T})}\big)\|$$
$$+ \beta_2(\|\boldsymbol{a}^{(\mathrm{T})}\| + \|\boldsymbol{a}^{(\infty)}\|)\|\boldsymbol{a}^{(\mathrm{T})} - \boldsymbol{a}^{(\infty)}\|$$
$$\lesssim \mathrm{L}_{\mathsf{aug}}\|\boldsymbol{a}^{(\mathrm{T})} - \boldsymbol{a}^{(\infty)}\|\sqrt{m} + \beta_2 U'\|\boldsymbol{a}^{(\mathrm{T})} - \boldsymbol{a}^{(\infty)}\|$$
$$\lesssim (\mathrm{L}_{\mathsf{aug}}\sqrt{m} + \beta_2 U')\epsilon_0 \lesssim J' \tag{197}$$

Combining with the same argument as above,

$$\widehat{\mathcal{L}}(\boldsymbol{a}^{(\mathrm{T})}) \lesssim J'. \tag{198}$$

By Assumption $3'$ on the loss function $\ell$, the function $F(\boldsymbol{a}) - \frac{\beta_2}{2}\|\boldsymbol{a}\|^2$ is convex and satisfies the gradient Lipschitz condition

By the similar arguments as in the proof of Lemma D.18 , the lip constant for $F$ satisfy $L_{\mathsf{lip}} \lesssim \mathrm{L}m + J'/U'^2$.

Therefore, by Lemma E.13, if the step size $\eta_2 \leq \frac{1}{L_{\mathsf{lip}}}$, gradient descent will approximate $\boldsymbol{a}^{(\infty)}$ to arbitrary accuracy in

$$\mathrm{T}_2 \gtrsim \frac{1}{\eta_2\beta_2} \log\Big(\|\boldsymbol{a}^{(\mathrm{T}_1)} - \boldsymbol{a}^{(\infty)}\|^2/\epsilon_0^2\Big) \tag{199}$$

steps. Note that when the step size $\eta_2 = \frac{1}{\mathrm{L}m + J'/U'^2}$

$$\frac{1}{\eta_2\beta_2} \lesssim \frac{U'^2\mathrm{L}m}{J'} \tag{200}$$

Using the assumption that $m \lesssim d^{\frac{D}{4}}$ and $\|\boldsymbol{a}^{(\infty)}\| \lesssim U, \|\boldsymbol{a}^{(\mathrm{T}_1)}\| \lesssim 1$, we can further simplify:

$$\frac{1}{\eta_2\beta_2} \log\left(\frac{\|\boldsymbol{a}^{(\mathrm{T}_1)} - \boldsymbol{a}^{(\infty)}\|^2}{\epsilon_0^2}\right) \lesssim \frac{U'^2\mathrm{L}m}{J'} \log(U'm\mathrm{L}_{\mathsf{aug}}) \lesssim \frac{m}{\log d} \log(m\kappa^T \log d) \tag{201}$$

Thus, we conclude that gradient descent will approximate $\boldsymbol{a}^{(\infty)}$ to arbitrary accuracy in at most

$$\mathrm{T}_2 \gtrsim \frac{m}{\log d} \log(m\kappa^T \log d) \tag{202}$$

steps. The proof is complete. $\qquad\square$

The following lemma bounds the gap between empirical loss and population loss.

**Lemma D.32.** *Under the event $\mathcal{E}$, where $\mathcal{E}$ is defined in Defination D.37 and the sample-size condition $n \gtrsim \mathrm{T}_1^2 d \log d \kappa^{2\mathrm{T}_1} + d^{1+1/\mathrm{T}_1}\kappa^2$, also $\mathrm{T}_1 = o(\log d)$ we have*

$$\left| \mathcal{L}(f_{\boldsymbol{\Theta}^{(\mathrm{T})}}) - \widehat{\mathcal{L}}(f_{\boldsymbol{\Theta}^{(\mathrm{T})}}) \right| \lesssim \mathrm{L}_{\mathsf{aug}} \sqrt{\frac{(\kappa^{\mathrm{T}_1})^{4p}(\log d)^{4p+1}}{n}}. \tag{203}$$

*Proof.* Use the same argument as the proof of Lemma D.19 in Appendix D.5.14 and use the same truncation, we can get

$$\sup_{f \in \mathcal{F}} \left| \widehat{\mathcal{L}}^{\mathcal{M}}(f) - \mathcal{L}^{\mathcal{M}}(f) \right| \lesssim \mathrm{L}_{\mathsf{aug}} U' \sqrt{\frac{m \log d}{n}}. \tag{204}$$

We notice that under the event $\mathcal{E}_\mu$, $\mathcal{M}$ holds for all $\boldsymbol{x}_i$, $i \in [n+1, 2n]$, thus $\widehat{\mathcal{L}}^{\mathcal{M}}(f) = \widehat{\mathcal{L}}(f)$.

For the other term, we have

$$\begin{aligned}
\mathcal{L}^{\mathcal{M}}(f) - \mathcal{L}(f) &= \mathbb{E}\big[\ell(f(\boldsymbol{x}), y)\big] - \mathbb{E}\big[\ell^{\mathcal{M}}(f(\boldsymbol{x}), y; \boldsymbol{x})\big] = \mathbb{E}\big[\ell(f(\boldsymbol{x}), y)\mathbf{1}\{\boldsymbol{x} \notin \mathcal{M}\}\big] \\
&\leq \mathrm{L}\mathbb{E}\big[(1 + |f(\boldsymbol{x})| + |f^*(\boldsymbol{x})|)\,|f(\boldsymbol{x}) - f^*(\boldsymbol{x})|\,\mathbf{1}\{\boldsymbol{x} \notin \mathcal{M}\}\big] \\
&\lesssim \mathrm{L}\Big(\mathbb{E}\big[|f(\boldsymbol{x})|^2 \mathbf{1}\{\boldsymbol{x} \notin \mathcal{M}\}\big] + \mathbb{E}\big[|f^*(\boldsymbol{x})|^2 \mathbf{1}\{\boldsymbol{x} \notin \mathcal{M}\}\big]\Big) \\
&\lesssim \mathrm{L}\Big(\sqrt{\mathbb{E}\big[f(\boldsymbol{x})^4 \mathbb{P}\{\boldsymbol{x} \notin \mathcal{M}\}\big]} + \sqrt{\mathbb{E}\big[(f^*(\boldsymbol{x}))^4 \mathbb{P}\{\boldsymbol{x} \notin \mathcal{M}\}\big]}\Big). 
\end{aligned} \tag{205}$$

Recall that it has been assumed that the activation function has bounded third derivatives as in Assumption 2. Therefore, there exists an absolute constant $C_\sigma > 0$ such that

$$|\sigma(u)| \leq C_\sigma(1 + |u|^3) \qquad \text{for all } u \in \mathbb{R}. \tag{206}$$

Via a crude moment estimation bound (plus some hypercontractivity arguments), we can see the right side is $\lesssim \mathrm{L}U'd^{-\frac{D}{8}}$. Thus, we have the following final estimation

$$\sup_{f \in \mathcal{F}} \left| \widehat{\mathcal{L}}(f) - \mathcal{L}(f) \right| \lesssim \mathrm{L}_{\mathsf{aug}} U' \sqrt{\frac{m \log d}{n}} + \mathrm{L}U'd^{-D/8}. \tag{207}$$

Since we have assume $n \leq d^{D/4}$, we have $\sup_{f \in \mathcal{F}} \left| \widehat{\mathcal{L}}(f) - \mathcal{L}(f) \right| \lesssim \mathrm{L}_{\mathsf{aug}} U' \sqrt{\frac{m \log d}{n}} + \mathrm{L}U'd^{-\frac{D}{8}}$. Plug in the formula for $U$ and we get our desired result. $\qquad\square$

We then have the final proposition.

**Proposition D.33.** *On the event $\mathcal{E}$ and the sample-size condition $n \gtrsim \mathrm{T}_1^2 d \log d \kappa^{2\mathrm{T}_1} + d^{1+1/\mathrm{T}_1}\kappa^2$, also $\mathrm{T}_1 = o(\log d)$ the excess population risk satisfies*

$$\mathcal{L}(f_{\boldsymbol{\Theta}^{(\mathrm{T})}}) - \mathcal{L}(f^*) \lesssim \mathrm{L}_{\mathsf{aug}} \kappa^{2p\mathrm{T}_1}(\log d)^{2p+1}\left(\sqrt{\frac{1}{n}} + \sqrt{\frac{1}{m}}\right). \tag{208}$$

**Roadmap to remove $\beta = 0$ assumption.** We note that $\beta = 0$ assumption is not essential and can be removed via a careful analysis. To be concrete, one can keep the $\Delta_k$ and $V_{p,\epsilon}$ terms in final generalization bound under the weakened assumption 3'. Next, we perform another round of balancing those terms. The minimizer shall be slightly different and a bit technical troublesome, but the final result remains the same up to logarithmic factors.

## D.5 PROOF OF SUPPORTING LEMMAS IN APPENDIX D.3

### D.5.1 PROOF OF LEMMA D.1

*Proof.* We split the expectation over $\boldsymbol{w}$ into the good event $\widetilde{\mathcal{E}_\beta}$ and its complement

$$\mathbb{E}[\|\boldsymbol{\Pi}^*\boldsymbol{r}\|^j] = \mathbb{E}\left[\|\boldsymbol{\Pi}^*\boldsymbol{r}\|^j \mathbf{1}_{\widetilde{\mathcal{E}_\beta}}\right] + \mathbb{E}\left[\|\boldsymbol{\Pi}^*\boldsymbol{r}\|^j \mathbf{1}_{\neg\widetilde{\mathcal{E}_\beta}}\right]. \tag{209}$$

We first deal with the first term.

By the high-probability bound, we have $\|\mathbf{\Pi}^* \boldsymbol{r}(\boldsymbol{w})\| \leq K$ for all $\boldsymbol{w}$, so

$$\mathbb{E}\left[\|\mathbf{\Pi}^* \boldsymbol{r}\|^j \mathbf{1}_{\widetilde{\mathcal{E}_\beta}}\right] \leq K^j \mathbb{P}(\widetilde{\mathcal{E}_\beta}) \leq K^j. \tag{210}$$

Next for the second one, we only need to invoke the trivial operator norm bound. By definition we have $\left\|\widehat{\mathbf{\Sigma}}_\ell^{\mathrm{T}_1}\right\| \leq (1+\gamma)^{\mathrm{T}_1}$. Thus

$$\|\boldsymbol{r}(\boldsymbol{w})\| \leq \|\widehat{\mathbf{\Sigma}}_\ell^{\mathrm{T}_1}\| + \|\mathbf{\Sigma}_\ell\|^{\mathrm{T}_1} \leq 2M \tag{211}$$

Hence

$$\mathbb{E}\left[\|\mathbf{\Pi}^* \boldsymbol{r}\|^j \mathbf{1}_{\neg \widetilde{\mathcal{E}_\beta}}\right] \leq (2M)^j \mathbb{P}(\neg \widetilde{\mathcal{E}_\beta}) \lesssim (2M)^j r \mathrm{T}_1 d^{-D} \tag{212}$$

Therefore we have

$$\mathbb{E}\|\mathbf{\Pi}^* \boldsymbol{r}\|^j \lesssim K^j + (2M)^j r \mathrm{T}_1 d^{-D} \quad \text{and} \quad \left[\mathbb{E}\|\mathbf{\Pi}^* \boldsymbol{r}\|^j\right]^{1/j} \lesssim K \left[1 + \left(\frac{2M}{K}\right)^j r \mathrm{T}_1 d^{-D}\right]^{1/j} \tag{213}$$

To bound the right side, further note

$$\frac{2M}{K} = \frac{2(1+\gamma)^{\mathrm{T}_1}}{\mathrm{T}_1 \gamma \sqrt{2rD \log d / d}} = \frac{2(1+\gamma)^{\mathrm{T}_1}}{\mathrm{T}_1 \gamma \sqrt{2rD \log d / d}} \quad \gamma \leq 1/(2\mathrm{T}_1) \tag{214}$$

Since $(1+\gamma)^{\mathrm{T}_1} \leq e^{\mathrm{T}_1 \gamma} \leq \sqrt{e}$, we get

$$\frac{2M}{K} \lesssim \frac{1}{\mathrm{T}_1 \gamma} \sqrt{\frac{d}{rD \log d}} \lesssim \frac{1}{\mathrm{T}_1 C_H} \sqrt{\frac{n}{rD \log d}} \lesssim \sqrt{n} \tag{215}$$

Thus via choosing a large enough universal constant $D$, we have for all $j \leq 4p$

$$\left(\frac{2M}{K}\right)^j r \mathrm{T}_1 d^{-D} \lesssim n^{2p} d^{-D} \lesssim 1 \tag{216}$$

and we finally obtain the desired conclusion via combining two terms. $\square$

### D.5.2 PROOF OF LEMMA D.2

*Proof.* We want to ensure $\frac{d}{\lambda_r^{2\mathrm{T}_1}} K^2 \leq c^2$. Taking square roots on both sides, we note that this is equivalent to

$$\frac{\sqrt{d}}{\lambda_r^{\mathrm{T}_1}} K \leq c \quad \text{and} \quad K \leq c \frac{\lambda_r^{\mathrm{T}_1}}{\sqrt{d}}. \tag{217}$$

To get a sufficient condition, we bound the two terms in $K$ separately. Recall that we set

$$K = C \left(\mathrm{T}_1 \gamma \sqrt{\frac{rD \log d}{d}} + \gamma^{\mathrm{T}_1}\right) \tag{218}$$

therefore we need

$$C\mathrm{T}_1 \gamma \sqrt{\frac{rD \log d}{d}} \leq c \frac{\lambda_r^{\mathrm{T}_1}}{2\sqrt{d}} \quad \text{and} \quad C\gamma^{\mathrm{T}_1} \leq c \frac{\lambda_r^{\mathrm{T}_1}}{2\sqrt{d}}. \tag{219}$$

Under these, we get $K \leq c \frac{\lambda_r^{\mathrm{T}_1}}{\sqrt{d}}$ as desired. Note that

$$\gamma = C_H \sqrt{\frac{d}{n}} \quad \text{and} \quad CC_H \mathrm{T}_1 \sqrt{\frac{d}{n}} \sqrt{\frac{rD \log d}{d}} \lesssim \mathrm{T}_1 \sqrt{\frac{rD \log d}{n}} \tag{220}$$

Then, to satisfy the first one, we require $\mathrm{T}_1 \sqrt{\frac{rD \log d}{n}} \lesssim \frac{\lambda_r^{\mathrm{T}_1}}{\sqrt{d}}$, which is equivalent to

$$n \gtrsim \frac{\mathrm{T}_1^2 rDd \log d}{\lambda_r^{2\mathrm{T}_1}} = \mathrm{T}_1^2 rDd \log d \, \kappa^{2\mathrm{T}_1} \tag{221}$$

For the second one, we require $\sqrt{\frac{d}{n}} \lesssim \left(\frac{\lambda_r^{\mathrm{T}_1}}{\sqrt{d}}\right)^{1/\mathrm{T}_1} = \frac{\lambda_r}{(\sqrt{d})^{1/\mathrm{T}_1}}$ which is equivalent to

$$n \gtrsim \frac{d^{1+1/\mathrm{T}_1}}{\lambda_r^2} = d^{1+1/\mathrm{T}_1} \kappa^2 \tag{222}$$

Combining the two parts above, the final conclusion directly follows. $\square$

### D.5.3   PROOF OF LEMMA D.3

*Proof.* From Lemma D.2 we know that $K \lesssim \frac{\lambda_r^{T_1}}{\sqrt{d}}$. Then on the intersection of event $\mathcal{E}_\alpha \cap \widetilde{\mathcal{E}_\beta}$ the residual satisfies

$$\|r(w)\| \lesssim K \lesssim \frac{\lambda_r^{T_1}}{\sqrt{d}} \leq \sqrt{\log d/d}. \tag{223}$$

Moreover $h_n(w) = h(w) + r(w)$ and $h(w) = \Sigma_\ell^{T_1} w$. Under $\widetilde{\mathcal{E}_\beta}$, we have $\|h(w)\| \lesssim \sqrt{\log d/d}$. Therefore, under the event $\mathcal{E}_\alpha \cap \widetilde{\mathcal{E}_\beta}$ we have

$$\|h_n(w)\| \leq \|h(w)\| + \|r(w)\| \lesssim \sqrt{\log d/d}. \tag{224}$$

$\square$

### D.5.4   PROOF OF LEMMA D.5

*Proof.* Note that since every vector in $\mathrm{span}(\mathrm{Mat}(\mathbb{E}_w[(\Pi^\star h_n(w))^{\otimes 2k}]))$ is a vectorized symmetric $k$ tensor, it suffices to show that

$$\mathbb{E}[(\Pi^\star h_n(w))^{\otimes 2k}](T, T) \gtrsim \left(\frac{d}{\lambda_r^{2T_1}}\right)^{-k} \tag{225}$$

for all symmetric $k$ tensors $T$ supported on $S^*$ with $\|T\|_F^2 = 1$.

For notation convenience, we denote $w_\epsilon := w/\epsilon_0$ where we know $w_\epsilon \sim \mathrm{Unif}(\mathbb{S}^{d-1})$. Recalling that $\Pi^\star h_n(w) = \Sigma_\ell^{T_1} w_\epsilon + \Pi^\star r(w)$ and the tensor binomial expansion, we have

$$(\Pi^\star h_n(w))^{\otimes k} = \sum_{i=0}^k \binom{k}{i} \left(\Sigma_\ell^{T_1} w_\epsilon\right)^{\otimes(k-i)} \otimes (\Pi^\star r(w))^{\otimes i} \tag{226}$$

Thus for any symmetric $k$-tensor $T$, we have

$$\left\langle T, (\Pi^\star h_n(w))^{\otimes k}\right\rangle = \left\langle T, \left(\Sigma_\ell^{T_1} w_\epsilon\right)^{\otimes k}\right\rangle + \delta(w) \tag{227}$$

where

$$\delta(w) := \sum_{i=1}^k \binom{k}{i} \langle T, \left(\Sigma_\ell^{T_1} w_\epsilon\right)^{\otimes(k-i)} \otimes (\Pi^\star r(w))^{\otimes i}\rangle \tag{228}$$

Note that by Young's inequality,

$$\mathbb{E}\left[\left\langle T, (\Pi^\star h_n(w))^{\otimes k}\right\rangle^2\right] \geq \mathbb{E}\left[\frac{1}{2}\left\langle T, \left(\Sigma_\ell^{T_1} w_\epsilon\right)^{\otimes k}\right\rangle^2\right] - \mathbb{E}\left[\delta(w)^2\right] \tag{229}$$

Therefore we need to upper bound $\mathbb{E}[\delta(w)^2]$ and lower bound $\mathbb{E}[\frac{1}{2}\langle T, (\Sigma_\ell^{T_1} w_\epsilon)^{\otimes k}\rangle^2]$. Intuitively the $\delta(w)$ term should be negligible since $r(w)$ is of small norm.

Note that there is a crude upper bound for $\delta(w)$. Contracting $T$ in its first $k - i$ slots and applying the Frobenius Cauchy-Schwarz inequality gives

$$\left|\langle T, \left(\Sigma_\ell^{T_1} w_\epsilon\right)^{\otimes(k-i)} \otimes (\Pi^\star r(w))^{\otimes i}\rangle\right| \leq \left\|T\left(\left(\Sigma_\ell^{T_1} w_\epsilon\right)^{\otimes(k-i)}\right)\right\|_F \|\Pi^\star r(w)\|^i \tag{230}$$

Hence

$$|\delta(w)| \leq \sum_{i=1}^k \binom{k}{i} \left\|T\left(\left(\Sigma_\ell^{T_1} w_\epsilon\right)^{\otimes(k-i)}\right)\right\|_F \|\Pi^\star r(w)\|^i$$

$$\lesssim \sum_{i=1}^k \left\|T\left(\left(\Sigma_\ell^{T_1} w_\epsilon\right)^{\otimes(k-i)}\right)\right\|_F \|\Pi^\star r(w)\|^i. \tag{231}$$

Via Cauchy–Schwarz again, we have

$$\mathbb{E}[\delta(\boldsymbol{w})^2] \lesssim \sum_{i=1}^{k} \sqrt{\mathbb{E}[\|\boldsymbol{T}((\boldsymbol{\Sigma}_\ell^{\mathrm{T}_1}\boldsymbol{w}_\epsilon)^{\otimes(k-i)})\|_F^4]\mathbb{E}[\|\boldsymbol{\Pi}^\star\boldsymbol{r}(\boldsymbol{w})\|^{4i}]}$$

$$\lesssim \sum_{i=1}^{k} \mathbb{E}[\|\boldsymbol{T}((\boldsymbol{\Sigma}_\ell^{\mathrm{T}_1}\boldsymbol{w}_\epsilon)^{\otimes(k-i)})\|_F^2]\sqrt{\mathbb{E}[\|\boldsymbol{\Pi}^\star\boldsymbol{r}(\boldsymbol{w})\|^{4i}]} \tag{232}$$

where the second line is derived via the Gaussian hypercontractivity lemma that we will state at Lemma E.22 in Section E.6 .

Next, let $\widehat{\boldsymbol{T}}$ be the symmetric $k$ tensor defined by $\widehat{\boldsymbol{T}}(\boldsymbol{v}_1, \ldots, \boldsymbol{v}_k) = \boldsymbol{T}(\boldsymbol{\Sigma}_\ell^{\mathrm{T}_1}\boldsymbol{v}_1, \ldots, \boldsymbol{\Sigma}_\ell^{\mathrm{T}_1}\boldsymbol{v}_k)$. For each $i$ and each unit-Frobenius norm tensor $\boldsymbol{V}$ supported on $S^*$ we have

$$\langle\boldsymbol{T}((\boldsymbol{\Sigma}_\ell^{\mathrm{T}_1}\boldsymbol{w}_\epsilon)^{\otimes(k-i)}), \boldsymbol{V}\rangle = \widehat{\boldsymbol{T}}(\boldsymbol{w}_\epsilon^{\otimes(k-i)}, (\boldsymbol{\Sigma}_\ell^{\mathrm{T}_1})^\dagger\boldsymbol{V}) \tag{233}$$

i.e. by moving $\boldsymbol{\Sigma}_\ell^{\mathrm{T}_1}$ from the arguments of $\boldsymbol{T}$ onto the test tensor one applies $(\boldsymbol{\Sigma}_\ell^{\mathrm{T}_1})^\dagger$ to each slot of $\boldsymbol{V}$. By Cauchy-Schwarz again,

$$\left|\boldsymbol{T}\left(\left(\boldsymbol{\Sigma}_\ell^{\mathrm{T}_1}(\boldsymbol{w}/\epsilon_0)\right)^{\otimes(k-i)}\right), \boldsymbol{V}\right| \leq \|\widehat{\boldsymbol{T}}(\boldsymbol{w}_\epsilon^{\otimes(k-i)})\|_F \|(\boldsymbol{\Sigma}_\ell^{\mathrm{T}_1})^\dagger\boldsymbol{V}\|_F. \tag{234}$$

Since $\|(\boldsymbol{\Sigma}_\ell^{\mathrm{T}_1})^\dagger\boldsymbol{V}\|_F \leq 1/\lambda_r^{i\mathrm{T}_1}$ for $\|\boldsymbol{V}\|_F = 1$, taking the supremum over unit $V$ gives the pointwise bound

$$\|\boldsymbol{T}\left(\left(\boldsymbol{\Sigma}_\ell^{\mathrm{T}_1}\boldsymbol{w}_\epsilon\right)^{\otimes(k-i)}\right)\|_F \leq 1/\lambda_r^{i\mathrm{T}_1}\|\widehat{\boldsymbol{T}}(\boldsymbol{w}_\epsilon^{\otimes(k-i)})\|_F \tag{235}$$

Squaring and taking expectation over $\boldsymbol{w}$ yields

$$\mathbb{E}[\|\boldsymbol{T}((\boldsymbol{\Sigma}_\ell^{\mathrm{T}_1}\boldsymbol{w}_\epsilon)^{\otimes(k-i)})\|_F^2] \leq \frac{1}{\lambda_r^{2i\mathrm{T}_1}}\mathbb{E}[\|\widehat{\boldsymbol{T}}(\boldsymbol{w}_\epsilon^{\otimes(k-i)})\|_F^2] \tag{236}$$

Combining with previous calculation, we have the estimation below

$$\mathbb{E}[\|\boldsymbol{T}((\boldsymbol{\Sigma}_\ell^{\mathrm{T}_1}\boldsymbol{w}_\epsilon)^{\otimes(k-i)})\|_F^2] \leq \frac{1}{\lambda_r^{2i\mathrm{T}_1}}\mathbb{E}[\|\widehat{\boldsymbol{T}}(\boldsymbol{w}_\epsilon^{\otimes(k-i)})\|^2]$$

$$\lesssim \left(\frac{d}{\lambda_r^{2\mathrm{T}_1}}\right)^i \mathbb{E}[\langle\boldsymbol{T}, (\boldsymbol{\Sigma}_\ell^{\mathrm{T}_1}\boldsymbol{w}_\epsilon)^{\otimes k}\rangle^2] \tag{237}$$

where the second inequality is deduced from the lemma below.

**Lemma D.34.** *Let $\boldsymbol{w} \sim \mathrm{Unif}(\mathbb{S}^{d-1})$ and for each tensor $\boldsymbol{T}$ and each $k \leq p$ we have*

$$\mathbb{E}\|\boldsymbol{T}(\boldsymbol{w}^{\otimes k})\|_F^2 \lesssim d^{p-k}\mathbb{E}\langle\boldsymbol{T}, \boldsymbol{w}^{\otimes p}\rangle^2. \tag{238}$$

The proof of Lemma D.34 is provided in Appendix E.5

We also know from Lemma D.1 that for any integer $1 \leq j \leq 4p$ we have under the high probability event of $\mathcal{E}_\alpha$, which happens with probability at least $1 - \mathcal{O}(d^{-D})$ that

$$\left[\mathbb{E}\|\boldsymbol{\Pi}^*\boldsymbol{r}(\boldsymbol{w})\|^j\right]^{1/j} \lesssim K \tag{239}$$

Therefore, we have

$$\mathbb{E}[\delta(\boldsymbol{w})^2] \lesssim \mathbb{E}[\langle\boldsymbol{T}, (\boldsymbol{\Sigma}_\ell^{\mathrm{T}_1}\boldsymbol{w}_\epsilon)^{\otimes k}\rangle^2] \sum_{i=1}^{k} \left(\frac{d}{\lambda_r^{2\mathrm{T}_1}}K^2\right)^i \tag{240}$$

Because we have assumed $n \gtrsim rDd\log d\kappa^{2\mathrm{T}_1} + d^{1+1/\mathrm{T}_1}\kappa^2$, we have the following from Lemma D.2 that

$$\frac{d}{\lambda_r^{2\mathrm{T}_1}}K^2 \leq c^2 \tag{241}$$

with some prechosen absolute constant $c$. Thus via picking a small $c$ and plugging the above bound into the previous equation we have

$$\mathbb{E}_{\boldsymbol{w}}[\delta(\boldsymbol{w})^2] \leq \frac{1}{4}\mathbb{E}[\langle\boldsymbol{T}, (\boldsymbol{\Sigma}_\ell^{\mathrm{T}_1}\boldsymbol{w}_\epsilon)^{\otimes k}\rangle^2]. \tag{242}$$

Combining everything gives

$$\mathbb{E}[\langle \boldsymbol{T}, (\boldsymbol{\Pi}^{\star}\boldsymbol{h}_n(\boldsymbol{w}))^{\otimes k}\rangle^2] \geq \mathbb{E}[\frac{1}{2}\langle \boldsymbol{T}, (\boldsymbol{\Sigma}_\ell^{T_1}\boldsymbol{w}_\epsilon)^{\otimes k}\rangle^2] - \mathbb{E}_{\boldsymbol{w}}[\delta(\boldsymbol{w})^2]$$

$$\geq \frac{1}{4}\mathbb{E}[\langle \boldsymbol{T}, (\boldsymbol{\Sigma}_\ell^{T_1}\boldsymbol{w}_\epsilon)^{\otimes k}\rangle^2] \tag{243}$$

$$\gtrsim d^{-k}\|\widehat{\boldsymbol{T}}\|_F^2$$

$$\geq \left(\frac{d}{\lambda_r^{2T_1}}\right)^{-k} \tag{244}$$

where the third line is via taking the expectation over $\boldsymbol{w}$, the detailed proof for this can be also found in Appendix E.5. The proof is finished. □

### D.5.5 PROOF OF LEMMA D.6

*Proof.* Let
$$z_{\boldsymbol{T}}(\boldsymbol{w}) := \mathrm{Vec}(\boldsymbol{T})^{\mathsf{T}}\mathrm{Mat}(\mathbb{E}[\boldsymbol{h}_n(\boldsymbol{w})^{\otimes 2k}])^{\dagger}\mathrm{Vec}(\boldsymbol{h}_n(\boldsymbol{w})^{\otimes k}). \tag{245}$$

Note that $\mathrm{Vec}(\boldsymbol{T}) \in \mathrm{span}(\mathrm{Mat}(\mathbb{E}[\boldsymbol{h}_n(\boldsymbol{w})^{\otimes 2k}]))$ by Lemma D.5. Therefore,

$$\mathbb{E}_{\boldsymbol{w}}[z_{\boldsymbol{T}}(\boldsymbol{w})(\boldsymbol{h}_n(\boldsymbol{w})^{\mathsf{T}}\boldsymbol{x})^k] = \mathbb{E}_{\boldsymbol{w}}[\mathrm{Vec}(\boldsymbol{T})^{\mathsf{T}}\mathrm{Mat}(\mathbb{E}[\boldsymbol{h}_n(\boldsymbol{w})^{\otimes 2k}])^{\dagger}\mathrm{Vec}(\boldsymbol{h}_n(\boldsymbol{w})^{\otimes k})\mathrm{Vec}(\boldsymbol{h}_n(\boldsymbol{w})^{\otimes k})^{\mathsf{T}}\mathrm{Vec}(\boldsymbol{x}^{\otimes k})]$$

$$= \langle \boldsymbol{T}, \boldsymbol{x}^{\otimes k}\rangle. \tag{246}$$

For the bounds on $z_{\boldsymbol{T}}(\boldsymbol{w})$ we have the following estimation

$$\mathbb{E}_{\boldsymbol{w}}[z_{\boldsymbol{T}}(\boldsymbol{w})^2] = \mathbb{E}_{\boldsymbol{w}}[\mathrm{Vec}(\boldsymbol{T})^{\mathsf{T}}\mathrm{Mat}(\mathbb{E}[\boldsymbol{h}_n(\boldsymbol{w})^{\otimes 2k}])^{\dagger}\mathrm{Vec}(\boldsymbol{h}_n(\boldsymbol{w})^{\otimes k})\mathrm{Vec}(\boldsymbol{h}_n(\boldsymbol{w})^{\otimes k})^{\mathsf{T}}\mathrm{Mat}(\mathbb{E}[\boldsymbol{h}_n(\boldsymbol{w})^{\otimes 2k}])^{\dagger}\mathrm{Vec}(\boldsymbol{T})]$$

$$= \mathrm{Vec}(\boldsymbol{T})^{\mathsf{T}}\mathrm{Mat}(\mathbb{E}[\boldsymbol{h}_n(\boldsymbol{w})^{\otimes 2k}])^{\dagger}\mathrm{Vec}(\boldsymbol{T})$$

$$\lesssim \left(\frac{d}{\lambda_r^{2T_1}}\right)^k \|\boldsymbol{T}\|_F^2, \tag{247}$$

and

$$|z_{\boldsymbol{T}}(\boldsymbol{w})| = \left|\mathrm{Vec}(\boldsymbol{T})^{\mathsf{T}}\mathrm{Mat}(\mathbb{E}[\boldsymbol{h}_n(\boldsymbol{w})^{\otimes 2k}])^{\dagger}\mathrm{Vec}(\boldsymbol{h}_n(\boldsymbol{w})^{\otimes k})\right| \lesssim \left(\frac{d}{\lambda_r^{2T_1}}\right)^k \|\boldsymbol{T}\|_F\|\boldsymbol{h}_n(\boldsymbol{w})\|^k. \tag{248}$$

The proof is complete. □

### D.5.6 PROOF OF LEMMA D.9

*Proof.* Recall that from Lemma D.3 and D.4, under the high probability event of $\mathcal{E}_\alpha \cap \widetilde{\mathcal{E}_\beta} \cap \widetilde{\mathcal{E}_\eta}$, which happens with probability at least $1 - \mathcal{O}(d^{-D})$ the following bounds hold

$$\max_{j\in[n+1,2n]} \|\boldsymbol{h}_n(\boldsymbol{w})^{\mathsf{T}}\boldsymbol{x}_j\| \lesssim \sqrt{\frac{\log^2 d}{d}} \quad \text{and} \quad \|\boldsymbol{h}_n(\boldsymbol{w})\| \lesssim \sqrt{\frac{\log d}{d}}. \tag{249}$$

Those bounds will be helpful for us to construct the low-dimensional approximation in the second stage.

Recall that we have set $\eta = \eta_1 = \frac{1}{C}\left(\frac{d}{\log^2 d}\right)^{1/(2T_1)}$ where the universal constant $C$ is chosen sufficiently large so we can absorb the implicit constants hidden in the $\lesssim$ bounds. This means we choose $C$ to make sure on the same high-probability event the following controls are valid

$$\eta^{T_1}\|\boldsymbol{h}_n(\boldsymbol{w})\| \leq 1 \quad \text{and} \quad \max_{j\in[n+1,2n]} |\eta^{T_1}\boldsymbol{h}_n(\boldsymbol{w})^{\mathsf{T}}\boldsymbol{x}_j| \leq 1 \tag{250}$$

which indicates that

$$\|\widetilde{\boldsymbol{w}}^{(T_1)}\| = \left\|\frac{1}{\epsilon_0}(-a_j)^{T_1}\eta^{T_1}\widehat{\boldsymbol{\Sigma}}_\ell^{T_1}\boldsymbol{w}\right\| \leq 1 \tag{251}$$

Note that $\boldsymbol{w}^{(T_1)} = \frac{1}{\epsilon_0}(-a_j)^{T_1}\eta^{T_1}\left(\prod_{t=0}^{T_1-1}\boldsymbol{M}^{(t)}\right)\boldsymbol{w}$ and by Lemma C.1, we have

$$\|\boldsymbol{w}^{(T_1)} - \widehat{\boldsymbol{w}}^{(T_1)}\| \lesssim K \lesssim \eta^{T_1}\sqrt{\frac{\log d}{d}} \tag{252}$$

Thus $\|\boldsymbol{w}^{(T_1)}\| \leq 2\|\widetilde{\boldsymbol{w}}^{(T_1)}\| \lesssim 1$ and we complete our proof.

$\square$

### D.5.7 PROOF OF LEMMA D.10

*Proof.* For any training point $\boldsymbol{x}$ and given initialization $\boldsymbol{w}$, on events $\mathcal{E}_\alpha \cap \mathcal{E}_\zeta \cap \widetilde{\mathcal{E}_\beta} \cap \widetilde{\mathcal{E}_\eta}$ we have the following control

$$z := (-\eta)^{T_1}\boldsymbol{h}_n(\boldsymbol{w})^\mathsf{T}\boldsymbol{x} \in [-1,1]. \tag{253}$$

By our previous assumption on the activation function, we have the following low dimensional approximation result uniformly for $z \in [-1,1]$

$$\left|\mathbb{E}_{a,b}[\boldsymbol{v}_k^{(\epsilon_k)}(a,b)\sigma(az+b)] - z^k\right| \leq \epsilon_k. \tag{254}$$

Divide both sides by $(-\eta)^{kT_1}$ and substitute the features back we obtain

$$\left|\frac{1}{(-\eta)^{kT_1}}\mathbb{E}_{a,b}[\boldsymbol{v}_k^{(\epsilon_k)}(a,b)\sigma((-\eta a)^{T_1}\boldsymbol{h}_n(\boldsymbol{w})^\mathsf{T}\boldsymbol{x}+b)] - (\boldsymbol{h}_n(\boldsymbol{w})^\mathsf{T}\boldsymbol{x})^k\right| \leq \frac{\epsilon_k}{\eta^{kT_1}}. \tag{255}$$

Recall that from Corollary D.6 under the high probability event $\mathcal{E}_\alpha$ we have

$$\mathbb{E}_{\boldsymbol{w}}[z_{\boldsymbol{T}}(\boldsymbol{w})(\boldsymbol{h}_n(\boldsymbol{w})^\mathsf{T}\boldsymbol{x})^k] = \langle \boldsymbol{T}, \boldsymbol{x}^{\otimes k}\rangle, \tag{256}$$

And also recall from Lemma D.4 and the statement before the lemma we know that under the high probability event $\mathcal{E}_\alpha \cap \mathcal{E}_\zeta \cap \widetilde{\mathcal{E}_\beta} \cap \widetilde{\mathcal{E}_\eta}$

$$|\eta^{T_1}\boldsymbol{h}_n(\boldsymbol{w})^\mathsf{T}\boldsymbol{x}_{i+n}| \leq 1 \tag{257}$$

$$\eta^{T_1}\|\boldsymbol{h}_n(\boldsymbol{w})\| \leq 1 \tag{258}$$

thus

$$\begin{aligned}
&\left|f_{h_{\boldsymbol{T}}}(\boldsymbol{x}) - \langle \boldsymbol{T}, \boldsymbol{x}^{\otimes k}\rangle\right| \\
&\leq \frac{\epsilon_k}{\eta^{kT_1}}\left|\mathbb{E}_{\boldsymbol{w}}[z_{\boldsymbol{T}}(\boldsymbol{w})\mathbf{1}_{\mathcal{E}_\alpha\cap\mathcal{E}_\zeta\cap\widetilde{\mathcal{E}_\beta}\cap\widetilde{\mathcal{E}_\eta}}]\right| + \frac{1}{\eta^{kT_1}}\left|\mathbb{E}_{\boldsymbol{w}}[z_{\boldsymbol{T}}(\boldsymbol{w})\mathbf{1}_{\mathcal{E}_\alpha\cap\mathcal{E}_\zeta\cap(\widetilde{\mathcal{E}_\beta}\cap\widetilde{\mathcal{E}_\eta})^c}]\right| \\
&\quad + \frac{1}{\eta^{kT_1}}\left|\mathbb{E}_{\boldsymbol{w}}[z_{\boldsymbol{T}}(\boldsymbol{w})\boldsymbol{v}_k^{(\epsilon_k)}(a,b)\sigma((-\eta a)^{T_1}\boldsymbol{h}_n(\boldsymbol{w})^\mathsf{T}\boldsymbol{x}+b)\mathbf{1}_{\mathcal{E}_\alpha\cap\mathcal{E}_\zeta\cap(\widetilde{\mathcal{E}_\beta}\cap\widetilde{\mathcal{E}_\eta})^c}]\right| \\
&\leq \frac{\epsilon_k}{\eta^{kT_1}}\left|\mathbb{E}_{\boldsymbol{w}}[z_{\boldsymbol{T}}(\boldsymbol{w})]\right| + \frac{1}{\eta^{kT_1}}\left|\mathbb{E}_{\boldsymbol{w}}[z_{\boldsymbol{T}}(\boldsymbol{w})\mathbf{1}_{\mathcal{E}_\alpha\cap\mathcal{E}_\zeta\cap(\widetilde{\mathcal{E}_\beta}\cap\widetilde{\mathcal{E}_\eta})^c}]\right| \\
&\quad + \eta^{-kT_1}\left|\mathbb{E}_{\boldsymbol{w}}[z_{\boldsymbol{T}}(\boldsymbol{w})\boldsymbol{v}_k^{(\epsilon_k)}(a,b)\sigma((-\eta a)^{T_1}\boldsymbol{h}_n(\boldsymbol{w})^\mathsf{T}\boldsymbol{x}+b)\mathbf{1}_{\mathcal{E}_\alpha\cap\mathcal{E}_\zeta\cap(\widetilde{\mathcal{E}_\beta}\cap\widetilde{\mathcal{E}_\eta})^c}]\right|
\end{aligned} \tag{259}$$

We bound the third term first. Denote

$$X(a,b,\boldsymbol{w}) := \frac{1}{\eta^{kT_1}}\boldsymbol{v}_k^{(\epsilon_k)}(a,b)z_{\boldsymbol{T}}(\boldsymbol{w})\sigma((-\eta a)^{T_1}\boldsymbol{h}_n(\boldsymbol{w})^\mathsf{T}\boldsymbol{x}+b). \tag{260}$$

By Cauchy-Schwarz,

$$\left|\mathbb{E}_{a,b,\boldsymbol{w}}[X(a,b,\boldsymbol{w})\mathbf{1}_{\mathcal{E}_\alpha\cap\mathcal{E}_\zeta\cap(\widetilde{\mathcal{E}_\beta}\cap\widetilde{\mathcal{E}_\eta})^c}]\right| \leq \sqrt{\mathbb{E}_{a,b,\boldsymbol{w}}[X(a,b,\boldsymbol{w})^2]}\sqrt{\mathbb{P}(\mathcal{E}_\alpha \cap \mathcal{E}_\zeta \cap (\widetilde{\mathcal{E}_\beta}\cap\widetilde{\mathcal{E}_\eta})^c)} \tag{261}$$

We bound the two factors on the right. Recall that it has been assumed that the activation function has bounded third derivatives as in Assumption 2. Therefore, there exists a constant $C_\sigma > 0$ such that

$$|\sigma(u)| \leq C_\sigma(1+|u|^3) \quad \text{for all } u \in \mathbb{R}. \tag{262}$$

and also
$$|X(a, b, \boldsymbol{w})| \leq V_{k,\epsilon_k} |z_{\boldsymbol{T}}(\boldsymbol{w})| C_\sigma (1 + |\eta^{\mathrm{T}_1} \boldsymbol{h}_n(\boldsymbol{w})^\mathsf{T} \boldsymbol{x} + b|^3). \tag{263}$$
We recall that the following crude bound works for all $\boldsymbol{w}$
$$\|\boldsymbol{h}_n(\boldsymbol{w})\| \leq (1 + \gamma)^{\mathrm{T}_1} \leq 2^{\mathrm{T}_1} \tag{264}$$
where $\gamma$ is defined in via directly bounding the operator norm of $\prod_{t=0}^{\mathrm{T}_1-1} \boldsymbol{M}^{(t)}$. Notice that under the event $\mathcal{E}_\zeta$, we have $\|\boldsymbol{x}\| \leq 2\sqrt{d}$.

Thus via another crude bound we have $|\eta^{\mathrm{T}_1} \boldsymbol{h}_n(\boldsymbol{w})^\mathsf{T} \boldsymbol{x} + b| \lesssim d$ and
$$|X| \lesssim V_{k,\epsilon_k} |z_{\boldsymbol{T}}(\boldsymbol{w})| d^3. \tag{265}$$
In addition, from Definition D.7 we also have $V_{k,\epsilon_k} = \|\boldsymbol{v}_k^{(\epsilon_k)}\|_\infty \lesssim \epsilon_k^{-\beta} \lesssim d^{D/12}$. Also invoking $\mathbb{E}_{\boldsymbol{w}}[z_{\boldsymbol{T}}^2] \lesssim d^k \kappa^{2\mathrm{T}_1 k} \|\boldsymbol{T}\|_F^2$ from D.6, note that for $\mathrm{T}_1 = o(\log d)$, we have $\kappa^{\mathrm{T}_1} \lesssim d$, thus it shows that we obtain
$$\sqrt{\mathbb{E}[X(a, b, \boldsymbol{w})^2]} \leq V_{k,\epsilon_k} d^3 \sqrt{\mathbb{E}[z_{\boldsymbol{T}}^2]} \lesssim d^{D/12 + 3k/2 + 3} \|\boldsymbol{T}\|_F. \tag{266}$$
Second, Lemma D.3 and D.4 give
$$\mathbb{P}(\mathcal{E}_\alpha \cap \mathcal{E}_\zeta \cap (\widetilde{\mathcal{E}_\beta} \cap \widetilde{\mathcal{E}_\eta})^c) \leq \mathbb{P}((\widetilde{\mathcal{E}_\beta} \cap \widetilde{\mathcal{E}_\eta})^c) \lesssim d^{-D/2} \tag{267}$$
when $\iota = 2D \log d$. Hence
$$\left| \mathbb{E}_{a,b,\boldsymbol{w}}[X(a, b, \boldsymbol{w}) \mathbf{1}_{\mathcal{E}_\alpha \cap \mathcal{E}_\zeta \cap (\widetilde{\mathcal{E}_\beta} \cap \widetilde{\mathcal{E}_\eta})^c}] \right| \lesssim d^{-D/8} \|\boldsymbol{T}\|_F \tag{268}$$
for sufficiently large universal constant $D$.

The above argument shows that the third term is negligible.

Then we bound the second term. Using essentially the same method and argument as above, we obtain
$$\frac{1}{\eta^{k\mathrm{T}_1}} \left| \mathbb{E}_{\boldsymbol{w}}[z_{\boldsymbol{T}}(\boldsymbol{w}) \mathbf{1}_{\mathcal{E}_\alpha \cap \mathcal{E}_\zeta \cap (\widetilde{\mathcal{E}_\beta} \cap \widetilde{\mathcal{E}_\eta})^c}] \right| \lesssim \frac{1}{\eta^{k\mathrm{T}_1}} \sqrt{\mathbb{E}_{\boldsymbol{w}}[z_{\boldsymbol{T}}^2]} \sqrt{\mathbb{P}(\mathcal{E}_\alpha \cap \mathcal{E}_\zeta \cap (\widetilde{\mathcal{E}_\beta} \cap \widetilde{\mathcal{E}_\eta})^c)}$$
$$\lesssim d^{3k/2} \|\boldsymbol{T}\|_F d^{-D/4}$$
$$\lesssim d^{-D/8} \|\boldsymbol{T}\|_F \tag{269}$$

For the first one, we define the approximation error
$$\Delta_k := \frac{\epsilon_k}{\eta^{k\mathrm{T}_1}} |\mathbb{E}_{\boldsymbol{w}}[z_{\boldsymbol{T}}(\boldsymbol{w})]|. \tag{270}$$
By Cauchy–Schwarz,
$$\Delta_k \leq \frac{\epsilon_k}{\eta^{k\mathrm{T}_1}} \sqrt{\mathbb{E}_{\boldsymbol{w}}[z_{\boldsymbol{T}}(\boldsymbol{w})^2]}. \tag{271}$$
Substitute the second-moment bound mentioned in Corollary D.6 and we obtain
$$\mathbb{E}_{\boldsymbol{w}}[z_{\boldsymbol{T}}(\boldsymbol{w})^2] \lesssim \left( \frac{d}{\lambda_r^{2\mathrm{T}_1}} \right)^k \|\boldsymbol{T}\|_F^2 \tag{272}$$
thus
$$\Delta_k \lesssim \epsilon_k \frac{1}{\eta^{k\mathrm{T}_1}} \left( \frac{d}{\lambda_r^{2\mathrm{T}_1}} \right)^{k/2} \|\boldsymbol{T}\|_F \lesssim \epsilon_k \kappa^{k\mathrm{T}_1} \log^k d \|\boldsymbol{T}\|_F. \tag{273}$$

For the other moments, from Definition D.7 we have $\|\boldsymbol{v}_k^{(\epsilon_k)}\|_\infty \leq V_{k,\epsilon_k}$ and
$$\mathbb{E}[h_{\boldsymbol{T}}^2] \leq \frac{V_{k,\epsilon_k}^2}{\eta^{2k\mathrm{T}_1}} \mathbb{E}[z_{\boldsymbol{T}}(\boldsymbol{w})^2] \lesssim V_{k,\epsilon_k}^2 \frac{1}{\eta^{2k\mathrm{T}_1}} \left( \frac{d}{\lambda_r^{2\mathrm{T}_1}} \right)^k \|\boldsymbol{T}\|_F^2. \tag{274}$$
Substitute $\eta$ back and we can get
$$\mathbb{E}[h_{\boldsymbol{T}}^2] \lesssim V_{k,\epsilon_k}^2 (\kappa^{2\mathrm{T}_1} \log^2 d)^k \|\boldsymbol{T}\|_F^2. \tag{275}$$
Similarly, using $|z_{\boldsymbol{T}}(\boldsymbol{w})| \lesssim \left( \frac{d}{\lambda_r^{2\mathrm{T}_1}} \right)^k \|\boldsymbol{T}\|_F \|\boldsymbol{h}_n(\boldsymbol{w})\|^k$ again and noticing the truncation terms in the definition of $h_{\boldsymbol{T}}(a, b, \boldsymbol{w})$, we have
$$\|h_{\boldsymbol{T}}\|_\infty \lesssim V_{k,\epsilon_k} (\kappa^{2\mathrm{T}_1} \log^2 d)^k \|\boldsymbol{T}\|_F \tag{276}$$
and the lemma is proved. $\qquad\square$

### D.5.8 PROOF OF LEMMA D.13

*Proof.* Since $h(a, b, \boldsymbol{w}) := \sum_{k \leq p} h_{\boldsymbol{T}_k}(a, b, \boldsymbol{w})$, combining with linearity of expectation, we have

$$f_h(\boldsymbol{x}) = \sum_{k \leq p} f_{h_{\boldsymbol{T}_k}}(\boldsymbol{x}) \tag{277}$$

For each $k$, define the empirical error vector $\boldsymbol{o}_k \in \mathbb{R}^n$ by

$$\boldsymbol{o}_k(i) := f_{h_{\boldsymbol{T}_k}}(\boldsymbol{x}_i) - \langle \boldsymbol{T}_k, \boldsymbol{x}_i^{\otimes k} \rangle, \quad i = n+1, \ldots, 2n. \tag{278}$$

Then for the aggregated function we have, for all $\boldsymbol{x}_i$ from the second stage

$$f_h(\boldsymbol{x}_i) - f^*(\boldsymbol{x}_i) = \sum_{k \leq p} \boldsymbol{o}_k(i), \tag{279}$$

For now throughout this proof we equip $\mathbb{R}^n$ with the empirical $\ell_2$ norm, defined for any vector $\boldsymbol{v} = (v(1), \ldots, v(n)) \in \mathbb{R}^n$ by

$$\|\boldsymbol{v}\|_n^2 := \frac{1}{n} \sum_{i=1}^n v(i)^2. \tag{280}$$

Hence we can write the empirical squared loss by

$$\frac{1}{n} \sum_{i=n+1}^{2n} (f_h(\boldsymbol{x}_i) - f^*(\boldsymbol{x}_i))^2 = \left\| \sum_{k \leq p} \boldsymbol{o}_k \right\|_n^2 \tag{281}$$

Note that we have

$$\left\| \sum_{k \leq p} \boldsymbol{o}_k \right\|_n^2 \leq \left( \sum_{k \leq p} \|\boldsymbol{o}_k\|_n \right)^2 \tag{282}$$

By Corollary D.11, each single-term error satisfies $\|\boldsymbol{o}_k\|_n^2 \lesssim \Delta_k^2 + d^{-D/4} \|\boldsymbol{T}_k\|_F^2$. Thus

$$\|\boldsymbol{o}_k\|_n \lesssim \Delta_k + d^{-D/8} \|\boldsymbol{T}_k\|_F. \tag{283}$$

Notice that by Lemma E.21 we have $\|\boldsymbol{T}_k\|_F \lesssim r^{(p-k)/4} \lesssim 1$. Therefore

$$\begin{aligned}
\left( \sum_{k \leq p} \|\boldsymbol{o}_k\|_n \right)^2 &\lesssim \left( \sum_{k \leq p} \Delta_k + d^{-D/8} \sum_{k \leq p} \|\boldsymbol{T}_k\|_F \right)^2 \\
&\lesssim \left( \sum_{k \leq p} \Delta_k + d^{-D/8} p \right)^2 \\
&\lesssim \left( \sum_{k \leq p} \Delta_k \right)^2 + \left( d^{-D/8} \right)^2 \\
&\lesssim \left( \sum_{k \leq p} \Delta_k \right)^2 + d^{-D/4}
\end{aligned} \tag{284}$$

where the last step absorbs $r, p$ into the $\lesssim$ constant. This proves equation (147).

For the moment bounds, we have by Cauchy–Schwarz

$$\mathbb{E}[h^2] = \mathbb{E}\left[ \left( \sum_{k \leq p} h_{\boldsymbol{T}_k} \right)^2 \right] = \sum_{k, \ell \leq p} \mathbb{E}[h_{\boldsymbol{T}_k} h_{\boldsymbol{T}_\ell}] \leq \left( \sum_{k \leq p} \sqrt{\mathbb{E}[h_{\boldsymbol{T}_k}^2]} \right)^2 \tag{285}$$

Substituting the second-moment bound mentioned in Lemma D.10, we obtain for each $k$,

$$\mathbb{E}[h_{\boldsymbol{T}_k}^2] \lesssim V_{k, \epsilon_k}^2 (\kappa^{2T_1} \log^2 d)^k \|\boldsymbol{T}_k\|_F^2 \tag{286}$$

hence

$$\sqrt{\mathbb{E}[h_{T_k}^2]} \lesssim V_{p,\epsilon}(\kappa^{T_1}\log d)^k \|T_k\|_F. \tag{287}$$

Using the bound $\|T_k\|_F \lesssim r^{\frac{p-k}{4}} \lesssim 1$ and summing over $k \leq p$, we obtain

$$\sum_{k\leq p}\sqrt{\mathbb{E}[h_{T_k}^2]} \lesssim V_{p,\epsilon}\sum_{k\leq p}(\kappa^{T_1})^k(\log d)^k \lesssim V_{p,\epsilon}(\kappa^{T_1})^p(\log d)^p. \tag{288}$$

where the last step absorbs the polynomial degree $p$ into the $\lesssim$ constant. Further, squaring both sides yields

$$\mathbb{E}[h^2] \lesssim V_{p,\epsilon}^2(\kappa^{T_1})^{2p}(\log d)^{2p} \tag{289}$$

which is the first inequality of Equation (293). For the sup-norm, substitute the sup-norm bound mentioned in Lemma D.10 we know, for each $k$,

$$\|h_{T_k}\|_\infty \lesssim V_{k,\epsilon_k}(\kappa^{2T_1}\log^2 d)^k\|T_k\|_F. \tag{290}$$

Combining the $k$ bounds together we have

$$\|h\|_\infty \leq \sum_{k\leq p}\|h_{T_k}\|_\infty \lesssim \sum_{k\leq p}V_{k,\epsilon_k}(\kappa^{2T_1}(\log d)^2)^k\|T_k\|_F \lesssim V_{p,\epsilon}(\kappa^{2T_1}(\log d)^2)^p \tag{291}$$

This proves the second one of Equation (293), concluding our proof. $\qquad\square$

### D.5.9 PROOF OF LEMMA D.14

*Proof.* We first bound $\|a^*\|$ where $a^* = (a_1^*, \ldots, a_m^*)$. Since

$$a_j^* := \frac{1}{m}h(a_j, b_j, w_j) \tag{292}$$

and via Lemma D.13 we have the following moment estimation

$$\|h\|_\infty \lesssim V_{p,\epsilon}(\kappa^{T_1})^{2p}(\log d)^{2p}. \tag{293}$$

Thus we directly get as desired

$$\|a^*\|^2 \lesssim \frac{V_{p,\epsilon}^2(\kappa^{T_1})^{4p}(\log d)^{4p}}{m} \tag{294}$$

To bound the square loss for finite width, take the basic decomposition

$$\frac{1}{n}\sum_{i=n+1}^{2n}\left(f_{\widetilde{\Theta}}(x_i) - f^*(x_i)\right)^2 \leq 2\frac{1}{n}\sum_{i=n+1}^{2n}\left(f_{\widetilde{\Theta}^*}(x_i) - f_h(x_i)\right)^2 + 2\frac{1}{n}\sum_{i=n+1}^{2n}\left(f_h(x_i) - f^*(x_i)\right)^2. \tag{295}$$

We next bound these two terms separately.

For the second term, by Lemma D.13 we have

$$2\frac{1}{n}\sum_{i=n+1}^{2n}\left(f_h(x_i) - f^*(x_i)\right)^2 \lesssim \left(\sum_{k\leq p}\Delta_k\right)^2 + d^{-D/4} \tag{296}$$

It thus suffices to control the Monte Carlo fluctuation term (the first term). Note that the parameters are initialized symmetrically and this symmetry is preserved by first stage training, that is, the first half neurons are free. Therefore we should group neurons into symmetric pairs to help the proof.

For $j \leq m/2$, define

$$Z_j(x) := a_j^*\sigma((\widetilde{w}_j^{(T_1)})^\mathsf{T}x + b_j) + a_{m-j}^*\sigma((\widetilde{w_{m-j}}^{(T_1)})^\mathsf{T}x + b_{m-j}), \quad \widetilde{w}_j^{(T_1)} = (-\eta a_j)^{T_1}h_n(w_j) \tag{297}$$

so that $f_{\widetilde{\Theta}^*}(x) - f_h(x) = \sum_{j\leq m/2}(Z_j(x) - \mathbb{E}[Z_j(x)])$. Introduce the truncated version

$$\overline{Z}_j(x) := Z_j(x)\mathbf{1}_{\{|\widetilde{w}_j^{(T_1)\mathsf{T}}x|\leq 1, |\widetilde{w}_{m-j}^{(T_1)\mathsf{T}}x|\leq 1\}} \tag{298}$$

By the concentration event $\mathcal{E}_\mu$, which occurs with probability at least $1 - O(d^{-D/2})$ we have $\overline{Z}_j(\boldsymbol{x}_i) = Z_j(\boldsymbol{x}_i)$ for all $n + 1 \leq i \leq 2n$ and all $j \leq m/2$. Hence, on $\mathcal{E}$, we have the following decomposition

$$f_{\widetilde{\Theta}^*}(\boldsymbol{x}_i) - f_h(\boldsymbol{x}_i) = \sum_{j \leq m/2} \left( \overline{Z}_j(\boldsymbol{x}_i) - \mathbb{E}\overline{Z}_j(\boldsymbol{x}_i) \right) + \underbrace{\frac{m}{2} \left( \mathbb{E}\overline{Z}_j(\boldsymbol{x}_i) - \mathbb{E}Z_j(\boldsymbol{x}_i) \right)}_{:=B_i}, \tag{299}$$

We bound the second term $B_i$ first.

Recall

$$B_i := \frac{m}{2}(\mathbb{E}\overline{Z}_j(\boldsymbol{x}_i) - \mathbb{E}Z_j(\boldsymbol{x}_i)) = \frac{m}{2}\mathbb{E}[(\overline{Z}_j(\boldsymbol{x}_i) - Z_j(\boldsymbol{x}_i))\mathbf{1}_{\mathcal{E}_\alpha \cap \mathcal{E}_\zeta \cap (\mathcal{E}_\beta \cap \mathcal{E}_\eta)^c}], \tag{300}$$

because $\overline{Z}_j = Z_j$ on $\mathcal{E}$. By Cauchy-Schwarz,

$$|B_i| \leq \frac{m}{2}\mathbb{E}[|Z_j(\boldsymbol{x}_i)|\mathbf{1}_{\mathcal{E}_\alpha \cap \mathcal{E}_\zeta \cap (\mathcal{E}_\beta \cap \mathcal{E}_\eta)^c}] + \frac{m}{2}\mathbb{E}[|\overline{Z}_j(\boldsymbol{x}_i)|\mathbf{1}_{\mathcal{E}_\alpha \cap \mathcal{E}_\zeta \cap (\mathcal{E}_\beta \cap \mathcal{E}_\eta)^c}]$$

$$\leq \frac{m}{2} \left( \sqrt{\mathbb{E}Z_j(\boldsymbol{x}_i)^2} + \sqrt{\mathbb{E}\overline{Z}_j(\boldsymbol{x}_i)^2} \right) \sqrt{\mathbb{P}(\mathcal{E}_\alpha \cap \mathcal{E}_\zeta \cap (\mathcal{E}_\beta \cap \mathcal{E}_\eta)^c)}$$

$$\leq m\sqrt{\mathbb{E}Z_j(\boldsymbol{x}_i)^2}\sqrt{\mathbb{P}(\mathcal{E}_\alpha \cap \mathcal{E}_\zeta \cap (\mathcal{E}_\beta \cap \mathcal{E}_\eta)^c)}. \tag{301}$$

The final inequality follows from the fact that truncation does not increase magnitude, that is, $|\overline{Z}_j(\boldsymbol{x}_i)| \leq |Z_j(\boldsymbol{x}_i)|$. For a symmetric pair

$$Z_j(\boldsymbol{x}_i) = a_j^* \sigma(\widetilde{\boldsymbol{w}}_j^{(\mathrm{T}_1)\mathsf{T}}\boldsymbol{x}_i + b_j) + a_{m-j}^* \sigma(\widetilde{\boldsymbol{w}}_{m-j}^{(\mathrm{T}_1)\mathsf{T}}\boldsymbol{x}_i + b_{m-j}) \tag{302}$$

Recall that it has been assumed that the activation function has bounded third derivatives as in Assumption 2. Therefore, there exists an absolute constant $C_\sigma > 0$ such that

$$|\sigma(u)| \leq C_\sigma(1 + |u|^3) \quad \text{for all } u \in \mathbb{R}. \tag{303}$$

Then, for any fixed $i, j$,

$$\mathbb{E}[Z_j(\boldsymbol{x}_i)^2] \lesssim (a_j^*)^2(1 + |\eta^{\mathrm{T}_1}\boldsymbol{h}_n(\boldsymbol{w}_j^{(0)})^\mathsf{T}\boldsymbol{x}_i + b_j|^3)^2 + (a_{m-j}^*)^2(1 + |\eta^{\mathrm{T}_1}\boldsymbol{h}_n(\boldsymbol{w}_{m-j}^{(0)})^\mathsf{T}\boldsymbol{x}_i + b_{m-j}|^3)^2. \tag{304}$$

Notice that under the event $\mathcal{E}_\zeta$, we have $\boldsymbol{x}_i$ is bounded for all $i \in [n+1, 2n]$ by $\|\boldsymbol{x}_i\| \leq 2\sqrt{d}$. Thus via another crude bound, since we have $|\eta^{\mathrm{T}_1}\boldsymbol{h}_n(\boldsymbol{w}_j^{(0)})^\mathsf{T}\boldsymbol{x}_i + b_j| \lesssim d$ and $|\eta^{\mathrm{T}_1}\boldsymbol{h}_n(\boldsymbol{w}_{m-j}^{(0)})^\mathsf{T}\boldsymbol{x}_i + b_{m-j}| \lesssim d$ and we have

$$\sqrt{\mathbb{E}Z_j(\boldsymbol{x}_i)^2} \lesssim d^3\sqrt{(a_j^*)^2 + (a_{m-j}^*)^2}$$

$$\lesssim d^3\frac{V_{p,\epsilon}(\kappa^{\mathrm{T}_1})^{4p}(\log d)^{4p}}{m} \tag{305}$$

In addition, from Definition D.7 we also have $V_{k,\epsilon_k} = \|\boldsymbol{v}_k^{(\epsilon_k)}\|_\infty \lesssim \epsilon_k^{-\beta} \lesssim d^{\frac{D}{12}}$, thus $V_{p,\epsilon}^2 \lesssim d^{\frac{D}{6}}$

Also note that for $\mathrm{T}_1 = o(\log d)$, we have $\kappa^{\mathrm{T}_1} \lesssim d$, thus it shows that

$$\sqrt{\mathbb{E}Z_j(\boldsymbol{x}_i)^2} \lesssim \frac{d^{4+4p+\frac{D}{12}}}{m} \tag{306}$$

This gives the intermediate bound

$$|B_i| \lesssim m\frac{d^{4+4p+\frac{D}{12}}}{m}\sqrt{\mathbb{P}((\mathcal{E}_\beta \cap \mathcal{E}_\eta)^c)} \lesssim d^{4+4p+\frac{D}{12}}\sqrt{\mathbb{P}((\mathcal{E}_\beta \cap \mathcal{E}_\eta)^c)}. \tag{307}$$

By the concentration estimates proved above we have $\Pr((\mathcal{E}_\beta \cap \mathcal{E}_\eta)^c) \lesssim d^{-D/2}$ (union bound over the $m/2$ independent draws yields the extra factor $m$), hence

$$|B_i| \lesssim d^{4+4p+\frac{D}{12}}d^{-D/4} \lesssim d^{-D/8}. \tag{308}$$

for sufficiently large universal constant $D$.

Fix any $\boldsymbol{x} \in \{\boldsymbol{x}_i\}_{i=n+1}^{2n}$. The variables $\{\overline{Z}_j(\boldsymbol{x}) - \mathbb{E}\overline{Z}_j(\boldsymbol{x})\}_{j \leq m/2}$ are i.i.d. and centered, so we can use Berstein to bound the sum. Notice that $(\boldsymbol{w}_j^{(\mathrm{T}_1)})^{\mathsf{T}}\boldsymbol{x}$ and $\boldsymbol{w}_{m-j}^{(\mathrm{T}_1)\mathsf{T}}\boldsymbol{x}$ is truncated by the defination of $Z_j(\boldsymbol{x})$, we have $\sigma((\boldsymbol{w}_j^{(\mathrm{T}_1)})^{\mathsf{T}}\boldsymbol{x} + b_j)$ and $\sigma((\boldsymbol{w}_{m-j}^{(\mathrm{T}_1)})^{\mathsf{T}}\boldsymbol{x} + b_{m-j})$ also bounded by a universal constant.

Combining with $a_j^* = \frac{1}{m}h(a_j, b_j, \boldsymbol{w}_j)$, we have for a universal constant $C$

$$|\overline{Z}_j(\boldsymbol{x}) - \mathbb{E}\overline{Z}_j(\boldsymbol{x})| \leq \frac{C}{m}\|h\|_\infty. \tag{309}$$

**Lemma D.35** (Hoeffding's inequality). *Let $X_1, X_2, \ldots, X_n$ be i.i.d. random variables with mean $\mu$ and $X_i \in [a, b]$. Then,*

$$\mathbb{P}\left\{\left|\frac{1}{n}\sum_{i=1}^{n} X_i - \mu\right| \geq t\right\} \leq 2\exp\left(-\frac{2nt^2}{(b-a)^2}\right). \tag{310}$$

Due to the fact that $|\overline{Z}_j(x)| \lesssim \frac{1}{m}\|h\|_\infty$, Hoeffding inequality then yields with probability at least $1 - \mathcal{O}(d^{-D})$

$$\left|\sum_{j \leq m/2}(\overline{Z}_j(\boldsymbol{x}) - \mathbb{E}\overline{Z}_j(\boldsymbol{x}))\right| \lesssim \frac{\log d \cdot \|h\|_\infty}{\sqrt{m}}. \tag{311}$$

Recall from Lemma D.13 we have $\|h\|_\infty \lesssim V_{p,\varepsilon}(\kappa^{\mathrm{T}_1})^{2p}(\log d)^{2p}$. Plugging these bounds into the above one, it shows that

$$\left|\sum_{j \leq m/2}(\overline{Z}_j(\boldsymbol{x}) - \mathbb{E}\overline{Z}_j(\boldsymbol{x}))\right| \lesssim \frac{V_{p,\varepsilon}(\kappa^{\mathrm{T}_1})^{2p}(\log d)^{2p+1}}{\sqrt{m}}. \tag{312}$$

Therefore

$$\left|f_{\widetilde{\Theta}^*}(\boldsymbol{x}_i) - f_h(\boldsymbol{x}_i)\right| \leq \left|\sum_{j \leq m/2}\left(\overline{Z}_j(\boldsymbol{x}_i) - \mathbb{E}\overline{Z}_j(\boldsymbol{x}_i)\right)\right| + \frac{m}{2}\left|\mathbb{E}\overline{Z}_j(\boldsymbol{x}_i) - \mathbb{E}Z_j(\boldsymbol{x}_i)\right|$$

$$\lesssim \sqrt{\frac{V_{p,\epsilon}^2(\kappa^{\mathrm{T}_1})^{4p}(\log d)^{4p+2}}{m}} + d^{-D/8} \tag{313}$$

Squaring both sides and applying a union bound over the $n$ training points, we obtain the following.

$$\frac{1}{n}\sum_{i=n+1}^{2n}\left(f_{\widetilde{\Theta}^*}(\boldsymbol{x}_i) - f_h(\boldsymbol{x}_i)\right)^2 \lesssim \frac{V_{p,\epsilon}^2(\kappa^{\mathrm{T}_1})^{4p}(\log d)^{4p+2}}{m} + d^{-D/4} \tag{314}$$

By the initial decomposition and Lemma D.13,

$$\frac{1}{n}\sum_{i=n+1}^{2n}\left(f_{\widetilde{\Theta}^*}(\boldsymbol{x}_i) - f^*(\boldsymbol{x}_i)\right)^2 \lesssim \left(\sum_{k \leq p}\Delta_k\right)^2 + d^{-\frac{D}{4}} + \frac{V_{p,\epsilon}^2(\kappa^{\mathrm{T}_1})^{4p}(\log d)^{4p+2}}{m}, \tag{315}$$

which is (177). This completes the proof. $\qquad\square$

### D.5.10 PROOF OF LEMMA D.15

*Proof.* From Lemma D.14 and Lemma D.13, we know that with high probability at least $1 - \mathcal{O}(d^{-D/2})$ there exists $\boldsymbol{a}^* \in \mathbb{R}^m$ such that for $\widetilde{\boldsymbol{\Theta}}^* = (\boldsymbol{a}^*, \boldsymbol{b}^{(\mathrm{T}_1)}, \widetilde{\boldsymbol{W}}^{(\mathrm{T}_1)})$,

$$
\begin{aligned}
\frac{1}{n} \sum_{i=n+1}^{2n} \left(f_{\widetilde{\boldsymbol{\Theta}}^*}(\boldsymbol{x}_i) - f^*(\boldsymbol{x}_i)\right)^2 &\lesssim \left(\sum_{k\leq p} \Delta_k\right)^2 + d^{-D/4} + \frac{V_{p,\epsilon}^2 (\kappa^{\mathrm{T}_1})^{4p} (\log d)^{4p+2}}{m} \\
&\lesssim \left(\sum_{k\leq p} \Delta_k\right)^2 + \frac{V_{p,\epsilon}^2 (\kappa^{\mathrm{T}_1})^{4p} (\log d)^{4p+2}}{m}
\end{aligned}
\tag{316}
$$

The last inequality is because $m \leq d^{D/4}$, so $d^{-D/4} \lesssim \frac{1}{m}$, thus $d^{-D/4}$ can be absorbed by $\frac{V_{p,\epsilon}^2 (\kappa^{\mathrm{T}_1})^{4p} (\log d)^{4p+2}}{m}$

By Jensen's inequality and the Lipschitz property of $\ell$, we have

$$
\begin{aligned}
\widehat{\mathcal{L}}(f_{\widetilde{\boldsymbol{\Theta}}^*}) - \widehat{\mathcal{L}}(f^*) &= \frac{1}{n} \sum_{i=1}^{n} \left[\ell(f_{\widetilde{\boldsymbol{\Theta}}^*}(\boldsymbol{x}_{i+n}), y_i) - \ell(f^*(\boldsymbol{x}_{i+n}), y_i)\right] \\
&\leq \mathrm{L} \frac{1}{n} \sum_{i=1}^{n} |f_{\widetilde{\boldsymbol{\Theta}}^*}(\boldsymbol{x}_{i+n}) - f^*(\boldsymbol{x}_{i+n})| \\
&\leq \mathrm{L} \sqrt{\frac{1}{n} \sum_{i=1}^{n} \left(f_{\widetilde{\boldsymbol{\Theta}}^*}(\boldsymbol{x}_{i+n}) - f^*(\boldsymbol{x}_{i+n})\right)^2} \\
&\lesssim \mathrm{L} \left(\sum_{k\leq p} \Delta_k + \sqrt{\frac{V_{p,\epsilon}^2 (\kappa^{\mathrm{T}_1})^{4p} (\log d)^{4p+2}}{m}}\right).
\end{aligned}
\tag{317}
$$

The proof is complete.

$\square$

### D.5.11 PROOF OF LEMMA D.16

*Proof.* Note that

$$
\begin{aligned}
\left|\widehat{\mathcal{L}}(f_{\widetilde{\boldsymbol{\Theta}}^*}) - \widehat{\mathcal{L}}(f_{\boldsymbol{\Theta}^*})\right| &= \left|\frac{1}{n} \sum_{i=1}^{n} \left[\ell(f_{\widetilde{\boldsymbol{\Theta}}^*}(\boldsymbol{x}_{i+n}), y_i) - \ell(f_{\boldsymbol{\Theta}^*}(\boldsymbol{x}_{i+n}), y_i)\right]\right| \\
&\leq \mathrm{L} \sup_{i\in[n]} \left|f_{\widetilde{\boldsymbol{\Theta}}^*}(\boldsymbol{x}_{i+n}) - f_{\boldsymbol{\Theta}^*}(\boldsymbol{x}_{i+n})\right| \\
&\leq \mathrm{L} m \sup_{i\in[n], j\in[m]} \left|\boldsymbol{a}_j^* \sigma((\widetilde{\boldsymbol{w}}_j^{(\mathrm{T}_1)})^{\mathsf{T}} \boldsymbol{x}_{i+n} + b_j) - \boldsymbol{a}_j^* \sigma((\boldsymbol{w}_j^{(\mathrm{T}_1)})^{\mathsf{T}} \boldsymbol{x}_{i+n} + b_j)\right| \\
&\leq \mathrm{L} m \|\boldsymbol{a}^*\| \sup_{i\in[n], j\in[m]} \left|\sigma((\widetilde{\boldsymbol{w}}_j^{(\mathrm{T}_1)})^{\mathsf{T}} \boldsymbol{x}_{i+n} + b_j) - \sigma((\boldsymbol{w}_j^{(\mathrm{T}_1)})^{\mathsf{T}} \boldsymbol{x}_{i+n} + b_j)\right|.
\end{aligned}
\tag{318}
$$

By Assumption 2 we know that $\sigma'(0) = 0$, $\sigma''(0) = 1$ and $\left|\sigma^{(3)}(\cdot)\right| \leq M$. So we can bound the gap of $\sigma((\widetilde{\boldsymbol{w}}_j^{(\mathrm{T}_1)})^{\mathsf{T}} \boldsymbol{x} + b_j) - \sigma((\boldsymbol{w}_j^{(\mathrm{T}_1)})^{\mathsf{T}} \boldsymbol{x} + b_j)$ by the following lemma.

**Lemma D.36.** *If an activation function $\sigma$ satisfies $\sigma'(0) = 0$, $\sigma''(0) = 1$ and $\left|\sigma^{(3)}(\cdot)\right| \leq M$, then we have for all $x, y \in \mathbb{R}$:*

$$
|\sigma(x) - \sigma(y)| \leq \left[\tfrac{1}{2}(|x| + |y|) + \tfrac{M}{2}(|x|^2 + |y|^2)\right] \cdot |x - y|.
\tag{319}
$$

*Proof of Lemma D.36.* Define $r(t) := \sigma(t) - \frac{1}{2}t^2 - \sigma(0)$. Since $\sigma''(0) = 1$ and $\|\sigma^{(3)}\|_\infty \le M$, we have that $\sigma''$ is $M$-Lipschitz. Hence

$$|r'(t)| = |\sigma'(t) - t| = \left|\int_0^t (\sigma''(s) - 1)\, ds\right| \le \tfrac{M}{2}|t|^2. \tag{320}$$

By the mean value theorem,

$$|r(x) - r(y)| \le \sup_{\xi \in [x,y]} |r'(\xi)| \cdot |x - y| \le \tfrac{M}{2}\max\{|x|, |y|\}^2 |x - y|. \tag{321}$$

Therefore,

$$|\sigma(x) - \sigma(y)| \le \tfrac{1}{2}|x^2 - y^2| + |r(x) - r(y)| \le \tfrac{1}{2}(|x| + |y|)|x - y| + \tfrac{M}{2}\max\{|x|, |y|\}^2|x - y|. \tag{322}$$

Finally, since $\max\{a, b\}^2 \le a^2 + b^2$, we get the conclusion. $\square$

Back to the proof of Lemma D.16, under event $\mathcal{E}_\mu$, from Lemma D.9 notice that for all $x$ in $\{x_i\}_{i=n+1}^{2n}$, we have $(w_j^{(\mathrm{T}_1)})^\mathsf{T}x$ and $w_{m-j}^{(\mathrm{T}_1)\mathsf{T}}x$ are truncated, so $\sigma((w_j^{(\mathrm{T}_1)})^\mathsf{T}x + b_j)$ and $\sigma((w_{m-j}^{(\mathrm{T}_1)})^\mathsf{T}x + b_{m-j})$ are bounded by a universal constant. Respectively, we have $(\widetilde{w_j}^{(\mathrm{T}_1)})^\mathsf{T}x$ and $\widetilde{w_{m-j}}^{(\mathrm{T}_1)\mathsf{T}}x$ are truncated, so $\sigma((\widetilde{w}_j^{(\mathrm{T}_1)})^\mathsf{T}x + b_j)$ and $\sigma((\widetilde{w_{m-j}}^{(\mathrm{T}_1)})^\mathsf{T}x + b_{m-j})$ are also bounded by a universal constant. Thus we have

$$\begin{aligned}
\left|\widehat{\mathcal{L}}(f_{\widetilde{\Theta}^*}) - \widehat{\mathcal{L}}(f_{\Theta^*})\right| &\le \mathrm{L}m\|a^*\| \sup_{i \in [n], j \in [m]} \left|\sigma((\widetilde{w}_j^{(\mathrm{T}_1)})^\mathsf{T}x_{i+n} + b_j) - \sigma((w_j^{(\mathrm{T}_1)})^\mathsf{T}x_{i+n} + b_j)\right| \\
&\lesssim \mathrm{L}m\|a^*\| \sup_{i \in [n], j \in [m]} \left|(\widetilde{w}_j^{(\mathrm{T}_1)})^\mathsf{T}x_{i+n} - (w_j^{(\mathrm{T}_1)})^\mathsf{T}x_{i+n}\right| \\
&\lesssim \mathrm{L}m\|a^*\|\sqrt{d} \sup_{i \in [n], j \in [m]} \left\|\widetilde{w}_j^{(\mathrm{T}_1)} - w_j^{(\mathrm{T}_1)}\right\| \\
&\lesssim \mathrm{L}m\frac{1}{\epsilon_0}\|a^*\|\sqrt{d}\eta^{\mathrm{T}_1}\|\Delta\|
\end{aligned} \tag{323}$$

where $\Delta$ is defined in Lemma C.7, and from this lemma, we have the bound

$$\|\Delta\| \lesssim \left(\tfrac{5}{4}\right)^{\mathrm{T}_1 - 1} \cdot Md^2 \cdot \epsilon_0^2 \tag{324}$$

By Assumption 7, we know that $\epsilon_0 \le \frac{1}{C_\epsilon mn^{1/2}d^{7/2}}\left(\tfrac{4}{5}\right)^{\mathrm{T}_1}$, where $C_\epsilon$ is a universal constant. Moreover, we have $\|a^*\|^2 \lesssim \frac{V_{p,\epsilon}^2(\kappa^{\mathrm{T}_1})^{4p}(\log d)^{4p}}{m}$. Thus, it holds that

$$\begin{aligned}
\left|\widehat{\mathcal{L}}(f_{\widetilde{\Theta}^*}) - \widehat{\mathcal{L}}(f_{\Theta^*})\right| &\lesssim \mathrm{L}m\frac{1}{\epsilon_0}\|a^*\|\sqrt{d}\eta^{\mathrm{T}_1}\|\Delta\| \\
&\lesssim \mathrm{L}m\|a^*\|d\mathrm{T}_1 \cdot \left(\tfrac{5}{4}\right)^{\mathrm{T}_1 - 1} \cdot Md^2 \cdot \epsilon_0 \\
&\lesssim \mathrm{L}m\sqrt{\frac{V_{p,\epsilon}^2(\kappa^{\mathrm{T}_1})^{4p}(\log d)^{4p}}{m}}d^3\log dM\frac{1}{C_\epsilon m\sqrt{n}d^{\frac{7}{2}}} \\
&\lesssim \mathrm{L}\sqrt{\frac{V_{p,\epsilon}^2(\kappa^{\mathrm{T}_1})^{4p}(\log d)^{4p}}{m}}\log d\frac{1}{\sqrt{n}} \lesssim \mathrm{L}\sqrt{\frac{V_{p,\epsilon}^2(\kappa^{\mathrm{T}_1})^{4p}(\log d)^{4p}}{m}}.
\end{aligned} \tag{325}$$

The proof is complete. $\square$

### D.5.12 PROOF OF LEMMA D.17

*Proof.* Lemma D.15 implies that

$$\widehat{\mathcal{L}}(f_{\widetilde{\Theta}^*}) \lesssim \mathrm{L}\left(\sum_{k \le p}\Delta_k + \sqrt{\frac{V_{p,\epsilon}^2(\kappa^{\mathrm{T}_1})^{4p}(\log d)^{4p+2}}{m}}\right). \tag{326}$$

Lemma D.16 further suggests that

$$\left|\widehat{\mathcal{L}}(f_{\widetilde{\Theta}^*}) - \widehat{\mathcal{L}}(f_{\Theta^*})\right| \lesssim \mathrm{L}\sqrt{\frac{V_{p,\epsilon}^2(\kappa^{\mathrm{T}_1})^{4p}(\log d)^{4p}}{m}}. \tag{327}$$

By combining these two bounds, we easily obtain

$$\left|\widehat{\mathcal{L}}(f_{\Theta^*}) - \widehat{\mathcal{L}}(f^*)\right| \lesssim \mathrm{L}\left(\sum_{k\leq p}\Delta_k + \sqrt{\frac{V_{p,\epsilon}^2(\kappa^{\mathrm{T}_1})^{4p}(\log d)^{4p+2}}{m}}\right). \tag{328}$$

The proof is complete.

$$\square$$

### D.5.13 PROOF OF LEMMA D.18

*Proof.* From Lemma D.17, we have the following bound

$$\widehat{\mathcal{L}}(f_{\Theta^*}) \lesssim \mathrm{L}\left(\sum_{k\leq p}\Delta_k + \sqrt{\frac{V_{p,\epsilon}^2(\kappa^{\mathrm{T}_1})^{4p}(\log d)^{4p+2}}{m}}\right) = J \tag{329}$$

Consider the training objective $F(\boldsymbol{a}) = \widehat{\mathcal{L}}\left((\boldsymbol{a}, \boldsymbol{b}^{(\mathrm{T}_1)}, \boldsymbol{W}^{(\mathrm{T}_1)})\right) + \frac{1}{2}\beta_2\|\boldsymbol{a}\|^2$ with $\beta_2 = \frac{J}{U^2}$.

Let

$$\boldsymbol{a}^{(\infty)} := \arg\min_{\boldsymbol{a}\in\mathbb{R}^m} F(\boldsymbol{a}) \tag{330}$$

be the minimizer of $F$.

We further denote $\epsilon_0$ is the prescribed optimization tolerance $\|\boldsymbol{a}^{(\mathrm{T})} - \boldsymbol{a}^{(\infty)}\| \leq \epsilon_0$. Furthermore, let $\epsilon_0 = \frac{1}{m\mathrm{L}}$.

By optimality of $\boldsymbol{a}^{(\infty)}$ and the *constructed* comparator $\boldsymbol{a}^*$ we have

$$F(\boldsymbol{a}^{(\infty)}) \leq F(\boldsymbol{a}^*) = \widehat{\mathcal{L}}(\boldsymbol{a}^*) + \frac{\beta_2}{2}\|\boldsymbol{a}^*\|_2^2. \tag{331}$$

Here we only use that $\boldsymbol{a}^*$ is a feasible point with $\|\boldsymbol{a}^*\| \lesssim U$ and $\widehat{\mathcal{L}}(\boldsymbol{a}^*) \lesssim J$ (Lemma D.17).

On the other hand,

$$F(\boldsymbol{a}^{(\infty)}) = \widehat{\mathcal{L}}(\boldsymbol{a}^{(\infty)}) + \frac{\beta_2}{2}\|\boldsymbol{a}^{(\infty)}\|^2 \geq \frac{\beta_2}{2}\|\boldsymbol{a}^{(\infty)}\|^2, \tag{332}$$

so combining with (331) gives

$$\|\boldsymbol{a}^{(\infty)}\|^2 \leq \|\boldsymbol{a}^*\|^2 + \frac{2}{\beta_2}\widehat{\mathcal{L}}(\boldsymbol{a}^*) \lesssim U^2 + \frac{2J}{\beta_2} = U^2 + \frac{2J}{J/U^2} \lesssim U^2. \tag{333}$$

Thus

$$\|\boldsymbol{a}^{(\infty)}\| \lesssim U. \tag{334}$$

and also

$$\|\boldsymbol{a}^{(\mathrm{T})}\| \leq \|\boldsymbol{a}^{(\infty)}\| + \|\boldsymbol{a}^{(\mathrm{T})} - \boldsymbol{a}^{(\infty)}\| \leq \epsilon_0 + U \lesssim U \tag{335}$$

$$\widehat{\mathcal{L}}(\boldsymbol{a}^{(\infty)}) = F(\boldsymbol{a}^{(\infty)}) - \frac{\beta_2}{2}\|\boldsymbol{a}^{(\infty)}\|_2^2 \leq F(\boldsymbol{a}^*) \lesssim J + \frac{\beta_2}{2}U^2 \lesssim J. \tag{336}$$

Thus we have

$$\begin{aligned}
|F(\boldsymbol{a}^{(\mathrm{T})}) - F(\boldsymbol{a}^{(\infty)})| &\lesssim \sup_{i\in[n+1,2n]}\left\{\mathrm{L}\left|\boldsymbol{a}^{(\mathrm{T})\top}\sigma(\boldsymbol{W}^{(\mathrm{T})}\boldsymbol{x}_i + \boldsymbol{b}^{(\mathrm{T})}) - \boldsymbol{a}^{(\infty)\top}\sigma(\boldsymbol{W}^{(\mathrm{T})}\boldsymbol{x}_i + \boldsymbol{b}^{(\mathrm{T})})\right|\right\} \\
&\quad + \beta_2(\|\boldsymbol{a}^{(\mathrm{T})}\| + \|\boldsymbol{a}^{(\infty)}\|)\|\boldsymbol{a}^{(\mathrm{T})} - \boldsymbol{a}^{(\infty)}\| \\
&\lesssim \mathrm{L}\|\boldsymbol{a}^{(\mathrm{T})} - \boldsymbol{a}^{(\infty)}\| \cdot \sup_{i\in[n+1,2n]}\|\sigma(\boldsymbol{W}^{(\mathrm{T})}\boldsymbol{x}_i + \boldsymbol{b}^{(\mathrm{T})})\| \\
&\quad + \beta_2(\|\boldsymbol{a}^{(\mathrm{T})}\| + \|\boldsymbol{a}^{(\infty)}\|)\|\boldsymbol{a}^{(\mathrm{T})} - \boldsymbol{a}^{(\infty)}\|
\end{aligned} \tag{337}$$

By previous proof $\|\sigma(\boldsymbol{W}^{(\mathrm{T})}\boldsymbol{x}_i + \boldsymbol{b}^{(\mathrm{T})})\|_\infty$ can be bounded by an absolute constant for all $i$ under event $\mathcal{E}$. Thus we can have a loose $\ell^2$ norm bound and

$$
\begin{aligned}
|F(\boldsymbol{a}^{(\mathrm{T})}) - F(\boldsymbol{a}^{(\infty)})| &\lesssim \mathrm{L}\|\boldsymbol{a}^{(\mathrm{T})} - \boldsymbol{a}^{(\infty)}\| \cdot \sup_{i \in [n+1, 2n]} \|\sigma(\boldsymbol{W}^{(\mathrm{T})}\boldsymbol{x}_i + \boldsymbol{b}^{(\mathrm{T})})\| \\
&\quad + \beta_2(\|\boldsymbol{a}^{(\mathrm{T})}\| + \|\boldsymbol{a}^{(\infty)}\|)\|\boldsymbol{a}^{(\mathrm{T})} - \boldsymbol{a}^{(\infty)}\| \\
&\lesssim \mathrm{L}\|\boldsymbol{a}^{(\mathrm{T})} - \boldsymbol{a}^{(\infty)}\|\sqrt{m} + \beta_2 U\|\boldsymbol{a}^{(\mathrm{T})} - \boldsymbol{a}^{(\infty)}\| \\
&\lesssim (\mathrm{L}\sqrt{m} + \beta_2 U)\epsilon_0
\end{aligned}
\tag{338}
$$

Denote $\epsilon_{\mathrm{opt}} = (L\sqrt{m} + \beta_2 U)\epsilon_0 = \frac{L\sqrt{m}+\beta_2 U}{mL}$. It is easy to know that $\epsilon_{opt} = \frac{L\sqrt{m}+\beta_2 U}{mL} = \frac{1}{\sqrt{m}} + \frac{J}{UmL} \lesssim J$. Thus, we obtain

$$
\widehat{\mathcal{L}}(\boldsymbol{a}^{(\mathrm{T})}) \lesssim J.
\tag{339}
$$

Denote

$$
\phi_i := \sigma(\boldsymbol{W}^{(\mathrm{T}_1)}\boldsymbol{x}_i + \boldsymbol{b}^{(\mathrm{T}_1)}) \in \mathbb{R}^m, \quad i = n+1, \ldots, 2n.
\tag{340}
$$

By Assumption 3 on the loss function $\ell$, the function $F(\boldsymbol{a}) - \frac{\beta_2}{2}\|\boldsymbol{a}\|^2$ is convex and satisfies the gradient Lipschitz condition

$$
\begin{aligned}
\|\nabla\widehat{\mathcal{L}}(\boldsymbol{a}_1) - \nabla\widehat{\mathcal{L}}(\boldsymbol{a}_2)\| &= \left\| \frac{1}{n} \sum_{i=n+1}^{2n} \left( \ell'(\boldsymbol{a}_1^\top \phi_i) - \ell'(\boldsymbol{a}_2^\top \phi_i) \right) \phi_i \right\| \\
&\leq \frac{1}{n} \sum_{i=n+1}^{2n} \left| \ell'(\boldsymbol{a}_1^\top \phi_i) - \ell'(\boldsymbol{a}_2^\top \phi_i) \right| \|\phi_i\| \\
&\leq \frac{\mathrm{L}}{n} \sum_{i=n+1}^{2n} \left| (\boldsymbol{a}_1 - \boldsymbol{a}_2)^\top \phi_i \right| \|\phi_i\| \\
&= \mathrm{L} \left( \frac{1}{n} \sum_{i=n+1}^{2n} \|\phi_i\|^2 \right) \|\boldsymbol{a}_1 - \boldsymbol{a}_2\|.
\end{aligned}
\tag{341}
$$

By previous arguments, $\|\sigma(\boldsymbol{W}^{(\mathrm{T})}\boldsymbol{x}_i + \boldsymbol{b}^{(\mathrm{T})})\|_\infty$ can be bounded by an absolute constant for all $i \in [n+1, 2n]$ under event $\mathcal{E}$. In addition, we have $\frac{1}{n}\sum_i \|\phi_i\|^2 = \mathrm{Tr}(\frac{1}{n}\sum_i \phi_i\phi_i^\top) \leq \sup_i \|\phi_i\|^2$. Thus the Lip constant for $F$ satisfies $L_{\mathrm{lip}} \lesssim Lm + J/U^2$. Therefore, by Lemma E.13, if the step size $\eta_2 \leq \frac{1}{L_{\mathrm{lip}}}$, gradient descent will approximate $\boldsymbol{a}^{(\infty)}$ to arbitrary accuracy in

$$
\mathrm{T}_2 \gtrsim \frac{1}{\eta_2 \beta_2} \log\left( \|\boldsymbol{a}^{(\mathrm{T}_1)} - \boldsymbol{a}^{(\infty)}\|^2 / \epsilon_0^2 \right)
\tag{342}
$$

steps.

Note that when the step size $\eta_2 = \frac{1}{Lm + J/U^2}$

$$
\frac{1}{\eta_2 \beta_2} \lesssim \frac{Lm + J/U^2}{\beta_2} \lesssim \frac{U^2 Lm}{J}
\tag{343}
$$

Using the assumption that $m \lesssim d^{\frac{D}{4}}$ and $\|\boldsymbol{a}^{(\infty)}\| \lesssim U, \|\boldsymbol{a}^{(\mathrm{T}_1)}\| \lesssim 1$, we can further simplify:

$$
\frac{1}{\eta_2 \beta_2} \log\left( \frac{\|\boldsymbol{a}^{(\mathrm{T}_1)} - \boldsymbol{a}^{(\infty)}\|^2}{\epsilon_0^2} \right) \lesssim \frac{U^2 Lm}{J} \log(Um)
\tag{344}
$$

Thus, we conclude that gradient descent will approximate $\boldsymbol{a}^{(\infty)}$ to arbitrary accuracy in at most

$$
\mathrm{T}_2 \gtrsim \frac{U^2 Lm}{J} \log(Um)
\tag{345}
$$

steps. The proof is complete. $\qquad\square$

### D.5.14 PROOF OF LEMMA D.19

*Proof.* Let $\mathcal{F}\big(\boldsymbol{b}^{(\mathrm{T})}, \boldsymbol{W}^{(\mathrm{T})}\big) := \{f(\boldsymbol{x}) = \boldsymbol{a}^\mathsf{T}\sigma(\boldsymbol{W}^{(\mathrm{T})}\boldsymbol{x} + \boldsymbol{b}^{(\mathrm{T})}), \boldsymbol{\Theta} : \|\boldsymbol{a}\| \lesssim U\}$. Note that under $\mathcal{E}$ we have $f_{\boldsymbol{\Theta}^{(\mathrm{T})}} \in \mathcal{F}$. We first define a new event for $\boldsymbol{x} \sim \mathcal{N}(0, I)$ under fixed $\boldsymbol{W}^{(\mathrm{T})}$ and $\boldsymbol{b}^{(\mathrm{T})}$

$$\mathcal{M}_\alpha^{(j)} := \left\{ |\boldsymbol{h}_n(\boldsymbol{w}_j^{(0)})^\mathsf{T}\boldsymbol{x}| \le \|\boldsymbol{h}_n(\boldsymbol{w}_j^{(0)})\|\sqrt{4D\log d} \right\} \tag{346}$$

Define the finite-width event for the initialization as the intersection $\mathcal{M}_\alpha := \bigcap_{j=1}^{m/2} \mathcal{M}_\alpha^{(j)}$. It is straightforward to have $\mathbb{P}(\mathcal{M}_\alpha | \boldsymbol{H}_\ell, \{\boldsymbol{w}_j^{(0)}\}_{j=0}^m) \ge 1 - \mathcal{O}(d^{-D})$. We also define another event for $\boldsymbol{x}$ in order to bound its first $r$ components

$$\mathcal{M}_\beta := \left\{ \sup_{k \in [r]} |(\boldsymbol{x})_k| \le \sqrt{4D\log d} \right\} \tag{347}$$

Applying a union bound gives immediately $\mathbb{P}(\mathcal{M}_\beta) \ge 1 - \mathcal{O}(d^{-D})$.

Denote $\mathcal{M} = \mathcal{M}_\alpha \cap \mathcal{M}_\beta$. Define the truncated loss

$$\ell^\mathcal{M}(t, y) := \ell(t, y)\mathbf{1}\{\boldsymbol{x} \in \mathcal{M}\}. \tag{348}$$

$$\widehat{\mathcal{L}}^\mathcal{M}(f) := \frac{1}{n}\sum_{i=1}^n \ell^\mathcal{M}\big(f(\boldsymbol{x}_{i+n}), y_{i+n}\big), \qquad \mathcal{L}^\mathcal{M}(f) := \mathbb{E}\big[\ell^\mathcal{M}(f(\boldsymbol{x}), y)\big]. \tag{349}$$

By the symmetrization lemma,

$$\sup_{f \in \mathcal{F}} |\widehat{\mathcal{L}}^\mathcal{M}(f) - \mathcal{L}^\mathcal{M}(f)| \le 2\mathbb{E}_\sigma\left[\sup_{f \in \mathcal{F}} \frac{1}{n}\sum_{i=1}^n \xi_i \ell^\mathcal{M}\big(f(\boldsymbol{x}_{i+n}), y_{i+n}\big)\right] \tag{350}$$

where $\{\xi_i\}_{i=1}^n$ are i.i.d. Rademacher signs independent of $\{(\boldsymbol{x}_{i+n}, y_{i+n})\}_{i=1}^n$. Denote $S := \{(\boldsymbol{x}_{i+n}, y_{i+n})\}_{i=1}^n$. For fixed $(\boldsymbol{x}_{i+n}, y_{i+n})$, the mapping $t \mapsto \ell^\mathcal{M}(t, y_{i+n})$ has Lipschitz constant in $t$ satisfying

$$\big|\ell^\mathcal{M}(t, y) - \ell^\mathcal{M}(s, y)\big| = |\ell(t, y) - \ell(s, y)| \cdot \mathbf{1}\{\boldsymbol{x} \in \mathcal{M}\} \le \mathrm{L}|t - s|. \tag{351}$$

Thus, by the Ledoux–Talagrand contraction inequality Lemma E.5 applied coordinatewise to the maps $t \mapsto \ell^\mathcal{M}(t, y_{i+n})$,

$$\mathbb{E}_\xi\left[\sup_f \frac{1}{n}\sum_{i=1}^n \xi_i \ell^\mathcal{M}\big(f(\boldsymbol{x}_{i+n}), y_{i+n}\big)\right] \le \mathrm{L}\mathbb{E}_\xi\left[\sup_f \frac{1}{n}\sum_{i=1}^n \xi_i f(\boldsymbol{x}_{i+n})\right] = \mathrm{L}\widehat{\mathrm{Rad}}_n(\mathcal{F}). \tag{352}$$

Recalling $\mathrm{Rad}_n(\mathcal{F}) = \mathbb{E}[\widehat{\mathrm{Rad}}_n(\mathcal{F})]$, we obtain the in-expectation bound

$$\mathbb{E}_S\left[\sup_{f \in \mathcal{F}} |\widehat{\mathcal{L}}^\mathcal{M}(f) - \mathcal{L}^\mathcal{M}(f)|\right] \le 2\mathrm{L}\,\mathrm{Rad}_n(\mathcal{F}). \tag{353}$$

Furthermore, for $\boldsymbol{x}$ under the event $\mathcal{M}$, by the same argument as in Lemma D.9, we have $|(\boldsymbol{w}_j^{(\mathrm{T})})^\mathsf{T}\boldsymbol{x}| \lesssim 1$ for all $j$, thus we have

$$\left\|\sigma\big(\boldsymbol{W}^{(\mathrm{T})}\boldsymbol{x} + \boldsymbol{b}^{(\mathrm{T})}\big)\right\| \lesssim \sqrt{m} \tag{354}$$

$$|f(\boldsymbol{x})| = \boldsymbol{a}^\mathsf{T}\sigma\big(\boldsymbol{W}^{(\mathrm{T})}\boldsymbol{x} + \boldsymbol{b}^{(\mathrm{T})}\big) \lesssim U\sqrt{m}. \tag{355}$$

$$\begin{aligned}
\ell^\mathcal{M}(f(\boldsymbol{x}), y; \boldsymbol{x}) &= \ell(f(\boldsymbol{x}), y)\mathbf{1}\{\boldsymbol{x} \in \mathcal{M}\} \\
&\le \big(\ell(f(\boldsymbol{x}), y) - \ell(f^*(\boldsymbol{x}), y)\big)\mathbf{1}\{\boldsymbol{x} \in \mathcal{M}\} \\
&\le \mathrm{L}\,|f(\boldsymbol{x}) - f^*(\boldsymbol{x})|\,\mathbf{1}\{\boldsymbol{x} \in \mathcal{M}\} \\
&\lesssim \mathrm{L}U\sqrt{m} + \mathrm{L}(\log d)^p \lesssim \mathrm{L}U\sqrt{m}.
\end{aligned} \tag{356}$$

We now give the following two high probability events in order to bound the gap between empirical loss and population loss.

**Definition D.37** (Event $\mathcal{E}_\tau$ and $\mathcal{E}$). *We say the event $\mathcal{E}_\tau$ holds if*

$$\sup_{f \in \mathcal{F}} \left| \widehat{\mathcal{L}}^{\mathcal{M}}(f) - \mathcal{L}^{\mathcal{M}}(f) \right| \lesssim \left( \mathrm{L} \operatorname{Rad}_n(\mathcal{F}) + \mathrm{L}U\sqrt{m}\sqrt{\frac{\log d}{n}} \right), \tag{357}$$

*Using the standard empirical-to-population deviation bound by Lemma E.4 , we have $\mathcal{E}_\tau$ holds with probability at least $1 - \mathcal{O}(d^{-D})$.*

*Furthermore, we define event*

$$\mathcal{E} := \mathcal{E}_\mu \cap \mathcal{E}_\tau. \tag{358}$$

*Hence, by the preceding probability bounds (and a union bound), $\mathcal{E}$ occurs with probability at least $1 - \mathcal{O}(d^{-D/2})$.*

By Lemma E.6 we have $\operatorname{Rad}_n(\mathcal{F}) \lesssim U\sqrt{\frac{m}{n}}$. Thus

$$\sup_{f \in \mathcal{F}} \left| \widehat{\mathcal{L}}^{\mathcal{M}}(f) - \mathcal{L}^{\mathcal{M}}(f) \right| \lesssim \mathrm{L}U\sqrt{\frac{m\log d}{n}}. \tag{359}$$

We notice that under the event $\mathcal{E}_\mu$, $\mathcal{M}$ holds for all $\boldsymbol{x}_i$, $i \in [n+1, 2n]$, thus $\widehat{\mathcal{L}}^{\mathcal{M}}(f) = \widehat{\mathcal{L}}(f)$. For the other term, we have

$$\begin{aligned}
\mathcal{L}^{\mathcal{M}}(f) - \mathcal{L}(f) &= \mathbb{E}\big[\ell(f(\boldsymbol{x}), y)\big] - \mathbb{E}\big[\ell^{\mathcal{M}}(f(\boldsymbol{x}), y; \boldsymbol{x})\big] \\
&= \mathbb{E}\big[\ell(f(\boldsymbol{x}), y)\mathbf{1}\{\boldsymbol{x} \notin \mathcal{M}\}\big] \\
&\leq \mathrm{L}\mathbb{E}\big[|f(\boldsymbol{x}) - f^*(\boldsymbol{x})|\,\mathbf{1}\{\boldsymbol{x} \notin \mathcal{M}\}\big] \\
&\leq \mathrm{L}\left(\mathbb{E}\big[|f(\boldsymbol{x})|\mathbf{1}\{\boldsymbol{x} \notin \mathcal{M}\}\big] + \mathbb{E}\big[|f^*(\boldsymbol{x})|\mathbf{1}\{\boldsymbol{x} \notin \mathcal{M}\}\big]\right) \\
&\lesssim \mathrm{L}\left(\sqrt{\mathbb{E}\big[f(\boldsymbol{x})^2\mathbb{P}\{\boldsymbol{x} \notin \mathcal{M}\}\big]} + \sqrt{\mathbb{E}\big[(f^*(\boldsymbol{x}))^2\mathbb{P}\{\boldsymbol{x} \notin \mathcal{M}\}\big]}\right). 
\end{aligned} \tag{360}$$

Recall that it has been assumed that the activation function has bounded third derivatives as in Assumption 2. Therefore, there exists an absolute constant $C_\sigma > 0$ such that

$$|\sigma(u)| \leq C_\sigma(1 + |u|^3) \qquad \text{for all } u \in \mathbb{R}. \tag{361}$$

Via a crude moment estimation bound, we can see the right side is $\lesssim \mathrm{L}Ud^{-\frac{D}{8}}$. Thus, we have the following final estimation

$$\sup_{f \in \mathcal{F}} \left| \widehat{\mathcal{L}}(f) - \mathcal{L}(f) \right| \lesssim \mathrm{L}U\sqrt{\frac{m\log d}{n}} + \mathrm{L}Ud^{-D/8}. \tag{362}$$

Since we have assume $n \leq d^{D/4}$, we have $\sup_{f \in \mathcal{F}} \left| \widehat{\mathcal{L}}(f) - \mathcal{L}(f) \right| \lesssim \mathrm{L}U\sqrt{\frac{m\log d}{n}}$. Plug in the formula for $U$ and we get our desired result.

$\square$

### D.5.15 PROOF OF LEMMA D.21

*Proof.* Let $\epsilon_{\min} := \min_{0 \leq k \leq p} \epsilon_k$. Because $x \mapsto x^{-\beta}$ is strictly decreasing on $(0, \infty)$, we have $\max_{0 \leq k \leq p} \epsilon_k^{-\beta} = \epsilon_{\min}^{-\beta}$. Also note that $\sum_{k=0}^{p} \epsilon_k \geq (p+1)\epsilon_{\min}$, with equality if and only if all $\epsilon_k$ are equal. Consequently,

$$\Phi(\{\epsilon_k\}) \geq A\big((p+1)\epsilon_{\min} + (C_m + C_n)\epsilon_{\min}^{-\beta}\big) \tag{363}$$

and equality holds precisely at symmetric choices $\epsilon_0 = \cdots = \epsilon_p$. Therefore the problem reduces to minimizing the strictly convex one-variable function

$$g(\epsilon) := (p+1)\epsilon + (C_m + C_n)\epsilon^{-\beta} \quad (\epsilon > 0). \tag{364}$$

Since

$$g'(\epsilon) = (p+1) - \beta(C_m + C_n)\epsilon^{-(\beta+1)} \tag{365}$$

and

$$g''(\epsilon) = \beta(\beta+1)(C_m + C_n)\epsilon^{-(\beta+2)} > 0, \tag{366}$$

there is a unique critical point, which is the global minimizer

$$\epsilon^* = \left( \frac{\beta(C_m + C_n)}{p+1} \right)^{\frac{1}{\beta+1}} \tag{367}$$

Substituting back and using $(C_m + C_n)(\epsilon^*)^{-\beta} = \frac{p+1}{\beta}\epsilon^*$ yields

$$\min g = (\beta+1)\beta^{-\frac{\beta}{\beta+1}}(p+1)^{\frac{\beta}{\beta+1}}(C_m + C_n)^{\frac{1}{\beta+1}}. \tag{368}$$

Multiplying by $LA$ gives the stated expression for $\min \Phi$.

Finally, as $p$ is fixed, its contribution can be absorbed into the $\lesssim$ constant, also note that $1 \leq (\beta+1)\beta^{-\frac{\beta}{\beta+1}} \leq 2$ thus we get

$$\min_{\{\epsilon_k > 0\}} \Phi = A \cdot (\beta+1)\beta^{-\frac{\beta}{\beta+1}}(p+1)^{\frac{\beta}{\beta+1}}(C_m + C_n)^{\frac{1}{\beta+1}}$$

$$\lesssim A(C_m + C_n)^{\frac{1}{\beta+1}}$$

$$\lesssim \kappa^{2p\mathrm{T}_1} \log^{2p+1} d \left( \sqrt{\frac{1}{m}} + \sqrt{\frac{1}{n}} \right)^{\frac{1}{\beta+1}}. \tag{369}$$

The proof is complete.

$\square$

### D.5.16 PROOF OF LEMMA D.23

*Proof.* Let $\tau := \sqrt{\frac{\log d}{\log \kappa}}$, so $\mathrm{T}_1 = \lceil \tau \rceil \in [\tau, \tau+1]$. Then

$$\kappa^{2\mathrm{T}_1} = \exp(2\mathrm{T}_1 \log \kappa) \in \left[ d^{2\sqrt{\frac{\log \kappa}{\log d}}}, \, \kappa^2 d^{2\sqrt{\frac{\log \kappa}{\log d}}} \right], \tag{370}$$

and therefore

$$\mathrm{T}_1^2 d \log d \cdot \kappa^{2\mathrm{T}_1} \lesssim \kappa^2 (\log d)^2 d^{1+2\sqrt{\frac{\log \kappa}{\log d}}} \tag{371}$$

Moreover, since $\frac{1}{\mathrm{T}_1} \leq \frac{1}{\tau} = \sqrt{\frac{\log \kappa}{\log d}}$, we have

$$d^{1+1/\mathrm{T}_1}\kappa^2 \leq \kappa^2 d^{1+\sqrt{\frac{\log \kappa}{\log d}}} \tag{372}$$

and combining the two bounds yields

$$\mathrm{T}_1^2 d \log d \kappa^{2\mathrm{T}_1} + d^{1+1/\mathrm{T}_1}\kappa^2 \lesssim \kappa^2 \log(d) d^{1+2\sqrt{\frac{\log \kappa}{\log d}}} \lesssim d^{1+2\sqrt{\frac{\log \kappa}{\log d}}}(\log d)^2. \tag{373}$$

The proof is complete. $\square$

### D.5.17 PROOF OF LEMMA D.25

*Proof.* Plugging in $U$ and $J$ we get

$$\mathrm{T}_2 = \frac{U^2 Lm}{J} \log(Um)$$

$$\lesssim \frac{\frac{V_{p,\epsilon}^2 (\kappa^{\mathrm{T}_1})^{4p}(\log d)^{4p}}{m} Lm}{L\sqrt{\frac{V_{p,\epsilon}^2 (\kappa^{\mathrm{T}_1})^{4p}(\log d)^{4p+2}}{m}}} \log\left( \sqrt{V_{p,\epsilon}^2 (\kappa^{\mathrm{T}_1})^{4p}(\log d)^{4p}m} \right) \lesssim V_{p,\epsilon}^2 \kappa^{5p\mathrm{T}_1}(\log d)^{5p}m \tag{374}$$

Note that we have $V_{k,\epsilon_k} \lesssim \epsilon_k^{-\beta}$ by Defination D.7 , and also note that $\epsilon_k = \left( \frac{\beta(C_m+C_n)}{p+1} \right)^{\frac{1}{\beta+1}}$ form Lemma D.21, thus

$$
\begin{aligned}
\mathrm{T}_2 &\lesssim V_{p,\epsilon}^2 \kappa^{5p\mathrm{T}_1} (\log d)^{5p} m \lesssim \epsilon_k^{-\beta} \kappa^{5p\mathrm{T}_1} (\log d)^{5p} m \\
&\lesssim \left( \frac{p+1}{\beta(\sqrt{\frac{1}{m}} + \sqrt{\frac{1}{n}})} \right)^{\frac{\beta}{\beta+1}} \kappa^{5p\mathrm{T}_1} (\log d)^{5p} m \\
&\lesssim \min(m,n)^{\frac{\beta}{2(\beta+1)}} \kappa^{5p\mathrm{T}_1} (\log d)^{5p} m \lesssim \kappa^{5p\mathrm{T}_1} (\log d)^{5p} m^2
\end{aligned}
\tag{375}
$$

Moreover plugging $m = (\log d)^{4p(\beta+1)+1} d^{4p(\beta+1)\sqrt{\frac{\log \kappa}{\log d}}}$ and $\mathrm{T}_1 := \lceil \sqrt{\frac{\log d}{\log \kappa}} \rceil$ in, we get

$$
\mathrm{T}_2 \lesssim \kappa^{5p\mathrm{T}_1} (\log d)^{5p} m^2 \lesssim \kappa^{5p} (\log d)^{8p\beta+13p+2} d^{(8p\beta+13p)\sqrt{\frac{\log \kappa}{\log d}}}.
\tag{376}
$$

The proof is complete. $\qquad\square$

### D.5.18   PROOF OF LEMMA D.26

*Proof.* By Lemma D.25 we have

$$
\mathrm{T}_2 \lesssim \kappa^{5p} (\log d)^{8p\beta+13p+2} d^{(8p\beta+13p)\sqrt{\frac{\log \kappa}{\log d}}}.
\tag{377}
$$

Plugging $m = (\log d)^{4p(\beta+1)+1} d^{4p(\beta+1)\sqrt{\frac{\log \kappa}{\log d}}}$, $n = C\kappa^2 (\log d)^2 d^{1+2\sqrt{\frac{\log \kappa}{\log d}}}$ in we get

$nmd \cdot \mathrm{T} = nmd \cdot (\mathrm{T}_1 + \mathrm{T}_2)$

$$
\begin{aligned}
&\lesssim \kappa^2 (\log d)^2 d^{1+2\sqrt{\frac{\log \kappa}{\log d}}} (\log d)^{4p(\beta+1)+1} d^{4p(\beta+1)\sqrt{\frac{\log \kappa}{\log d}}} \kappa^{5p} (\log d)^{8p\beta+13p+2} d^{(8p\beta+13p)\sqrt{\frac{\log \kappa}{\log d}}} \\
&\lesssim \kappa^{5p+2} (\log d)^{12p\beta+17p+5} d^{2+(12p\beta+17p+2)\sqrt{\frac{\log \kappa}{\log d}}}
\end{aligned}
\tag{378}
$$

$$\tag{379}$$

The proof is complete. $\qquad\square$

## E   TECHNICAL BACKGROUND

### E.1   HERMITE POLYNOMIALS

We briefly introduce Hermite Polynomials and Hermite Tensors.

**Definition E.1** (1D Hermite polynomials)**.** *The $k$-th normalized probabilist's Hermite polynomial, $h_k : \mathbb{R} \to \mathbb{R}$, is the degree $k$ polynomial defined as*

$$
h_k(x) = \frac{(-1)^k}{\sqrt{k!}} \frac{\frac{d^k \mu_\beta}{dx^k}(x)}{\mu_\beta(x)},
\tag{380}
$$

*where $\mu_\beta(x) = \exp(-x^2/2)/\sqrt{2\pi}$ is the density of the standard Gaussian.*

The first such Hermite polynomials are

$$
h_0(z) = 1, h_1(z) = z, h_2(z) = \frac{z^2 - 1}{\sqrt{2}}, h_3(z) = \frac{z^3 - 3z}{\sqrt{6}}, \cdots
\tag{381}
$$

Denote $\beta = \mathcal{N}(0,1)$ to be the standard Gaussian in 1D. A key fact is that the normalized Hermite polynomials form an orthonormal basis of $L^2(\beta)$; that is $\mathbb{E}_{x\sim\beta}[h_j(x)h_k(x)] = \delta_{jk}$.

The multidimensional analogs of the Hermite polynomials are Hermite tensors.

**Definition E.2** (Hermite tensors)**.** *The $k$-th Hermite tensor $He_k : \mathbb{R}^d \to (\mathbb{R}^d)^{\otimes k}$ is*

$$
He_k(\boldsymbol{x}) := \frac{(-1)^k}{\sqrt{k!}} \frac{\nabla^k \mu_\gamma(\boldsymbol{x})}{\mu_\gamma(\boldsymbol{x})},
\tag{382}
$$

*where $\mu_\gamma(\boldsymbol{x}) = \exp(-\frac{1}{2}\|\boldsymbol{x}\|^2)/(2\pi)^{d/2}$ is the density of the d-dimensional Gaussian.*

The Hermite tensors form an orthonormal basis of $L^2(\gamma)$; that is, for any $f \in L^2(\gamma)$, one can write the Hermite expansion

$$f(\boldsymbol{x}) = \sum_{k \geq 0} \langle \boldsymbol{C}_k(f), He_k(\boldsymbol{x}) \rangle \quad \text{where} \quad \boldsymbol{C}_k(f) := \mathbb{E}_{\boldsymbol{x} \sim \gamma}[f(\boldsymbol{x}) He_k(\boldsymbol{x})] \in (\mathbb{R}^d)^{\otimes k}. \tag{383}$$

### E.2 Uniform generalization bounds

**Definition E.3.** *The empirical Rademacher complexity of a function class $\mathcal{F}$ on finite samples is defined as*

$$\widehat{\text{Rad}}_n(\mathcal{F}) = \mathbb{E}_\xi \left[ \sup_{f \in \mathcal{F}} \frac{1}{n} \sum_{i=1}^n \xi_i f(X_i) \right] \tag{384}$$

*where $\xi_1, \xi_2, \ldots, \xi_n$ are i.i.d. Rademacher random variables $\mathbb{P}(\xi_i = 1) = \mathbb{P}(\xi_i = -1) = \frac{1}{2}$. Let $\text{Rad}_n(\mathcal{F}) = \mathbb{E}[\widehat{\text{Rad}}_n(\mathcal{F})]$ be the population Rademacher complexity.*

Then we recall the uniform law of large number via Rademacher complexity, which can be found in Theorem 3.3 in Mohri et al. (2018)

**Lemma E.4.** *Assume that $f$ ranges in $[0, R]$ for all $f \in \mathcal{F}$. For any $\delta \in (0, 1)$, with probability at least $1 - \delta$ over the i.i.d. training set $S = \{X_1, \ldots, X_n\}$ we have*

$$\sup_{f \in \mathcal{F}} \left| \frac{1}{n} \sum_{i=1}^n f(X_i) - \mathbb{E}f(X) \right| \leq 2 \text{Rad}_n(\mathcal{F}) + R\sqrt{\frac{\log(4/\delta)}{n}} \tag{385}$$

Then we recall the contraction Lemma in (Vershynin, 2018, Exercise 6.7.7) to compute Rademacher complexity.

**Lemma E.5** (Contraction Lemma)**.** *Let $\varphi_i : \mathbb{R} \mapsto \mathbb{R}$ with $i = 1, \ldots, n$ be $\beta$-Lispchitz continuous. Then*

$$\frac{1}{n} \mathbb{E}_\xi \sup_{f \in \mathcal{F}} \sum_{i=1}^n \xi_i \varphi_i \circ f(x_i) \leq \beta \widehat{\text{Rad}}_n(\mathcal{F}) \tag{386}$$

The next Lemma estimates the Rademacher complexity.

**Lemma E.6.** *Let $f(\boldsymbol{x}) = \boldsymbol{a}^\top \sigma(\boldsymbol{W}\boldsymbol{x} + \boldsymbol{b})$ be our two layer neural network with our previous assumptions on the activation functions in Assumption 2. Let*

$$\mathcal{F} = \{f_{\boldsymbol{\Theta}} : \|\boldsymbol{a}\| \leq B_a, \|\boldsymbol{w}_j\| \leq C, \|\boldsymbol{b}\|_\infty \leq C\} \tag{387}$$

*where $C$ is a universal constant. Then we have the following estimation $\text{Rad}_n(\mathcal{F}) \lesssim \frac{B_a \sqrt{m}}{\sqrt{n}}$.*

*Proof.*

$$\begin{aligned}
\text{Rad}_n(\mathcal{F}) &= \mathbb{E}_{\boldsymbol{x}, \boldsymbol{\xi}} \left[ \sup_{f \in \mathcal{F}} \left| \frac{1}{n} \sum_i \xi_i (\boldsymbol{a}^\top \sigma(\boldsymbol{W}\boldsymbol{x}_i + \boldsymbol{b})) \right| \right] \\
&\leq \frac{B_a}{n} \mathbb{E}_{\boldsymbol{x}, \boldsymbol{\xi}} \left[ \sup_{f \in \mathcal{F}} \left\| \sum_i \xi_i \sigma(\boldsymbol{W}\boldsymbol{x}_i + \boldsymbol{b}) \right\| \right] \\
&\leq \frac{B_a}{n} \sqrt{\mathbb{E}_{\boldsymbol{x}, \boldsymbol{\xi}} \left[ \sup_{f \in \mathcal{F}} \left\| \sum_i \xi_i \sigma(\boldsymbol{W}\boldsymbol{x}_i + \boldsymbol{b}) \right\|^2 \right]} \\
&= \frac{B_a}{\sqrt{n}} \sqrt{\mathbb{E}_{\boldsymbol{x}} \left[ \sum_{j=1}^m \sigma(\boldsymbol{w}_j^\top \boldsymbol{x} + b_j)^2 \right]} \lesssim \frac{B_a \sqrt{m}}{\sqrt{n}}
\end{aligned} \tag{388}$$

$\square$

### E.3 UNIVARIATE APPROXIMATION

We will consider the following activation function

$$\sigma(t) = \begin{cases} 2|t| - 1, & |t| \geq 1, \\ t^2, & |t| < 1. \end{cases} \tag{389}$$

**Lemma E.7.** *There exists $v_0(a, b)$, supported on $\{\pm 1\} \times [2, 3]$, such that for any $|z| \leq 1$*

$$\mathbb{E}_{a,b}[v_0(a, b)\sigma(az + b)] = 1, \quad \sup_{a,b} |v(a, b)| \lesssim 1. \tag{390}$$

*Proof.* Let $v_0(a, b) = 12 \cdot \mathbf{1}_{a=1}(b - \frac{5}{2}) \cdot \frac{\mathbf{1}_{b \in [2,3]}}{\mu(b)}$. Then, since $z + b \geq 1$,

$$\mathbb{E}_{a,b}[v_0(a, b)\sigma(az + b)] = 6 \int_2^3 (b - \frac{5}{2})\sigma(z + b)db \tag{391}$$

$$= 6 \int_2^3 (b - \frac{5}{2})(2z + 2b - 1)db \tag{392}$$

$$= z \cdot 6 \int_2^3 (b - \frac{5}{2})db + 6 \int_2^3 (b - \frac{5}{2})(2b - 1)db \tag{393}$$

$$= 1. \tag{394}$$

The proof is complete. $\square$

**Lemma E.8.** *There exists $v_1(a, b)$, supported on $\{\pm 1\} \times [2, 3]$, such that for any $|z| \leq 1$*

$$\mathbb{E}_{a,b}[v_1(a, b)\sigma(az + b)] = z, \quad \sup_{a,b} |v(a, b)| \lesssim 1. \tag{395}$$

*Proof.* Let $v_0(a, b) = \mathbf{1}_{a=1}(-24b + 61) \cdot \frac{\mathbf{1}_{b \in [2,3]}}{\mu(b)}$. Then, since $z + b \geq 1$,

$$\mathbb{E}_{a,b}[v_1(a, b)\sigma(az + b)] = \frac{1}{2} \int_2^3 (-24b + 61)\sigma(z + b)db \tag{396}$$

$$= \frac{1}{2} \int_2^3 (-24b + 61)(2z + 2b - 1)db \tag{397}$$

$$= z \int_2^3 (-24b + 61)db + \frac{1}{2} \int_2^3 (-24b + 61)(2b - 1)db \tag{398}$$

$$= z. \tag{399}$$

The proof is complete. $\square$

**Lemma E.9.** *There exists $v_2(a, b)$, supported on $\{\pm 1\} \times [-2, 3]$, such that for any $|z| \leq 1$*

$$\mathbb{E}_{a,b}[v_2(a, b)\sigma(az + b)] = z^2, \quad \sup_{a,b} |v(a, b)| \lesssim 1. \tag{400}$$

*Proof.* First, see that

$$\int_{-2}^2 \sigma(z + b)db = \int_{-2+z}^{2+z} \sigma(b)db \tag{401}$$

$$= \int_{-1}^{-1} (-2b - 1)db + \int_{-1}^1 b^2 db + \int_1^{2+z} (2b - 1)db \tag{402}$$

$$= [-b^2 - b]_{-2+z}^{-1} + \frac{2}{3} + [b^2 - b]_1^{2+z} \tag{403}$$

$$= (z - 2)^2 + (z - 2) + \frac{2}{3} + (z + 2)^2 - (z + 2) \tag{404}$$

$$= 2z^2 + \frac{14}{3}. \tag{405}$$

Let $v_2(a,b) = \mathbf{1}_{a=1} \frac{\mathbf{1}_{b \in [-2,2]}}{\mu(b)} - \frac{7}{3} v_0(a,b)$ Then

$$\mathbb{E}_{a,b}[v_2(a,b)\sigma(az+b)] = \frac{1}{2} \int_{-2}^{2} \sigma(z+b)db - \frac{7}{3} \tag{406}$$

$$= z^2 + \frac{7}{3} - \frac{7}{3} = z^2. \tag{407}$$

The proof is complete. $\qquad\square$

**Lemma E.10.** *Let* $v(b) = -\frac{1}{2}k(k-1)(k-2)(1-b)^{k-3} \cdot \frac{\mathbf{1}_{b \in [0,1]}}{\mu(b)}$. *Then*

$$\mathbb{E}_b[v_k(b)\sigma(z+b)] = z^k \cdot \mathbf{1}_{z>0} - \frac{k(k-1)}{2}z^2 - kz - 1. \tag{408}$$

*Proof.* Plugging in $v_k(b)$ and applying integration by parts yields

$$\mathbb{E}_b[v_k(b)\sigma(z+b)] = \int_0^1 -\frac{1}{2}k(k-1)(k-2)(1-b)^{k-3}\sigma(z+b)db \tag{409}$$

$$= [\frac{1}{2}k(k-1)(1-b)^{k-2}\sigma(z+b)]_0^1 - \int_0^1 \frac{1}{2}k(k-1)(1-b)^{k-2}\sigma'(z+b)db \tag{410}$$

$$= -\frac{1}{2}k(k-1)\sigma(z) + [\frac{1}{2}k(1-b)^{k-1}\sigma'(z+b)]_0^1 - \int_0^1 \frac{1}{2}k(1-b)^{k-1}\sigma''(z+b)db \tag{411}$$

$$= -\frac{1}{2}k(k-1)\sigma(z) - \frac{1}{2}k\sigma'(z) - \int_0^1 k(1-b)^{k-1}\mathbf{1}_{|z+b|\leq 1}db \tag{412}$$

When $1 \geq z > 0$, we have

$$-\int_0^1 k(1-b)^{k-1}\mathbf{1}_{|z+b|\leq 1}db = -\int_0^{1-z} k(1-b)^{k-1}db = [(1-b)^k]_0^{1-z} = z^k - 1. \tag{413}$$

When $-1 \leq z \leq 0$, we have

$$-\int_0^1 k(1-b)^{k-1}\mathbf{1}_{|z+b|\leq 1}db = -\int_0^1 k(1-b)^{k-1}db = -1. \tag{414}$$

Since $z \in [-1,1]$, we have that $\sigma(z) = z^2$ and $\sigma'(z) = 2z$. Therefore for $z \in [-1,1]$

$$\mathbb{E}_b[v_k(b)\sigma(z+b)] = z^k \cdot \mathbf{1}_{z>0} - \frac{k(k-1)}{2}z^2 - kz - 1. \tag{415}$$

The proof is complete. $\qquad\square$

**Lemma E.11.** *There exists* $v_k(a,b)$, *supported on* $\{\pm 1\} \times [-2,3]$, *such that for any* $|z| \leq 1$

$$\mathbb{E}_{a,b}[v_k(a,b)\sigma(az+b)] = z^k, \quad \sup_{a,b} |v_k(a,b)| \lesssim poly(k). \tag{416}$$

*Proof.* We focus on $k \geq 3$. We have that

$$\mathbb{E}_b[v_k(b)\sigma(z+b)] = z^k \cdot \mathbf{1}_{z>0} - \frac{k(k-1)}{2}z^2 - kz - 1. \tag{417}$$

$$\mathbb{E}_b[v_k(b)\sigma(-z+b)] = (-z)^k \cdot \mathbf{1}_{z<0} - \frac{k(k-1)}{2}z^2 + kz - 1. \tag{418}$$

Therefore if $k$ is even

$$\mathbb{E}_b[v(b)\sigma(z+b) + v(b)\sigma(-z+b)] = z^k - k(k-1)z^2 - 2. \tag{419}$$

Let $v_k(a,b) = 2av_k(b) + k(k-1)v_2(a,b) + 2$. Then

$$\mathbb{E}_{a,b}[v_k(a,b)\sigma(az+b)] = \mathbb{E}_b[v_k(b)\sigma(z+b) + v_k(b)\sigma(z-b)] + k(k-1)z^2 + 2 = z^k. \tag{420}$$

If $k$ is odd,

$$\mathbb{E}_b[v(b)\sigma(z+b) - v(b)\sigma(-z+b)] = z^k - 2kz. \tag{421}$$

Let $v_k(a,b) = 2av_k(b) + 2av_1(a,b)$. Then

$$\mathbb{E}_{a,b}[v_k(a,b)\sigma(az+b)] = \mathbb{E}_b[v_k(b)\sigma(z+b) - v_k(b)\sigma(z-b)] + 2kz = z^k. \tag{422}$$

The proof is complete. $\qquad\square$

### E.4 Convex optimization

Denote $f(\boldsymbol{x})$ as a $C^1$ function defined in $\mathbb{R}^d$. Assume that

- There exists $m > 0$ such that $f(\boldsymbol{x}) - \frac{m}{2}\|\boldsymbol{x}\|^2$ is convex.
- $\|\nabla f(\boldsymbol{x}) - \nabla f(\boldsymbol{y})\| \le L\|\boldsymbol{x} - \boldsymbol{y}\|$.

The following result is standard and can be found in most convex optimization textbooks like Boyd & Vandenberghe (2004).

**Lemma E.12.** *There exists a unique $\boldsymbol{x}^*$ such that $f(\boldsymbol{x}^*) = \inf_{\boldsymbol{x}} f(\boldsymbol{x})$. And if we start at the point $\boldsymbol{x}^0$ and do gradient descent with learning rate $\eta$, if $\eta \le \frac{1}{m+L}$, then we will get*

$$\|\boldsymbol{x}^k - \boldsymbol{x}^*\|^2 \le c^k \|\boldsymbol{x}^0 - \boldsymbol{x}^*\|^2 \tag{423}$$

*where $c = 1 - \eta \frac{2mL}{m+L}$.*

We compute the minimal number of the required iterations.

**Corollary E.13.** *Let $\epsilon \in \big(0, \|\boldsymbol{x}^0 - \boldsymbol{x}^*\|\big)$ be a target squared-distance accuracy. In order to have $\|\boldsymbol{x}^k - \boldsymbol{x}^*\|^2 \le \epsilon^2$ it is enough to have*

$$k \gtrsim \frac{m+L}{\eta m L} \log\Big(\|\boldsymbol{x}^0 - \boldsymbol{x}^*\|^2 / \epsilon^2\Big) \tag{424}$$

The above corollary can be verified from a direct computation.

### E.5 Sphere lemmas

**Lemma E.14.** *Let $\nu \sim \chi(d)$. Then,*

$$\mathbb{E}[\nu^{2k}] = \prod_{j=0}^{k-1}(d+2j) = d(d+2)\cdots(d+2k-2) = \boldsymbol{\Theta}(d^k). \tag{425}$$

*Proof.* Write $\nu^2 \sim \chi^2(d)$ with density proportional to $x^{\frac{d}{2}-1}e^{-x/2}$ on $x > 0$. Using the Gamma function identity,

$$\mathbb{E}[\nu^{2k}] = \mathbb{E}\big[(\nu^2)^k\big] = 2^k \frac{\Gamma\big(k+\frac{d}{2}\big)}{\Gamma\big(\frac{d}{2}\big)} = \prod_{j=0}^{k-1}(d+2j). \tag{426}$$

Since $\prod_{j=0}^{k-1}(d+2j) \asymp d^k$ for fixed $k$ (e.g., by Stirling or monotonicity of the ratio), we obtain $\boldsymbol{\Theta}(d^k)$. $\qquad\square$

**Lemma E.15.** *Let $\overline{\boldsymbol{w}} \sim \mathrm{Unif}(\mathbb{S}^{d-1})$. Then,*

$$\mathbb{E}\big[\overline{\boldsymbol{w}}^{\otimes 2k}\big] = \frac{\mathbb{E}_{\boldsymbol{w}\sim\mathsf{N}(0,\boldsymbol{I}_d)}[\boldsymbol{w}^{2k}]}{\mathbb{E}_{\nu\sim\chi(d)}[\nu^{2k}]}. \tag{427}$$

*Proof.* This follows from $\boldsymbol{w} = \nu\overline{\boldsymbol{w}}$ with $\nu \sim \chi(d)$, $\overline{\boldsymbol{w}} \sim \mathrm{Unif}(\mathbb{S}^{d-1})$ independent. $\qquad\square$

**Corollary E.16.** *By Wick/Isserlis's formula for Gaussian moments,*

$$\mathbb{E}_{\boldsymbol{w}\sim\mathsf{N}(0,\boldsymbol{I}_d)}[\boldsymbol{w}^{\otimes 2k}] = (2k-1)!!\mathrm{Sym}(\boldsymbol{I}^{\otimes k}) \tag{428}$$

*Meanwhile, Lemma E.14 shows $\mathbb{E}_{\nu\sim\chi(d)}[\nu^{2k}] = \boldsymbol{\Theta}(d^k)$. Hence overall we have*

$$\mathbb{E}[\overline{\boldsymbol{w}}^{\otimes 2k}] \asymp d^{-k}\mathrm{Sym}(\boldsymbol{I}^{\otimes k}). \tag{429}$$

**Corollary E.17.** *Let $\boldsymbol{T}$ be a symmetric $p$-tensor with $\dim(span(\boldsymbol{T})) = r$. With probability at least $1 - 2d^{-D}$*

$$\|\boldsymbol{T}(\overline{\boldsymbol{w}}^{\otimes k})\|_F \lesssim \|\boldsymbol{T}\|_F \sqrt{\frac{r^{\lfloor \frac{k}{2}\rfloor}(\log d)^k}{d^k}}. \tag{430}$$

*Proof.* Denote $Y := \|\boldsymbol{T}(\overline{\boldsymbol{w}}^{\otimes k})\|_F$. We first bound the second moment. Flatten $\boldsymbol{T}$ as a matrix $\boldsymbol{M} \in \mathbb{R}^{d^{p-k} \times d^k}$ so that $\boldsymbol{T}(\overline{\boldsymbol{w}}^{\otimes k}) = \boldsymbol{M}(\overline{\boldsymbol{w}}^{\otimes k})$. Then

$$\mathbb{E}[Y^2] = \mathbb{E}\|\boldsymbol{M}(\overline{\boldsymbol{w}}^{\otimes k})\|_F^2 = \langle \boldsymbol{M}, \boldsymbol{M}C_k \rangle = \|\boldsymbol{M}C_k^{1/2}\|_F^2 \le \|\boldsymbol{M}\|_F^2 \|C_k\|_{\mathrm{op}}, \tag{431}$$

where $C_k := \mathbb{E}[\overline{\boldsymbol{w}}^{\otimes k}(\overline{\boldsymbol{w}}^{\otimes k})^\top] = \mathbb{E}[\overline{\boldsymbol{w}}^{\otimes 2k}]$ on the $d^k$-dimensional $k$-mode space. Next, from Lemma E.15 and Wick/Isserlis' formula for Gaussian even moments, $C_k = c_{k,d}\mathrm{Sym}(I^{\otimes k})$ with $c_{k,d} \asymp d^{-k}$. Restricting to the $r$-dimensional span of $\boldsymbol{T}$ (on each paired contraction) yields an extra factor $r^{\lfloor k/2 \rfloor}$ from pairing counts. Hence

$$\mathbb{E}[Y^2] \lesssim \frac{r^{\lfloor k/2 \rfloor}}{d^k}\|\boldsymbol{M}\|_F^2 = \frac{r^{\lfloor k/2 \rfloor}}{d^k}\|\boldsymbol{T}\|_F^2. \tag{432}$$

We then bound $Y$ with high probability. The map $\overline{\boldsymbol{w}} \mapsto \boldsymbol{T}(\overline{\boldsymbol{w}}^{\otimes k})$ is a homogeneous polynomial of degree $k$ restricted to $\mathbb{S}^{d-1}$. By (spherical) hypercontractivity for degree-$k$ polynomials on $\mathbb{S}^{d-1}$, Lemma E.22, for all $q \ge 2$,

$$\|Y\|_{L_{2q}(\mathbb{S}^{d-1})} \le (C\sqrt{q})^k \|Y\|_{L_2(\mathbb{S}^{d-1})}. \tag{433}$$

Combining with equation (432) gives $\|Y\|_{L_{2q}} \lesssim (C\sqrt{q})^k \|\boldsymbol{T}\|_F \sqrt{\frac{r^{\lfloor k/2 \rfloor}}{d^k}}$ and applying Markov's inequality to $Y^{2q}$ with $q \asymp D\log d$ yields

$$\mathbb{P}\left(Y \gtrsim \|\boldsymbol{T}\|_F \sqrt{\frac{r^{\lfloor k/2 \rfloor}(\log d)^k}{d^k}}\right) \le 2d^{-D} \tag{434}$$

which concludes the proof. $\square$

### E.5.1 Proof of Lemma D.34

*Proof.* As in the previous proof, flatten $\boldsymbol{T}$ into $\boldsymbol{M} \in \mathbb{R}^{d^{p-k} \times d^k}$. Then

$$\mathbb{E}\|\boldsymbol{T}(\overline{\boldsymbol{w}}^{\otimes k})\|_F^2 = \|\boldsymbol{M}C_k^{1/2}\|_F^2 \le \|\boldsymbol{M}\|_F^2\|C_k\|_{\mathrm{op}}, \quad C_k = \mathbb{E}[\overline{\boldsymbol{w}}^{\otimes 2k}] = c_{k,d}\mathrm{Sym}(\boldsymbol{I}^{\otimes k}), \tag{435}$$

so $\|C_k\|_{\mathrm{op}} \asymp c_{k,d} \asymp d^{-k}$. On the other hand,

$$\mathbb{E}\langle \boldsymbol{T}, \overline{\boldsymbol{w}}^{\otimes p}\rangle^2 = \langle \boldsymbol{T}, \mathbb{E}[\overline{\boldsymbol{w}}^{\otimes 2p}]\boldsymbol{T}\rangle = c_{p,d}\|\boldsymbol{T}\|_F^2 \quad \text{with} \quad c_{p,d} \asymp d^{-p}. \tag{436}$$

Therefore

$$\mathbb{E}\|\boldsymbol{T}(\overline{\boldsymbol{w}}^{\otimes k})\|_F^2 \lesssim \|\boldsymbol{T}\|_F^2 d^{-k} \lesssim d^{p-k}c_{p,d}\|\boldsymbol{T}\|_F^2 = d^{p-k}\mathbb{E}\langle \boldsymbol{T}, \overline{\boldsymbol{w}}^{\otimes p}\rangle^2 \tag{437}$$

which proves the claim. $\square$

### E.6 Supporting technical lemmas

**Lemma E.18** (Norm Concentration for Gaussian Samples)**.** *Let* $\boldsymbol{x}_1, \ldots, \boldsymbol{x}_n \sim \mathsf{N}(0, \boldsymbol{I}_d)$ *be i.i.d. Gaussian vectors. Then with probability at least* $1 - n\exp(-cd)$, *we have*

$$\|\boldsymbol{x}_i\| \le 2\sqrt{d} \quad \forall i \in [n] \tag{438}$$

*where $c$ is some positive universal constant.*

*Proof.* Note that $\|\boldsymbol{x}\|^2$ follows a chi-sqaured distribution with freedom $d$ and standard concentration for the chi-squared distribution, for instance Theorem 2.8.2 in Vershynin (2018) gives us

$$\mathbb{P}(\|\boldsymbol{x}_i\| \ge 2\sqrt{d}) \le \exp(-cd) \tag{439}$$

with some universal positive constant $c$. The result follows with a union bound argument. $\square$

**Lemma E.19** (Coordinate concentration parameterized by $D$)**.** *Let* $\boldsymbol{x}_1, \ldots, \boldsymbol{x}_n \overset{i.i.d.}{\sim} \mathsf{N}(0, I_d)$ *and fix* $r \in [d]$. *For any $D > 0$, with probability at least* $1 - 2nrd^{-D}$,

$$\max_{i \in [n], k \le r} |(\boldsymbol{x}_i)_k| \le 2\sqrt{D\log d}. \tag{440}$$

*Proof.* For $Z \sim \mathsf{N}(0,1)$, $\mathbb{P}(|Z| \geq t) \leq 2e^{-t^2/2}$. Setting $t = \sqrt{4D \log d}$ gives $\mathbb{P}(|Z| \geq \sqrt{4D \log d}) \leq 2d^{-2D}$. Then we do a union bound over all $i \in [n]$ and $k \leq r$. $\qquad \square$

Let the Hermite expansion of our target be $f(\boldsymbol{x}) = \sum_{k=0}^p \frac{1}{k!} \langle \boldsymbol{C}_k, He_k(\boldsymbol{x}) \rangle$ where $\boldsymbol{C}_k \in (\mathbb{R}^d)^{\otimes k}$ is the $k$-tensor defined by $\boldsymbol{C}_k := \mathbb{E}_{\boldsymbol{x}}[\nabla^k f(\boldsymbol{x})]$ and supported on $S^*$. Here $He_k$ is the unnormalized $k$-th Hermite tensor.

**Lemma E.20** (Parseval's Identity)**.**

$$1 = \mathbb{E}_{\boldsymbol{x}}[f(\boldsymbol{x})^2] = \sum_{k=0}^p \frac{1}{k!} \|\boldsymbol{C}_k\|_F^2 \tag{441}$$

As an immediate consequence of Lemma E.20 we have $\|\boldsymbol{C}_k\|_F^2 \leq k!$. Note that $f(\boldsymbol{x})$ also has the following tensor expansion

$$f(\boldsymbol{x}) = \sum_{k \leq p} \langle \boldsymbol{T}_k, \boldsymbol{x}^{\otimes k} \rangle \tag{442}$$

The following lemma shows $\|\boldsymbol{T}_k\|_F \lesssim r^{\frac{p-k}{4}} \lesssim 1$.

**Lemma E.21.** *There exist $\boldsymbol{T}_0, \ldots, \boldsymbol{T}_p$ such that*

$$f(\boldsymbol{x}) = \sum_{k \leq p} \langle \boldsymbol{T}_k, \boldsymbol{x}^{\otimes k} \rangle \tag{443}$$

*and $\|\boldsymbol{T}_k\|_F \lesssim r^{\frac{p-k}{4}} \lesssim 1$ for $k \leq p$.*

*Proof.* Note that from the Taylor expansion of $f(\boldsymbol{x})$ we have

$$\boldsymbol{T}_k = \frac{\nabla^k f(0)}{k!} = \sum_{j \leq p-k} \frac{\boldsymbol{C}_{j+k}(He_j(0))}{k! j!} = \sum_{2j \leq p-k} \frac{(-1)^j (2j-1)!! \boldsymbol{C}_{2j+k}(I^{\otimes j})}{k!(2j)!}. \tag{444}$$

We recall from the consequence of Lemma E.20 that $\|\boldsymbol{C}_k\|_F^2 \leq k!$ and therefore,

$$\|\boldsymbol{T}_k\|_F \lesssim \sum_{2j \leq p-k} \|\boldsymbol{C}_{2j+k}(I^{\otimes j})\| \lesssim r^{\frac{p-k}{4}} \lesssim 1. \tag{445}$$

$\qquad \square$

We shall use the next Gaussian hypercontractivity lemma from Theorem 4.3, Prato & Tubaro (2007).

**Lemma E.22.** *For any $\ell \in \mathbb{N}$ and $f \in L^2(\gamma)$ to be a degree $\ell$ polynomial where the input distribution $\gamma = \mathsf{N}(0, \boldsymbol{I}_d)$, for any $q \geq 2$, we have*

$$\mathbb{E}_{\boldsymbol{z} \sim \gamma}[f(\boldsymbol{z})^q] \leq C_{q,\ell} \left( \mathbb{E}_{\boldsymbol{z} \sim \gamma}[f(\boldsymbol{z})^2] \right)^{q/2} \tag{446}$$

*where we use $C_{q,\ell}$ to denote some universal positive constant that only depends on $q, \ell$.*

We also need this Spherical hypercontractivity lemma from Beckner (1992); O'Donnell (2014).

**Lemma E.23.** *Let $Y_m$ be a spherical harmonic of degree $m$ on $\mathbb{S}^{d-1}$. Then for all $1 < p \leq q < \infty$ we have*

$$\|Y_m\|_{L^q(\mathbb{S}^{d-1})} \leq \left( \frac{q-1}{p-1} \right)^{m/2} \|Y_m\|_{L^p(\mathbb{S}^{d-1})}. \tag{447}$$

*In particular, for any polynomial $P$ of degree at most $m$ and $q \geq 2$ we have $\|P\|_{L^q(\mathbb{S}^{d-1})} \leq (q-1)^{\frac{m}{2}} \|P\|_{L^2(\mathbb{S}^{d-1})}$.*

