# OpenReview forum: "Neural Networks Learn Generic Multi-Index Models Near Information-Theoretic Limit"
_ICLR.cc/2026/Conference — ICLR 2026 Poster_

### Official Review · Reviewer_5Tcn · 2025-10-27

**Soundness:** 3
**Presentation:** 4
**Contribution:** 3
**Rating:** 6
**Confidence:** 3

**Summary:**

The paper studies the problem of learning multi-index models with a two layer network using gradient descent. In particular the authors show that first training the first layer for a few step, and then freezing it an only training the second one it's possible to achieve vanishing test error with linearly many samples in the input dimension and quadratically many iterations. The key insight is that with a carefully chosen initialization (first layer with infinitesimally small weights, second layer either +1 or -1) GD is essentially performing a fixed point iteration on the empirical covariance of the data, which means that after $O(log(d))$ iterations the learned features are in the subspace of the multi-index model directions. One can then train the second layer to have good generalization.

**Strengths:**

The paper is extremely well written and does an incredibly good job at explaining the general idea of the paper and its proof directly in the main text. I believe this work to be a significant contribution to the theoretical understanding of machine learning, extending the literature on multi-index models significantly.

**Weaknesses:**

Both the data model and the training procedure is extremely idealized and ad-hoc, and it's unclear to me how these results would generalize in more realistic settings. Even within the theory landscape, the target multi-index model is just a polynomial, in contrast with what stated in the abstract. In particular training the layers one at a time is not standard practice, and this is made even worse by the need to control the number of iterations when training the first layer. Additionally the paper lacks any experimental validation or numerical check of the results.

**Questions:**

1. I am puzzled by the need to train the first layer for no more than $O(log(d))$ iterations. I find this very surprisingly low, could you comment on this?
2. More on 1: is that something that is a direct consequence of the layer-wise training? Beyond what you can prove, how much of a role do you expect layer-wise training to play in your results?
3. In Troiani et al. '25 the authors show that AMP undergoes a saddle to saddle dynamic. Do you expect GD to undergo similar dynamics? Would something like the "grand staircase" in chapter 5 of their paper happen for GD?
4. Would the results change if $g$ would be a generic leap exponent 2 function instead of a polynomial?
5. I find the total absence of numerical experiments slightly disappointing. I would greatly appreciate having a set of experiments, especially investigating theeffect of changing $T_1$.

---

> ### Author Response · Authors · 2025-11-16
> **Response(1)**
>
> We sincerely thank the reviewer for the careful reading and helpful comments. In the revised version, we made several changes.
> More concrete details will be highlighted in a global response, but we summarize a bit here.
> (1) We relaxed the loss assumptions: besides the original Lipschitz case (Assumption 3), we now introduce Assumption 3', which allows losses whose derivative has at most linear growth in the prediction. This covers square loss, \ell_1 loss, Huber loss, and pseudo-Huber loss, and Theorem 1 is stated and proved under this more general setting (see Section 2,3 and Appendix B.4).
> (2) We substantially expanded the related-work section, especially on DMFT / state evolution and spectral methods for multi-index models and feature learning.
> (3) We polished the main text and the appendix.
> (4) We added two numerical results: one illustrating that the sample-size exponent α(d) needed indeed drifts towards 1 (the n =\widetilde O(d) prediction) which matches the master theorem of this paper, and one showing that for the Hermite-4 target, square loss fails while Huber loss succeeds, in line with Assumption 5 (Section 6) and showing the importance of choosing a proper loss function.
>
> **Weakness (1)**
> For the data model, we agree that it is simple, but we would argue it is already quite standard and reasonably rich in the theory literature. Multi-index models include many familiar cases: generalized linear models, sparse parity, intersections of halfspaces, low-degree polynomials depending on a low-dimensional subspace, and two-layer neural networks with (O(1)) hidden units. These are exactly the kinds of “low-dimensional signal in high-dimensional space” structures that are used to study feature learning.
>
> For the training procedure, the two-stage, layer-wise scheme is used as a proof device. It lets us isolate the feature-learning phase to study it in depth. To check that this is not a purely artificial behavior, we added two experiments (Appendix A) where we train both layers jointly using a standard mini-batch Adam procedure, without sample splitting and without a layer-wise schedule. In these experiments:
> * the first-layer weights still learn the correct low-dimensional span, and
> * the observed sample complexity matches the (n=\widetilde O(d)) prediction.
> This suggests that the two-stage scheme is not essential algorithmically, but rather helps keep the theory tractable.
>
> ---
>
> **Weakness (2) and Question (4)**
>
> You are correct that in the current statement the target multi-index model is a polynomial, while the abstract speaks more generally. Our view is that this restriction is mostly for technical clarity rather than a real limitation.
> On a bounded domain, polynomial functions can approximate all targets. Under the same non-degeneracy conditions (as in Assumption 5), one can extend the analysis to general non-generative-staircase leap-exponent-2 targets by approximating them with polynomials and tracking the induced error (which is small). The core mechanism (a super-constant number of GD steps mimics power iteration on the right operator) remains the same. Since this extension does not change the conceptual message and only adds technical overhead, we decided to present the polynomial case in full detail and briefly explain this extension in the discussion.
>
> ---
>
> **Questions (1) and (2)**
>
> The bound on the number of first-layer iterations may look small, but from a theoretical point of view it is already large: it grows as a super-constant scale, and hence diverges with dimension. This is in contrast to almost all of the earlier work that analyzes only a fixed constant number of steps. Constant-step results are often easier to obtain within a DMFT / state evolution framework, but they are not sharp in our regime because they cannot learn a second-order feature at all without an explicit hot start.
>
> This is **not** a direct consequence of the layer-wise schedule. The layer-wise scheme only simplifies the analysis by decoupling the feature-learning phase from the second-layer dynamics. Our new experiments with joint training show that when we train both layers together, using a standard optimizer and no explicit stopping rule for the first layer, we empirically observe the same qualitative sample complexity and alignment behavior. Thus we expect the main phenomenon (super-constant, but still quite short, feature-learning phase) to hold beyond the layer-wise setting, and we view the layer-wise schedule as a theoretical tool rather than a crucial algorithmic ingredient.

---

> ### Author Response · Authors · 2025-11-16
> **Response(2)**
>
> **Question (3)**
>
> The dynamics studied in Troiani et al. (2025) for AMP involve a saddle-to-saddle evolution and a “grand staircase” structure. As far as we understand, parts of their picture rely on non-rigorous or physics-style arguments, and translating this type of description to gradient descent on two-layer neural networks is highly non-trivial. In our framework, a “generative leap-2 with staircase structure” appears very special and is an edge case in practice. We believe that, in practical settings, a suitable preprocessing or a suitable choice of loss tends to produce a non-degenerate weighted covariance of the kind we assume in Assumption 5. That is, the spectrum reveals the full signal subspace rather than a staircase that forces one to learn directions in strict sequence. It is quite hard to think up a case with a generative-staircase structure in multi-index model.
> If one insists on a true generative-leap-2 generative-staircase model (which is an edge case), then it is plausible that gradient descent could also exhibit a multi-stage feature-learning behavior, because in such a model any method has to learn the subspace “step by step” regardless of the loss or preprocessing. However, we do not yet have a precise theory for such staircase dynamics for GD, and we view this as an interesting open problem rather than something that we can firmly claim in the current paper.
>
> ---
>
> **Question (5) – Numerical experiments**
>
> We fully agree that numerical experiments are valuable. In the revised version:
> * Appendix A contains experiments where we jointly train both layers using a standard mini-batch Adam optimizer, confirming that the layer-wise schedule is not essential in practice.
> In practice, we expect the first layer to learn the relevant span within a time scale of order (\sqrt{\log d}), after which the second layer naturally takes over and fits the low-dimensional link function on the learned subspace. The precise value of the first stage training time is not something we tune in practice; it is a parameter that appears in the proof to mark the end of the early-time feature-learning window.
>
> We hope these changes and clarifications address your concerns. We are happy to further revise the paper to make the scope and implications of our results clearer and better. We believe this work is quite solid, and we want to make it as good as possible. Please let us know any concern and new suggestion.

---

### Official Review · Reviewer_jpVz · 2025-10-31

**Soundness:** 3
**Presentation:** 2
**Contribution:** 2
**Rating:** 4
**Confidence:** 4

**Summary:**

The authors investigate the problem of learning a polynomial Gaussian multi-index model using a two-layer neural network trained via gradient descent. Through a layer-wise training procedure, and under mild assumptions on the loss function and activation, they prove that there exists a suitable choice of hyperparameters for which the network efficiently learns the hidden subspace and achieves a small generalization error, close to the information-theoretic sample complexity.

**Strengths:**

The paper presents a novel theoretical analysis of the ability of shallow neural networks to learn multi-index models. The assumptions on the loss function and activation are fairly general and their necessity is discussed transparently. The connection between the early-stage training dynamics and power iteration provides insight into the trade-off between training time and full subspace recovery. The bound in Theorem 1 improves upon previous results, showing that, in principle, two-layer neural networks can generalize polynomial multi-index models with $n = \widetilde{O}(d)$, approaching the information-theoretic limit. The manuscript is well structured, with Section 4 (and B.1) offering clear intuition behind the main steps of the proof.

**Weaknesses:**

1. The analysis strongly depends on specific initialization assumptions and on the particular layer-wise training scheme. The manuscript lacks a discussion of their necessity or the validity of the claims beyond these settings.

2. Several references are missing or misplaced:

- Line 95: [1] and [2] are two other relevant references on learning thresholds of single-index models.

- Line 116: spectral methods for multi-index models (with generative exponent 2) achieving the optimal learning threshold have been studied in [3], concurrent with [Kovacevic et al. (2025)]. Both works should also be cited on line 45, together with [Damian et al. (2025)].

- [Troiani et al. (2025)]: this work shows that many multi-index models of interest can be learned with $\Theta(d)$ samples; therefore, it should be included in the discussions on lines 110 and 437. It could also be mentioned on line 72.

- [Lee et al. (2024)], [Arnaboldi et al. (2025)]: both works cite each other as concurrent and show equivalent results. Moreover, could the authors clarify the meaning of the statement “the recovered features are learned only in a quite weak sense” (lines 52 and 139)?

3. The paper is purely theoretical and lacks numerical validation. While not a weakness per se, a few illustrative examples would strengthen the claims and improve the overall presentation.

4. The bibliography should be harmonized, and missing journal or conference names should be added when necessary.



[1] Barbier et al. "Optimal Errors and Phase Transitions in High-Dimensional Generalized Linear Models"

[2] Lu, Li "Phase Transitions of Spectral Initialization for High-Dimensional Nonconvex Estimation"

[3] Defilippis et al. " Optimal Spectral Transitions in High-Dimensional Multi-Index Models"

**Questions:**

1. Following point 1 in Weaknesses, could you offer some insights on which assumptions regarding initialization and the training scheme (e.g., sample splitting, bias reinitialization) could be relaxed, under a more careful analysis, without changing the qualitative results? Which ones do you believe are necessary for the theoretical guarantees to hold?

2. At the end of Sections 1.1 and 4, you mention that choosing $T_1 = o(\log d)$ is necessary to prevent the weights from collapsing onto the top eigenvector of $\hat{\Sigma}_\ell$. However, Appendix A also states that the power-iteration approximation remains accurate only up to $o(\log d)$ steps. Could you clarify the relationship between these two statements? In particular:
 - does the breakdown of the power-iteration approximation beyond $o(\log d)$ still imply that part of the signal is suppressed, or the analysis no longer applies?
 - more generally, could you provide some intuition on what happens to subspace recovery if the first-layer training continues for longer times, when corrections to the power-iteration dynamics become non-negligible?

3. The discussion of Assumption 5 could be expanded. In particular, which polynomial target tasks would violate it for standard choices of the loss function ($\ell_1$, Huber, squared loss, etc.)? Conversely, could you give examples of reasonably natural losses for which the assumption would be satisfied for most polynomial targets?

---

> ### Author Response · Authors · 2025-11-17
> **Response(1)**
>
> We are extremely thankful for reviewer's insightful feedback.
>
> In the revised version, we made several changes.
> More concrete details will be highlighted in a global response, but we summarize a bit here.
>
>
> (1) We relaxed the loss assumptions: besides the original Lipschitz case (Assumption 3), we now introduce Assumption 3', which allows losses whose derivative has at most linear growth in the prediction. This covers square loss, \ell_1 loss, Huber loss, and pseudo-Huber loss, and Theorem 1 is stated and proved under this more general setting (see Section 2,3 and Appendix B.4).
> (2) We substantially expanded the related-work section, especially on DMFT / state evolution and spectral methods for multi-index models and feature learning.
> (3) We polished the main text and the appendix.
> (4) We added two numerical results: one illustrating that the sample-size exponent α(d) needed indeed drifts towards 1 (the n =\widetilde O(d) prediction) which matches the master theorem of this paper, and one showing that for the Hermite-4 target, square loss fails while Huber loss succeeds, in line with Assumption 5 (Section 6) and showing the importance of choosing a proper loss function.
>
> **Weakness 1 and 3, Question 1 – Initialization and training scheme**
>
> We added numerical experiments(Appendix A) to validate the master theorem. In Appendix A, we use a completely standard joint-training setup: both layers are trained together with mini-batch Adam and standard (O(1) size) initialization, without layer-wise training, sample splitting, or bias reinitialization. The empirical behavior matches our theoretical predictions. This strongly suggests that the specific initialization and the layer-wise schedule are mainly proof devices rather than essential algorithmic ingredients.
>
> From a theoretical point of view, under a finer analysis we expect that:
>
> * sample splitting and bias reinitialization can be removed;
> * the initialization can be taken larger (any scale that is (o_d(1)) should be admissible); and
> * the dynamics of joint training should yield the same qualitative picture.
>
> In the current proof, the only assumption that we believe is truly important for the guarantees is the o_d(1) size of the initialization of the first layer. This allows us to make sure the power-iteration approximation is accurate. For this reason, we keep a relatively small initialization in the theory. However, as Appendix A shows, the same phenomenon is observed in practice with standard (O(1)) initializations and without any layer-wise schedule. We now explain and expand this more explicitly in the revised version.
>
> ---
>
> **Weakness 2 – Missing and misplaced references**
>
> We have added and properly discussed all the references you pointed out in the introduction and related-work sections (including [Barbier et al.], [Lu–Li], [Defilippis et al.], [Troiani et al.], [Lee et al.], [Arnaboldi et al.]).
>
> Concerning [Arnaboldi–Dandi–Köhn et al., 2025], our use of “weakly recover” is as follows. Their result shows that a two-layer network trained by gradient descent can produce first-layer features that have nontrivial correlation with the true directions, using (\widetilde\Theta(d)) samples. However, these features are not guaranteed to span the full signal subspace, and the analysis does not show that they can be used directly to drive the final prediction error to zero in later low-dimensional fitting stages. In this sense, the recovery is “weak”. In contrast, our result aims at strong recovery in the sense that the recovered feature subspace suffices to learn the target up to vanishing error.
>
> ---
>
> **Weakness 4 – Bibliography**
>
> We have started harmonizing the bibliography in the revised version and will restore all missing journal and conference names in the final version. The current revision already follows a consistent format, and we will complete the remaining details in the final version. (We first rush out a revised version to better explain the main revision.)

---

> ### Author Response · Authors · 2025-11-17
> **Response(2)**
>
> **Question 2**
>
> Let us clarify the relation between the choice of first stage time and the validity of the power-iteration approximation.
>
> 1. The statement that the power-iteration approximation is accurate only up to (o(\log d)) iterations means that after this time scale, the dynamics does not follow a power-iteration approximation no matter how we choose the first stage learning time.
> 2. One reason for this breakdown is the scale of the initialization. With a small but not extremely small initialization (as in our main theorems), this is what people should expect since we have exponential growth. If we used an even smaller initialization, we could in principle keep the power-iteration approximation accurate for a longer time window.
> 3. Even in a regime where the power-iteration description remains valid, a pure power iteration on a fixed operator eventually collapses onto its top eigenvector. Thus, from the perspective of recovering the *whole* signal subspace, we do not want to iterate indefinitely. There is an intrinsic tension: too few steps and the noise is not sufficiently suppressed; too many steps and the spectrum becomes effectively one-dimensional.
> 4. For times beyond (o(\log d)), we currently do not have a rigorous description of the dynamics. To our knowledge, a theory of this later phase for learning with gradient descent is largely absent from the existing literature.
> 5. Intuitively, and in line with our experiments, a practical picture is as follows: once (t =\sqrt{\log d}), the first layer has already learned the relevant low-dimensional span to good accuracy, and the second-layer weights start to adapt and fit the low-dimensional link in (O(1)) additional time. In this picture, there is no need to drive the first-layer iteration all the way for long time; one can stop once the subspace is well recovered.
>
> We have added clarifying remarks along these lines in the revised version.
>
> ---
>
> **Question 3**
>
> We expanded the discussion of Assumption 5 and added a numerical illustration. In particular, in the second experiment in Appendix A we consider a fourth-order Hermite target. With square loss, there is no learning as expected. With Huber loss, the weighted covariance becomes non-degenerate, and the network succeeds in learning the target. This example shows concretely that the loss choice can move a given target from violating Assumption 5 to satisfying it.
>
> For targets under square loss, Assumption 5 is exactly the same as the non-degenerate Hessian condition in [arXiv:2206.15144, Assumption 2]. Our assumption 5 can be viewed as a refinement of that condition but adapted to all loss functions. In particular, under square loss our non-degeneracy requirement exactly matches theirs, while our results strictly and largely improve the sample complexity and provide a sharper and more precise picture.
>
> Conversely, for many “natural” losses (for instance, Huber or pseudo-Huber, square), we expect Assumption 5 to hold for most polynomial targets that do not satisfy very special patterns in their Hermite expansion. The Hermite-4 example mentioned above is representative of this situation.
>
> ---
>
> We believe that our work is quite solid and important along the literature of theoretical understanding of neural network learning. We hope that these clarifications and revisions address your concerns. We are extremely happy to further adjust the exposition or add additional results or discussions or references if there are other points you see that would help make the scope and implications of our results clearer and get a better score.

---

### Official Review · Reviewer_t3JK · 2025-11-02

**Soundness:** 3
**Presentation:** 3
**Contribution:** 3
**Rating:** 8
**Confidence:** 4

**Summary:**

This paper studies learning multi-index models with two-layer neural nets using gradient descent (GD). The authors prove that for polynomials with leap exponent at most 2, $\Theta(\sqrt{\log d})$ steps of GD on the first layer with $\tilde{O}(d)$ samples learns the latent representation, which is almost information-theoretically optimal, with a computational complexity of $\tilde{O}(nd)$ which is also nearly optimal. The main intuition is that with a super-constant number of steps, GD mimics power iteration on a certain matrix and act as a spectral method.

**Strengths:**

The finding that a super-constant number of steps can improve the sample complexity of feature learning under the multi-index model is interesting, and the proof sketch seems intuitive and can be used beyond this work.

**Weaknesses:**

1. In comparison with the recent literature, Assumption 5 is a bit restrictive. It would be interesting if one could show that gradient descent would automatically reduce leap complexity down to at most 2, similar to Lee et al., 2024 for the single-index case. I agree however that it would be technically challenging to achieve such result, and it's perhaps an interesting direction for future research in this area.

2. The Lipschitz assumption on loss (Assumption 3) rules out squared loss, and I'm not sure if it's really necessary.

3. Similar to other papers in this area, the authors have to consider a layer-wise training dynamics. Having an analysis of training both layers at the same time however is highly non-trivial.

**Questions:**

4. The authors mention the possibility of extending their analysis to non-Gaussian inputs. One relevant work here is [1], where the authors prove that GD can learn generic functions with near information-theoretically optimal sample complexity, at the expense of a potentially super-polynomial computational complexity. One interesting aspect of such result is that it can also accommodate non-isotropy in the input. In fact, I'm curious if a sample complexity better than $\tilde{O}(d)$ could be achieved if we assume a structured covariance matrix for the input that reveals some information about the target direction.

5. Wouldn't you need some assumption on the loss that enforces the prediction and ground-truth label to be close (e.g. by enforcing that the loss grows as $t$ and $y$ diverge)? Currently the loss seems too general, but for the network to learn the correct representation (which is how the proof works) I suspect that such an assumption might be necessary?

6. I might've missed this but I couldn't find the assumption $r=\mathcal{O}(1)$ written in the paper. I think one either needs this, or explicit dependence on $r$ in the bounds.


References:

[1] A. Mousavi-Hosseini, D. Wu, M. A. Erdogdu. "Learning Multi-Index Models with Neural Networks via Mean-Field Langevin Dynamics." ICLR 2025.

---

> ### Author Response · Authors · 2025-11-16
> **Response**
>
> We sincerely thank the reviewer for the careful reading and helpful comments. In the revised version, we made several concrete changes.
> More concrete details will be highlighted in a global response, but we summarize a bit here.
> (1) We relaxed the loss assumptions: besides the original Lipschitz case (Assumption 3), we now introduce Assumption 3', which allows losses whose derivative has at most linear growth in the prediction. This covers square loss, \ell_1 loss, Huber loss, and pseudo-Huber loss, and Theorem 1 is stated and proved under this more general setting (see Section 2,3 and Appendix B.4).
> (2) We substantially expanded the related-work section, especially on DMFT / state evolution and spectral methods for multi-index models and feature learning.
> (3) We polished the main text and the appendix.
> (4) We added two numerical results: one illustrating that the sample-size exponent α(d) needed indeed drifts towards 1 (the n =\widetilde O(d) prediction) which matches the master theorem of this paper, and one showing that for the Hermite-4 target, square loss fails while Huber loss succeeds, in line with Assumption 5 (Section 6) and showing the importance of choosing a proper loss function.
>
> **Weakness(1)**
> We agree that Assumption 5 is nontrivial and that it would be very interesting to show that plain gradient descent, without changing the loss, can automatically “reduce” the effective leap complexity to at most 2.
> * Conceptually, changing the loss is a controlled way to implement data preprocessing. The map (y \mapsto \ell'(0,y)) reweights the samples and effectively replaces the original covariance (\mathbb{E}[yxx^\top]) by the “preprocessed” covariance (\Sigma_\ell = \mathbb{E}[\ell'(0,y)xx^\top]). Batch-reusing or other algorithmic tricks also change the effective preprocessing seen by gradient descent. In this sense, our loss choice plays a role very similar to the “data reusing” mechanisms in [DTA+24].
> * Technically, proving that the unmodified loss plus GD automatically selects a “good” preprocessing (that yields generative leap exponent ≤ 2) is quite harder. Even in the single-index case this requires rather delicate arguments. We therefore treat Assumption 5 as a structural condition on the chosen loss/target pair and leave automatic reduction under plain GD as future work.
>
> **Weakness(2)**
> We agree that excluding square loss in the original Assumption 3 was unnatural. In the revision we do:
> * We add Assumption 3′, which only requires (|\partial_t \ell(t,y)| \le L(1+|t|+|y|)). Under Assumption 3' the entire analysis still goes through, with at most additional polylogarithmic factors in d in the final bound (Theorem 1 and Appendix B.4). This covers square loss and other standard regression losses.
> We believe that further extending to losses with higher-order polynomial growth would be routine but not conceptually enlightening; the cost is just more bookkeeping of higher moment bounds and polylog(d) factors, so we opted not to overload the presentation.
>
> **Weakness(3)**
> We fully agree that analyzing the simultaneous training of both layers is a difficult and important problem. Our theoretical proof uses layer-wise training with sample splitting as a device that lets us isolate the feature-learning phase and cleanly analyze it. To address the concern that this might be an artifact of the analysis, we added a new experiment where we train both layers jointly with a standard online SGD/Adam setup (single dataset, no layer-wise schedule). In that setting we still observe that:
> * the first-layer weights align with the true subspace, and
> * the empirical sample complexity is consistent with the predicted behavior.
> This does not replace a full theory of joint training, but it suggests that the layer-wise scheme is a proof tool rather than an essential algorithmic modification. A rigorous analysis of fully coupled two-layer dynamics at the near-linear sample scale seems to require new techniques and is beyond the scope of the current paper.
>
> **Question(4)**
> We agree that this is an interesting direction, and we are not sure about the answer.
> We are not aware of such results in a neural network learning setting for now (if we restricted to the class of efficient, polynomial time algorithm).
>
> **Question(5)**
> You are right that one needs a minimal condition on the loss. We indeed used this in the proofs but forgot to state it in the main text of the first version. In the revision, we make this explicit:
>
> * We now state the assumption (\ell(y,y)=0). This is the only extra condition we need, since convexity of (\ell) in its first argument was already assumed in the original version. The revised draft makes clear that (\ell(y,y)=0) is used in the analysis.
>
> **Question(6)**
> This is at the beginning of Section 2.

---

### Official Review · Reviewer_Quxi · 2025-11-03

**Soundness:** 2
**Presentation:** 2
**Contribution:** 3
**Rating:** 4
**Confidence:** 4

**Summary:**

The paper studies the theoretical ability of two-layer neural networks trained with gradient descent to learn so-called multi-index models of the form f(x) = g(Ux). The authors assume that the link function g is a polynomial function of a few linear projections of the input. They analyze a specific layer-wise training algorithm: first, the inner weights are updated by a few small gradient-descent steps (the “feature-learning” phase), and then the outer weights are optimized on a fresh dataset (the “readout” phase). The main theoretical result (Theorem 1) shows that, under certain regularity and non-degeneracy assumptions on the loss and activation, the resulting network achieves vanishing test error using O(d log d) samples—thus matching the information-theoretic limit of O(d) up to logarithmic factors.

The central idea is to replace the usual spectral initialization by a short sequence of small gradient-descent steps followed by one large step, and to prove that with sufficiently many samples this procedure achieves perfect learning. In other words, the first phase acts as a substitute for spectral initialization, while the final large gradient step ensures convergence to the correct subspace. This approach avoids relying on an explicit spectral start, which is the standard technique in related problems.

While this is an interesting and elegant way to re-establish an expected result, the paper in its current form contains several overstated claims, imprecise statements made to artificially broaden its generality, and misleading references to prior work. The contribution is promising and potentially valuable, but the presentation requires revision and clarification (see detailed comments below).

This issue is the main reason for my low overall score: the technical results are solid and interesting, but the framing, scope, and literature discussion are overstated. Importantly, these problems should be relatively easy for the authors to correct through a careful revision that accurately reflects what is proved and properly situates the work within the existing literature.

**Strengths:**

It was already known from Chen & Meka (2020) and related works that low-dimensional polynomial functions (i.e., generative-exponent-2 multi-index models) can be learned efficiently, near the information-theoretic limit. The present paper’s contribution is to re-establish this result within a neural-network training framework, showing that a standard two-layer network trained by gradient descent can implicitly reproduce the behavior of these earlier polynomial-learning algorithms. Replacing an explicit spectral initialization by a short sequence of small gradient steps is an elegant idea that bridges classical spectral methods with early gradient-descent dynamics.

Overall, this is a good and solid piece of work. It addresses a well-motivated and long-standing open question—whether standard gradient-descent training of neural networks can, on its own, achieve near–information-theoretic learning of structured high-dimensional targets. The main statements are interesting and relevant to the current theoretical understanding of representation learning, and the approach provides a clear and rigorous connection between classical spectral algorithms and neural-network optimization dynamics. Even though the setting is somewhat restricted, the paper gives a precise and constructive answer to this important question, and the results are likely to be of interest to researchers working on the theory of learning dynamics and high-dimensional inference.

**Weaknesses:**

I think the main result is genuinely interesting—essentially showing that, in a specific regime, the network dynamics are equivalent to a power iteration on a certain matrix. But the framing is off: the paper presents this as a broad statement about neural networks learning generic multi-index models, whereas in reality it addresses a much narrower, well-understood setting. A more honest and focused framing would make the contribution stronger and easier to appreciate. As of now, the paper currently overstates its contributions, blurs the limits of its applicability, and misrepresents parts of the existing literature.

1. Statements and Overclaimed Results

• Overstated main claim.
The paper does not fully align with either its title or its main motivating question. It is therefore misleading to describe the work as proving that “neural networks learn multi-index models near the information-theoretic limit.” In practice, the analysis concerns only non-staircase, generative-exponent-2 polynomial functions, a regime already known to be learnable efficiently since the work of Chen & Meka (2020). The authors simply replace the spectral start used in those earlier algorithms by a short gradient-descent phase with a (largely unspecified) modified loss that mimics the spectral initialization. This is a nice idea and could justify publication, but it is essential to correctly reflect the actual scope and contribution of the work.

From the title and framing, readers are led to believe that the authors have proved that “neural networks learn multi-index models near the information-theoretic limit.” In reality, it is not even clear which specific class of models this applies to (see Assumption 5). The claim is clearly overstated; even for single-index models, such a general statement would not be justified.

Indeed, the spectral-start strategy (or its gradient-descent surrogate) can only access functions that are not staircase, or more precisely, not generative-staircase in the terminology of Troiani et al. Using the rigorous interpolation method of Aubin et al., those authors showed that learning general multi-index models requires an iterative staircase mechanism—learning one direction at a time and restarting the spectral step. All these regimes are excluded by construction here. While such an extension might be possible, it is far from trivial and should not be implicitly claimed.

• On Assumptions 4 and 5.
These assumptions effectively linearize the loss and amount to introducing a spectral start. The text states:

“Recent studies on generative exponents Damian et al. (2024; 2025) show that for almost all common tasks … there exists a pre-processing function that ensures the second-order information is non-degenerate.”

For generative exponent 2, however, this result is in fact due to Yue Lu (2017) or Mondelli & Montanari (2018). Reading Damian et al. (2024, 2025) reveals no genuinely new results for exponent 2—they mainly restate existing work and extend it to higher exponents. It is therefore misleading to credit them for ideas that are much older. The fact that the paper never references the original works on spectral methods—single-index (Lu 2017; Mondelli & Montanari 2018; Maillard et al. 2020) or multi-index (Troiani et al. 2025; Defilippis et al. 2025; Kovačević et al. 2025)—is surprising. The key matrix Σ already appears in these earlier analyses. (See, for instance, Lu 2017, arXiv:1702.06435; Maillard et al., arXiv:2012.04524; Aubin et al., arXiv:2010.03460, none of which are cited despite predating most of the works referenced here.)


2. Imprecise Statements

The paper relies on a modified loss, special activation functions, and other restrictive assumptions, which make it unclear what it truly means for these models to be “learned by neural networks.” For instance, using a standard ℓ₂ loss would not work, nor would a ReLU activation. This is not a problem in itself, but given the ambitious title and claims, it is misleading.

• Choice of loss function.
The authors do not use the square loss but rather a specially chosen loss that ensures non-degeneracy. Ass.4 and 5 are so vague that it is unclear which functions are actually covered. Given the limitations above, the authors should clearly specify the excluded cases (e.g., all staircase structures). Even if such a loss can be generic, this remains a limitation that should be clearly acknowledged. Saying that “neural networks learn” is somewhat misleading, since the success of the algorithm depends on a modified loss. Similar connections have already been discussed in the literature—for instance arXiv:2403.02418, arXiv:2510.18435, and, in the DMFT context, arXiv:2509.23527 and arXiv:2103.04902—none of which are cited. (Notably, Yue Lu’s 2017 paper was itself titled “Spectral Start.”)


3. Discussion of Related Work

The discussion of related work is incomplete and historically inaccurate, which substantially weakens the scholarly framing of the paper. In fact, reading it one might believe that the relevant literature began in 2024. Several key theoretical developments that underpin the results are misrepresented or omitted. More generally, the section over-emphasizes extremely recent U.S.-based works while under-citing the earlier theoretical literature that established (i) Bayes-optimal errors and phase diagrams for GLMs, (ii) spectral thresholds (BBP-type) for pre-processed covariance methods, and (iii) AMP algorithmic limits—all directly relevant to the present theorems.

(1) Information-theoretic foundations.
The paper attributes the identification of the information-theoretic scaling n = Θ(d) for single- and multi-index models to Dudeja & Hsu (2024) and Damian et al. (2024). In fact, these results were rigorously established much earlier by Barbier, Krzakala, Macris, Miolane & Zdeborová (2017, PNAS), who derived the exact Bayes-optimal risk and phase transitions in high-dimensional GLMs using the Guerra/Talagrand interpolation method. The multi-index generalization appeared the following year in Aubin et al. (NeurIPS 2018, “The Committee Machine: Computational to Statistical Gaps in Learning a Two-Layer Neural Network”)—not to mention the non-rigorous statistical-physics literature from the 1980s that already predicted these results.

(2) Spectral and AMP results for single-index models.
The manuscript credits Mondelli & Montanari (2018) for “generic single-index models and sharply deriving the learning threshold.” Yet this threshold was obtained earlier in Barbier et al. (2017, PNAS), where Eq. (11) gives the celebrated “generative-exponent 2” transition derived via AMP. The contribution of Mondelli & Montanari was to employ pre-processing within a spectral method—but even this approach was introduced earlier by Yue M. Lu and collaborators (arXiv:1702.06435; arXiv:1811.04420), who analyzed spectral initialization and its BBP-type phase transitions for a wide class of models.

The quoted papers by Damian et al. on generative-exponent 2 for single-index models do not introduce new results; they merely restate those earlier works. Their genuine (and important) contribution was the extension to higher exponents, which is not relevant to the present paper, as it remains confined to exponent 2.

(3) Causality and scope mismatches: The following sentence is incorrect: “Subsequently, Troiani et al. (2025) … recovers the results of Mondelli & Montanari (2018).” In fact:
- Causality is reversed: The Bayes-AMP theory (Barbier et al., 2017–2019) came first; Mondelli & Montanari (2018) recovered its single-index specialization.
- Scope mismatch: Mondelli & Montanari treated single-index models only. The generalization to multi-index models followed Troiani et al. (2025), which inspired two independent recent spectral studies:
(i) Kovačević, Zhang & Mondelli (2025, COLT, arXiv:2502.01583)
(ii) Defilippis, Dandi, Mergny, Krzakala & Loureiro (2025, NeurIPS, arXiv:2502.02545)

Both derived equivalent results, and the latter appeared slightly earlier (as acknowledged in the former’s conclusion). Only the first is cited here. Because these spectral starts are central to the present paper, such omissions and reversed chronology substantially weaken the discussion.

(4) On staircase and generative-exponent structures.
The authors emphasize the staircase or leap-complexity results of Abbe et al. but ignore the more relevant generative-exponent framework. Unfortunately, no gradient-descent results are known for that case, and the strategy used here cannot address it. These limitations are never discussed, even though the paper is restricted to polynomial targets (see Assumption 1). This omission is problematic given the claimed generality of the title (see Section 1 of this review).

(5) DMFT and dynamical analyses.
No discussion of DMFT-based results appears apart from the recent Montanari & Urbani (2025). Rigorous DMFT studies have long shown that, with an appropriate “hot start,” two-layer networks can be learned efficiently (see e.g. arXiv:1710.04894; arXiv:2006.06098; arxiv:2112.07572 arXiv:2210.06591, rXiv:2303.00055). Instead, the authors cite an unpublished and identifiable work, “Andrea Montanari and Zihao Wang, Spiked Model for the Hessian of Two-Layer Neural Networks, In preparation, 2025.”

We note that this citation violates the ICLR anonymity policy: it (i) names specific, non-anonymous researchers; (ii) refers to unverifiable, unpublished work; and (iii) could reveal the submitting authors’ identities through known collaborations. Even if the reference is genuinely independent, such a citation breaks the spirit of the double-blind process and should be removed.

**Questions:**

To strengthen the paper and make its contribution both accurate and impactful, I suggest the authors consider the following revisions:

* Reframe the scope and claims: The current title and framing promise a general solution to “learning multi-index models near the information-theoretic limit,” while the actual analysis applies only to non-staircase, generative-exponent-2 polynomial functions. The paper would benefit from a more precise and honest statement of scope. A few possible titles that would better reflect the contribution: “Learning Low-Dimensional Polynomial Features via Layer-Wise Gradient Descent”

* Clarify conceptual novelty: The key insight is that gradient descent can mimic spectral initialization—an elegant technical observation. The authors should emphasize this as the main conceptual contribution, rather than claiming a fundamentally new learning phenomenon. It would be useful to explicitly contrast their method with the spectral algorithms, clarifying what is gained by expressing them in the language of neural-network training.

* Specify assumptions and limitations clearly. The paper should explicitly list the restrictions implied by Assumptions 1–5, particularly that (i) the link function must be a low-degree polynomial, (ii) the activation is smooth and quadratic near zero, and (iii) the loss must be specially chosen to ensure non-degeneracy. The authors could also discuss whether these conditions are essential or merely technical conveniences.

* Correct and balance the related-work section. The discussion should be revised to include and properly attribute earlier foundational results. Correcting the chronology and expanding the citations will make the contribution clearer and situate it fairly within the literature. It would be useful to comment explicitly on how this analysis relates to the established dynamical mean-field (DMFT) or “state evolution” approaches. Are the authors’ results a finite-time counterpart of those dynamical equations? A short discussion would make the connection to the broader learning-dynamics literature much clearer.

* Address anonymity issues: The reference to “Andrea Montanari and Zihao Wang, in preparation (2025)” should be removed or replaced by an anonymized placeholder consistent with the ICLR double-blind policy.

---

> ### Author Response · Authors · 2025-11-17
> **Response(1)**
>
> We are extremely grateful for your detailed and constructive feedback. It is clear that you are quite a domain expert, so we will address your points directly. We note that during the completion of the first draft, we are excited about the results and some of the senior authors have not finished checking some of the writing/framing details due to time limit. We made substantial changes in this revised version.
> As outlined in the global response, the revised version (i) relaxes the loss assumptions to include square loss, (ii) adds numerical experiments, (iii) extensively adjusts the framing, (iv) substantially expands the list of references, and (v) polishes the main text and appendices. We also removed the Montanari–Wang reference.
>
> First, regarding the overall framing and overclaiming: we fully agree that the original framing was too strong relative to what was actually proved. This is the main reason we are revising the title and abstract. After a careful discussion, we plan to change the title to “Neural Networks Learn Generic Multi-Index Models Near the Information-Theoretic Limit.” The intent is to keep three key messages visible in the title: (1) the model is a neural network, (2) the target is a multi-index model, and (3) the sample complexity is optimal up to leading order. Adding “generic” is meant to reflect the fact that, our result does not work for all multi-index model, but if one takes a low-dimensional polynomial and perturbs its coefficients by a small random noise, then the non-degeneracy conditions we impose are satisfied with probability one. We now make it explicit that our results apply to non-generative-staircase, generative-exponent-two polynomial link functions. This is stated in the abstract via “non-degenerate,” and then made clear and precise in the first paragraph in Section 1.1 (“Our result”) and again in Section 3. From our perspective, the restriction to polynomials is mainly technical: it allows us to use Hermite expansions and Gaussian hypercontractivity. One can in principle extend the arguments to non-polynomial links under suitable moment and approximation conditions, but this is not technically straightforward and we no longer claim this directly in the revised version. We explicitly acknowledge that staircase structures and higher leap exponents are excluded, and this line of works has not been finished.
>
> Concerning the comment that we “simply replace” spectral initialization by a short GD phase on a modified loss (The reviewer said "The key insight is that gradient descent can mimic spectral initialization—an elegant technical observation. The authors should emphasize this as the main conceptual contribution, rather than claiming a fundamentally new learning phenomenon."): our view is that the situation is more delicate. A key point is the choice of an intermediate, but still diverging, time scale. If we run only a constant number of gradient steps, we cannot recover the hidden subspace efficiently. If we run too long, the dynamics collapse onto the top eigenvector of the relevant operator, which is not enough to recover the full low-dimensional subspace. Therefore, a simple spectral initialization or its imitation will not work at all. The main insight is that there is a super-constant yet still smaller than convergence for power iteration time window where GD behaves like a power method that learns all signal directions above noise but has not yet collapsed to rank one. This is the regime and the insight we want to emphasize. In the revised version we also cover square loss, so our results now strictly strengthen those of Damian et al. (2022) in their setting. We highlight that our results concern the strong recovery problem (vanishing test error), not just weak recovery, which is quite different from that line of works.
>
> On the class of models/targets covered: with the revised loss assumptions and explicit non-degeneracy conditions, we now at least recover (and improve on) the setting of Damian et al. (2022) under square loss, while also making clear which staircase-type regimes are excluded. We agree that in the original version this was not clearly stated, and we now describe these limitations explicitly.
>
> We will discuss the loss function, the activations, the related works and others in the next response.

---

> ### Author Response · Authors · 2025-11-17
> **Response(2)**
>
> On the choice of loss and activation: the original draft used a specially designed loss without making the scope and limitations sufficiently clear, which made the claim “neural networks learn” misleading. In the revision, we (i) extend the theory to square loss, and (ii) clearly state that we require a locally quadratic activation. With these changes, we now at least recover the square-loss setting considered in prior work (Damain et al. 2022) and, under the same non-degenerate Hessian conditions, obtain strictly stronger guarantees. We now explicitly state that we exclude generative-staircase structures and explain why they are technically delicate. We also acknowledge that the choice of loss is not arbitrary. The modified loss can be seen as a preprocessing for the labels, and is essential for those spectral analysis. We have added discussions of related works that makes similar connections between loss design, preprocessing, and spectral methods. Some (but not much) of the references you mentioned (e.g. arXiv:2509.23527 arXiv:2510.18435) are non-rigorous or were unpublished (even not on Arxiv!) at the time we wrote the initial draft and submitted to ICLR; we now cite them where appropriate.
>
> Regarding missing references on spectral methods (Lu 2017; Maillard et al. 2020; Kovačević et al. 2025, and related GLM work, and others): you are right that omitting them was incorrect, and this has been corrected. We now explain that similar insights already appear in the earlier analyses (but not in a neural network setting), and we properly credit these works in the introduction and related work. We also clarify that many of these results are for single-index models; their techniques and thresholds do not directly carry over to the multi-index, feature-learning regime via neural networks, but their spectral viewpoint is clearly useful to what we do.
>
> Regarding other related works, we completely agree that the first version of this draft under-cited the earlier theoretical literature that established (i) Bayes-optimal errors and phase diagrams for GLMs, (ii) BBP-type spectral thresholds for preprocessed covariance methods, and (iii) AMP algorithmic limits, all of which are directly connected to our results. In the revised version, we have added these works and discussed their relation to our theorems in a more systematic way. We apologize that some of the junior authors were not sufficiently familiar with this literature at the time of submission; this has now been corrected.
>
> Concretely:
>
> (a). We have revised the part “Information-theoretic foundations” to properly reflect the original literature you pointed out.
> (b). We have revised the part “Spectral and AMP results for single-index models” in light of your references, and we corrected several causality and historical ordering issues.
> (c). We now discuss Troiani et al. (2025) more carefully. While we agree that this work is highly insightful, our understanding is that parts of its results rely on non-rigorous arguments (side information), especially when extending beyond a finite time horizon and when introducing “side information.” For this reason, we hesitate to treat all of its conclusions, in particular the full “grand staircase” picture, as already fully proved. Defilippis, Dandi, Mergny, Krzakala, and Loureiro (2025, NeurIPS, arXiv:2502.02545) make some of these results rigorous in the absence of staircase structure, but this still does not cover the full staircase regime which we believe is much harder.
> (d). We have also added the DMFT / state evolution papers you cited and positioned them more clearly relative to our setting. As we emphasize at the end of Section 4, these DMFT / SE analyses address finite-time dynamics, while our main result substantially on a super-constant (e.g. (O(\sqrt{\log d}))) time scale, which is necessary to learn the hidden subspace at near-linear sample complexity. Under a DMFT/SE framework, under our setup, the neurons are just trapped at initialization. For this reason, our work definitely should not be viewed as a simple finite-time analogue of those dynamical equations; rather, it addresses a different regime where existing SE / DMFT techniques are impossible to apply.
>
> We are very grateful to you for pointing out these missing and important references; your comments have substantially improved the accuracy and balance of our paper writing. Overall, we believe the revised version now presents a solid, important and impressive result, with a more careful and accurate framing of its scope and relation to prior work. We are extremely happy to further adjust the paper if you think additional clarifications would be helpful, and we are fully willing to revise again in line with any further suggestions.

---

### Author Response · Authors · 2025-11-16
**Global Response**

We sincerely thank the reviewers for their detailed and constructive feedback.

In this round of response, we have made the following revisions.

(1) **Relaxed loss assumptions.**
Beyond the original Lipschitz setting (Assumption 3), we now introduce Assumption 3', which allows loss functions whose derivative has at most linear growth in the prediction. This covers square loss, (\ell_1) loss, Huber loss, and pseudo-Huber loss. Theorem 1 is now stated and proved under this more general assumption (see Sections 2–3 and Appendix B.4). We cost an additional polylog(d) factor for this.

More generally, our techniques extend to any loss whose first and second derivatives satisfy a polynomial growth condition. This extension is technically straightforward but notationally heavy, and does not introduce new ideas, so we do not include it in full detail. The main motivation for the current revision is to explicitly include quadratic loss. With this change, our results strictly strengthen those of [arXiv:2206.15144], which to our knowledge is (at least, one of) the state-of-the-art theoretical works on learning multi-index models with neural networks up to vanishing error under generic assumptions (and achieving strong, not just weak, recovery).

(2) **Expanded related work.**
We substantially expanded the related-work discussion, especially on DMFT / state evolution and spectral methods or spectral initialization for multi-index models and feature learning. We would be happy to incorporate further references if there are important references we are still missing.

(3) **Refined framing and exposition.**
We polished the main text and appendix, and adjusted the framing to make the statements more technically precise. In particular, we plan to slightly change the title to “Neural networks learn generic multi-index models near the information-theoretic limit” or "Neural networks learn almost all multi-index models near the information-theoretic limit" in the final version and in the arxiv version (after the rebuttal) (the title of the submitted version cannot be updated at this stage).
In the abstract, we now clearly state that we require a non-degeneracy assumption on the link function. In Section 1.1 (“Our result”) we spell out this non-degeneracy condition, explain its role, and compare it carefully to the assumptions in previous works. We also emphasize this point in Sections 3 and 4 to address the reviewers’ concerns, and to make clear in what sense our results still represent a substantial improvement over the existing theory of neural network learning.

(4) **Added numerical experiments.**
We added two numerical studies (Appendix A). The first shows that the sample-size exponent (\alpha(d)) required for the test error to vanish drifts towards (1), consistent with our master theorem predicting (n = \widetilde O(d)). The second study considers a Hermite-4 target and compares different loss functions: with square loss, training fails to recover the correct representation and does not generalize, while with Huber loss (which satisfies Assumption 5) the network succeeds. This experiment, detailed in Appendix A, illustrates the importance of choosing an appropriate loss and is in line with our theoretical predictions.

We also addressed other minor issues. We will detail them in the individual response.

We believe our revision should address the concerns. However, please let us know any further suggestions and concerns, and we are willing to revise accordingly!

---

### Meta-Review · Area_Chair_i4SS · 2026-01-21

**Summary:**

The paper studies the ability of two-layer neural networks to learn multi-index functions of the form f(x) = g(Ux), where g is a polynomial of constant degree. The main technical result is that under certain regularity and non-degeneracy assumptions on the loss and activation, the neural network achieves vanishing test loss with sample complexity close to the information-theoretic limit.

The reviewers agree that the theoretical results and the proof techniques are interesting, many of them pointed out that the paper has overstated claims and misleading references, such as not stating that g is a polynomial until in the method sections. The authors acknowledged these problems and admitted that it was due to the hurry to catch the paper submission deadline.

Other main concerns are about how realistic the assumptions. E.g., squared loss was excluded by the assumption on the Lipschitz constant.

**Reviewer Concerns:**

Since the reviewers generally agree that the technical contribution is solid, it remains to see whether the authors fixed the overstated claims and the misleading references. By reading the rebuttal and the revised paper, I think the authors have fixed most problems. Concerns regarding the assumptions are also fixed. Therefore, I recommend acceptance of the paper.

I noticed that the assumption on $g$ being a polynomial is not stated in the abstract. I strongly encourage the authors to revise this in the next version, as clear and precise statements are necessary and would benefit the theory community a lot.

**Reviewer Scores:**

Reviewer Quxi may raise their score as they liked the paper and the concerns regarding claims and references are mostly resolved.

The other reviewers may raise or keep their respective scores.

---

### Decision · Program_Chairs · 2026-01-26

Accept (Poster)